# OSCAR: Orthogonal Stochastic Control for Alignment-Respecting Diversity in Flow Matching

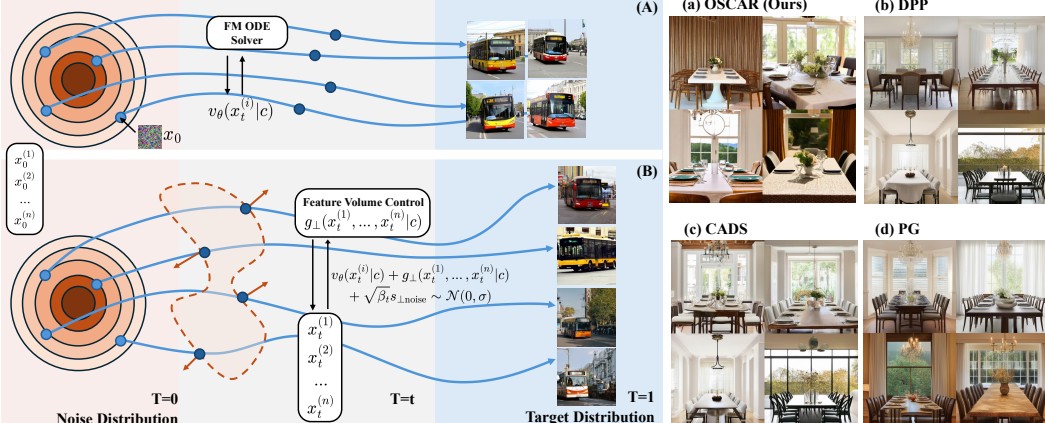

Figure 1: **Left**: A conceptual comparison of the generation process. **(A)** The standard flow matching shows independent trajectories collapsing to similar modes. **(B) OSCAR** (our method) introduces an orthogonal control mechanism that forces the interacting trajectories to diverge and cover a wider semantic space. **Right**: A qualitative comparison of generated images against strong baselines. The combined results illustrate that our method significantly increases the diversity of generations while retaining high output quality.

## Abstract

Flow-based text-to-image models follow deterministic trajectories, forcing users to repeatedly sample to discover diverse modes, which is a costly and inefficient process. We present a training-free, inference-time control mechanism that makes the flow itself diversity-aware. Our method simultaneously encourages lateral spread among trajectories via a feature-space objective and reintroduces uncertainty through a time-scheduled stochastic perturbation. Crucially, this perturbation is projected to be orthogonal to the generation flow, a geometric constraint that allows it to boost variation without degrading image details or prompt fidelity. Our procedure requires no retraining or modification to the base sampler and is compatible with common flow-matching solvers. Theoretically, our method is shown to monotonically increase a volume surrogate while, due to its geometric constraints, approximately preserving the marginal distribution. This provides a principled explanation for why generation quality is robustly maintained. Empirically, across multiple text-to-image settings under fixed sampling budgets, our method consistently improves diversity metrics such as the Vendi Score and Brisque over strong baselines, while upholding image quality and alignment.

## 1 Introduction

Recent advances in text-to-image (T2I) synthesis have unlocked unprecedented capabilities, enabling the creation of photorealistic visuals for applications ranging from digital art and design to scientific visualization (Rombach et al., 2022b; Saharia et al., 2022; Ramesh et al., 2022; Zhang et al., 2023).

Among the leading paradigms, Flow Matching (FM) and Rectified Flow (RF) have gained prominence due to their efficient inference and solid theoretical foundation (Lipman et al., 2022; Liu et al., 2022). While significant efforts have pushed the fidelity and prompt-alignment of these models to new heights (Rombach et al., 2022b; Esser et al., 2024), a critical challenge remains: a striking lack of semantic diversity.

This challenge manifests as an "illusion of variety": while users can generate numerous images by changing random seeds, the outputs often collapse to a few high-probability modes, exploring only a narrow slice of the concept's true semantic space. This tendency is not merely an incidental artifact. Still, it is exacerbated by the field's prevailing focus on aligning models with human preferences, a process that often implicitly rewards outputs that conform to common expectations at the expense of novelty (Hemmat et al., 2023; Hall et al., 2023). This limitation is then deeply rooted in the models' mechanics, as the learned flows are often contractive, pulling different initial trajectories toward similar high-density modes of the target distribution. This effect is further amplified by strong Classifier-Free Guidance (CFG), which over-weights the conditional score, narrowing the explored region of the conditional manifold, improving prompt fidelity but reducing semantic diversity (Ho & Salimans, 2022). Naively drawing more samples is an inefficient remedy, as the required Number of Function Evaluations (NFE) to find rare modes is prohibitive under finite computational budgets.

Existing approaches to mitigate this issue, whether applied during training or inference, often force a difficult compromise. Training-time solutions are often sensitive to the specific dataset and training parameters, and are typically computationally prohibitive, rendering them inapplicable to pre-trained models. While more practical, training-free methods for enhancing diversity, whether by modifying the sampling process or directly augmenting the generation dynamics, typically achieve this goal at the expense of sample quality, often introducing artifacts. Consequently, a low-overhead, training-free method that can enhance diversity while rigorously preserving quality remains a key desideratum.

To address these limitations, we introduce Orthogonal Stochastic Control for Alignment-Respecting diversity (OSCAR), a novel, training-free control mechanism for continuous-time flow-matching inference. Our core idea is to reshape the sampling dynamics to encourage trajectories under the same conditions to naturally fan out toward complementary semantics. At each time step, we first employ a finite-difference endpoint extrapolation, inspired by Heun (Süli & Mayers, 2003), to predict a local endpoint for the current state (see App. C for details). We then maximize the feature-space volume of these predicted endpoints, defined via the log-determinant of their centered Gram matrix. The gradient of this volume potential is efficiently pulled back to the latent space. To further encourage exploration, we complement this deterministic guidance with a controllable, time-scheduled stochastic noise.

The key to our method's success lies in a unifying geometric principle: both the deterministic control signal and the stochastic noise are projected to be strictly orthogonal to the base flow velocity. This ensures our guidance only provides a "lateral push" for diversity without fighting the model's forward "quality-seeking" momentum. Through comprehensive numerical comparisons against strong baselines, we demonstrate that OSCAR resolves the trade-off between the diversity and quality of the output in conditional flow-matching. Our method achieves superior performance on diversity-centric metrics, including mode coverage and intra-class entropy, while consistently preserving generation quality. The overall process and its effectiveness are conceptually illustrated in Figure 1, which showcases how OSCAR guides sampling trajectories toward more diverse outcomes.

## 2 RELATED WORK

**From Diffusion to Continuous-Time Generative Flows**    The development of modern generative models began with Denoising Diffusion Probabilistic Models (DDPM) (Ho et al., 2020), which achieved stable training and strong fidelity by simulating a progressive denoising process. A key drawback of the canonical sampler, however, is its high computational cost, requiring a large neural function evaluations (NFE) to produce a high-quality image. This limitation spurred two complementary research streams. On the quality and alignment side, researchers steadily improved realism and prompt adherence by scaling model capacity, strengthening text encoders, and incorporating preference alignment objectives (Meng et al., 2021; Saharia et al., 2022; Esser et al., 2024; Xu et al., 2023). On the efficiency side, few-step Ordinary Differential Equation (ODE) solvers and consistency-based models substantially reduced the NFE while preserving fidelity (Song et al., 2020; Lu et al., 2022; Song et al., 2023; Luo et al., 2023). While continuous-time generative frameworks

like Flow Matching (FM) and Rectified Flow (RF) have greatly improved the efficiency and fidelity of modern generative models (Lipman et al., 2022; Liu et al., 2022), the field's primary focus has remained on these aspects. Consequently, diversity and mode coverage have been overlooked, particularly under strong CFG, which compresses the solution space and causes generations to collapse to similar high-probability modes (Ho & Salimans, 2022; Sadat et al., 2023).

**Training-Time Diversity Enhancement**    One line of work enhances diversity during training, often inspired by reinforcement learning. For instance, Miao et al. (2024) proposes a framework that employs a reward model to score the diversity of generated images. However, this method's effectiveness hinges on a pre-defined reference distribution of real images, a requirement that can be impractical to satisfy and limits the approach's generality. Other methods frame the reverse sampling process as a multi-step Markov Decision Process and apply policy gradient optimization to the entire sampling trajectory (Zhang et al., 2024; Black et al., 2023; McAllister et al., 2025). While powerful, this web-scale training paradigm is not only computationally prohibitive, but its effectiveness is also highly sensitive to the specific dataset and training parameters. Our work is therefore situated within the more practical and versatile paradigm of training-free, inference-time enhancement.

**Inference-Time Diversity: Sampling Strategies vs. Gradient-Based Guidance**    Inference-time methods for diversity enhancement can be broadly categorized into two paradigms: sampling strategies and gradient-based guidance. The first class indirectly broadens exploration by modifying the sampling process. For instance, Sehwag et al. (2022) proposes biasing sampling toward low-density regions of the data manifold, which may lead to a distribution shift. Building on this idea, Condition-annealed diffusion sampling (CADS) dynamically anneals the conditioning signal to encourage exploration in early sampling stages (Sadat et al., 2023). Unlike the scheduling-based CADS, whose performance can be highly dependent on the precise schedule and scale of the injected noise, our method introduces noise strictly in the subspace orthogonal to the base flow velocity. This quality-preserving design makes our method far less dependent on the injection schedule and thus more robust across CFG scales.

A second paradigm, gradient-based guidance, takes a more direct approach by actively pushing samples apart using diversity objectives. While early methods, such as Particle Guidance (PG), applied repulsion directly in the latent space, this can conflict with the model's quality-generating flow (Corso et al., 2023). A significant evolution is DiverseFlow, which elevates the guidance objective to a semantic feature space operating on predicted endpoints (Morshed & Boddeti, 2025). Our work builds on this idea by introducing a more efficient $O(K^2)$ objective that largely reduces DiverseFlow's $O(K^3)$ complexity cost, and, more critically, a set of rigorous geometric constraints for fidelity preservation. Orthogonal to these deterministic guidance methods, another line of thought replaces the ODE solver with a Stochastic Differential Equation (SDE) solver inspired by diffusion models (Liu et al., 2025). However, such generic stochasticity is not explicitly designed to enhance semantic diversity. Our method is unique in that it integrates the strengths of both approaches: we utilize principled semantic guidance while also injecting staged, controllable noise, both of which are constrained to be orthogonal to the base flow, ensuring a safe and effective diversity boost.

## 3    PROBLEM FORMULATION

Our work is situated within the framework of continuous-time generative models, particularly those based on FM and RF. Specifically, the RF framework defines the path between a real data sample $x_1$ and a noise sample $x_0$ via a simple linear interpolation $x_t = (1-t)x_1 + tx_0$, for $t \in [0, 1]$. A model is then trained to learn a velocity field $v_\theta(x_t, t \mid c)$ by minimizing an objective function:

$$\mathcal{L}(\theta) = \mathbb{E}_{t, x_0, x_1} \left[ \|(x_0 - x_1) - v_\theta(x_t, t \mid c)\|^2 \right]$$

This process drives $v_\theta$ to fit the constant target velocity field connecting the data and noise distributions. During inference, this learned velocity field $v_\theta$ guides a deterministic ODE from noise to data:

$$\frac{dx_t}{dt} = v_\theta(x_t, t \mid c), \qquad t \in [1, 0]$$

However, the deterministic and often contractive nature of the inference ODE suffers from a significant drawback in the form of a striking lack of set-level diversity. We visually demonstrate this inherent

limitation in a 2D toy experiment, as shown in Fig. 2. While the standard flow matching process successfully converges to the means of the nine Gaussian components, the generated samples are overly concentrated around these points, failing to capture the full support of the distribution. In contrast, our method maintains alignment with all nine modes while significantly improving diversity, yielding a sample distribution that better covers the variance of each component while preserving the cluster structure. We acknowledge that this active repulsion introduces a controlled bias relative to the exact target density(see in Appendix B). However, this remains minimal in standard regimes. We also provide a detailed analysis of this trade-off in the high-particle limit in Appendix E.1.

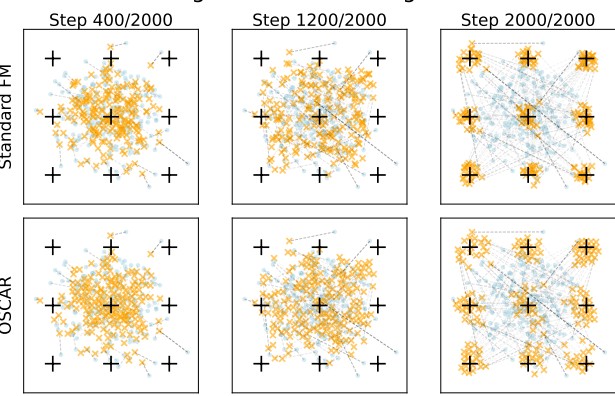

Flow Matching vs. OSCAR — Progress Over Time

Figure 2: **2D toy visualization of diversity enhancement on a 3x3 GMM.** The top row shows the standard Flow Matching baseline, while the bottom row shows our method. Columns are snapshots at early, middle, and final generation steps. Black "+" markers indicate the GMM means, blue dots are the shared initial particles, and orange "x" markers are the particle locations at the given step. Our method yields more uniform within-mode coverage and better-separated modes.

The central challenge in overcoming this limitation lies in the well-known trade-off between diversity and quality. Naively augmenting the learned ODE dynamics, for instance, by adding arbitrary noise or simple repulsion forces, can easily disrupt the model's carefully calibrated path to high-fidelity samples, often introducing artifacts and degrading generation quality.

# 4 METHODOLOGY

In this work, we evaluate set-level diversity, defined as the ability of a model to generate a semantically varied set of a given size for a fixed condition, purely by changing the initial random seed. Therefore, our goal is to enhance set-level diversity by augmenting the base dynamics without retraining the pretrained velocity field $v_\theta$. We achieve this by introducing two components: (1) a deterministic, geometry-aware control signal $g(x_t, t)$, designed to drive samples apart in a semantic feature space, and (2) a controllable stochastic noise term $\sqrt{\beta(t)}\, dW_t$ to inject quality-preserving uncertainty. This transforms the original ODE into a controlled SDE that governs our sampler:

$$dx_t = \left[ v_\theta(x_t, t \mid c) - g(x_t, t \mid c) \right] dt + \sqrt{\beta(t)}\, dW_t, \tag{1}$$

where $W_t$ is standard Brownian motion. The principled design of the control signal $g$ and its associated fidelity safeguards are detailed in the following sections.

## 4.1 ENDPOINT-AWARE FEATURE-VOLUME CONTROL

To drive the set of trajectories $\{x_t^{(i)}\}$ towards diverse configurations, our control mechanism operates not on the current states, but on their local estimates of the final trajectory endpoint $\{x_0^{(i)}\}$ in a semantic feature space. The core idea is to define a differentiable, permutation-invariant set energy [1] $\mathcal{E}_s(Z)$ for a collection of endpoint features $Z = [z_1; \ldots; z_m]$. This energy is designed to be low when features are spread out and high when they are clustered. We then steer the features towards a lower energy state using a stochastic control law:

$$dZ = -\gamma(t)\, \nabla_Z \mathcal{E}_s(Z)\, dt + \sqrt{\beta(t)}\, dW_t, \tag{2}$$

where $Z \in \mathbb{R}^{m \times D}$ stacks the features, and schedules $\gamma(t)$ and $\beta(t)$ modulate the diversity-seeking drift and exploration. The key components of this control law are derived as follows.

---

[1]The term "energy" is used by analogy to physics: we define an objective function where lower values correspond to more diverse sets, and its negative gradient provides a 'force' that drives diversity.

**Endpoint-Aware Features** At any time $t$, we form an endpoint-aware feature by first applying a one-step predictor $\hat{\psi}(x, t)$ and then a semantic encoder $\phi : \mathcal{X} \to \mathbb{R}^D$ (e.g., a pretrained image tower):

$$z_i(t) = \phi(\hat{\psi}(x^{(i)}(t), t)), \qquad Z(t) = [z_1(t); \ldots; z_m(t)].$$

For the predictor $\hat{\psi}$, we use a finite-difference endpoint extrapolation, which provides an accurate local estimate of the trajectory's endpoint without any extra NFE.

**Set Energy as Feature Volume** We instantiate the set energy $\mathcal{E}_s(Z)$ using a log-determinant objective that corresponds to the volume spanned by the features. To avoid manual centering, we use a trace-stabilized form:

$$\mathcal{E}_s(Z) = -\tfrac{1}{2} \log \det \left( I + \tau ZZ^\top + \varepsilon_{\mathrm{tr}} I \right), \tag{3}$$

where we fix $\tau = 1$ in practice and $\varepsilon_{\mathrm{tr}}$ is a trace-based stabilizer valued as $\varepsilon \frac{\mathrm{tr}(ZZ^\top)}{m}$ with negligible $\varepsilon$. Minimizing this energy is equivalent to maximizing the regularized volume of the parallelepiped formed by the feature vectors $\{z_i\}$.

**Pulling the Gradient Back to State Space** The feature-space drift from Eq.2 is mapped back to the sampler's state space $\mathcal{X}$ to form the control signal $g(x, t)$. We achieve this via the chain rule:

$$g_i(x_i, t) = \left( J_x \hat{\psi}(x_i, t) \right)^\top \left( J_u \phi(u) \right)^\top \left[ \nabla_Z \mathcal{E}_s(Z) \right]_i \Big|_{u = \hat{\psi}(x_i, t)}, \tag{4}$$

where $[\cdot]_i$ denotes the gradient component for the $i$-th sample. Equation 4, which defines the control signal $g(x, t)$ used in our final controlled SDE, is obtained by pulling back the feature-space gradient via the chain rule and implemented efficiently per sample using two reverse-mode vector–Jacobian products (VJPs) without explicitly forming Jacobians (see full derivation in Appendix B, Lemma 1).

### 4.2 Fidelity Safeguards

A powerful diversity signal is not sufficient. It must be applied in a manner that respects the underlying quality manifold learned by the pretrained model. To this end, we introduce two principled, geometry-aware constraints that ensure our control mechanism is quality-preserving.

#### 4.2.1 Orthogonal Projection

Simply adding the control signal $g(x_t, t)$ to the base velocity $v_\theta$ can create conflicts, as the direction for maximal diversity may oppose the direction for maximal fidelity. To resolve this, we enforce that the control operates strictly in a subspace that is orthogonal to the primary alignment/quality direction. We define this direction as the base velocity itself, $q(x, t) \approx v_\theta(x, t \mid c)$, which empirically points towards the conditional data manifold.

We then construct a projection operator $\Pi_\perp$ that nullifies any component of our control signal parallel to this quality direction:

$$g_i^\perp(t) = g_i(t) - \lambda \frac{g_i(t)^\top v_i(t)}{\|v_i(t)\|^2 + \delta} v_i(t).$$

where $\lambda \in [0, 1]$ controls the strictness of the orthogonality. By applying this projection to both the deterministic control $g$ and the stochastic noise $dW_t$, we ensure our guidance only performs a "lateral spread" to enhance diversity, without interfering with the model's primary, "forward" trajectory towards a high-fidelity output (See theoretical analysis in Appendix B).

#### 4.2.2 Redundancy-Aware Reweighting

Instead of applying a uniform, hard cap to the control signal, we employ a more adaptive and fine-grained mechanism to modulate its strength on a per-sample basis. This is achieved by reweighting the set energy $\mathcal{E}_s$ based on the geometric arrangement of the endpoint features. Specifically, we compute per-sample leverage scores $s_i$ from the inverse of the feature Gram matrix $K = ZZ^\top$: $s_i = \left[ (K + \varepsilon I)^{-1} \right]_{ii}$. The score $s_i$ is inversely proportional to how "central" or "redundant" a sample is within the set. Geometrically, under-covered samples receive higher scores. These scores are then used to form normalized weights $w_i$, which reweight the Gram matrix used in our volume objective:

$$\tilde{K} = W^{1/2} K W^{1/2}, \quad \text{where} \quad W = \mathrm{diag}(w_1, \ldots, w_m). \tag{5}$$

where $w_i = \frac{s_i^\alpha}{\sum_j s_j^\alpha}$. By using this reweighted Gram matrix $\tilde{K}$ inside the log-determinant energy (Eq. 3), our method naturally allocates more guidance strength to the under-represented samples that need it most, while attenuating the influence of dominant, redundant samples. This "push weak, not strong" principle provides a more stable and efficient control than a uniform cap. The complete, step-by-step implementation of our guided sampler is presented in Algorithm 1.

---

**Algorithm 1 The OSCAR Sampling Algorithm**

---

1: **Input:** Initial noise samples $\{x_0^{(i)}\}_{i=1}^m \sim \mathcal{N}(0, I)$; condition $c$; models: velocity field model $v_\theta$, feature encoder $\phi$, endpoint predictor $\hat{\psi}$; time grid $0 = t_0 < \cdots < t_T = 1$.

2: **Hyperparameters:** Stability $\varepsilon$, volume scale $\tau$, orthogonality $\lambda \in [0, 1]$, leverage exponent $\alpha \in [0.5, 1]$, noise exponent $p$.

3: **Output:** Final samples $\{x_1^{(i)}\}_{i=1}^m$.

4: **for** $\ell = 0$ **to** $T - 1$ **do**

5:      Let $x_i \leftarrow x_{t_\ell}^{(i)}$ be the current sample at time $t_\ell$.

6:      Set step size $\Delta t \leftarrow t_{\ell+1} - t_\ell$; evaluate base velocity $v_i \leftarrow v_\theta(x_i, t_\ell \mid c)$.

7:      **Predict endpoints:** Use extrapolation $\hat{\psi}$ to get $\hat{x}_i^{\text{end}}$ from $(x_i, v_i, t_\ell)$.

8:      **Compute diversity gradient $g_i$ (Sec. 4.1):**

9:          Encode endpoints $z_i \leftarrow \phi(\hat{x}_i^{\text{end}})$, stack into matrix $Z = [z_1; \ldots; z_m]$

10:         Compute leverage scores $s_i$ and weights $w_i$ from Gram matrix $ZZ^\top$ (Eq. 5).

11:         Compute reweighted feature-space gradient $G^Z = \nabla_Z \mathcal{E}_s(Z)$.

12:         Pull back to state space: $g_i \leftarrow (J\hat{\psi})^\top (J\phi)^\top [G^Z]_i$ (Eq. 4).

13:      **Apply fidelity safeguards(Sec. 4.2):**

14:         Project determin control: $g_i \leftarrow \left(g_i - \lambda \frac{\langle g_i, v_i \rangle}{\|v_i\|^2 + \varepsilon} v_i\right) \times \min\left(1, \frac{\|v_i\|}{\|g_i\| + \varepsilon}\right)$.

15:         Project stochastic noise: $\xi_i \sim \mathcal{N}(0, I)$; $\xi_i^\perp \leftarrow \xi_i - \frac{\langle \xi_i, v_i \rangle}{\|v_i\|^2 + \varepsilon} v_i$; $\eta_i \leftarrow \sqrt{t_\ell^p |\Delta t|}\, \xi_i^\perp$.

16:      **Perform controlled Step(Apdix C.1):**

17:         Effective velocity: $v_i^{\text{eff}} \leftarrow v_i - \gamma(t_\ell)\, g_i$.

18:         Update: $x_{t_{\ell+1}}^{(i)} \leftarrow x_i + \Delta t\, v_i^{\text{eff}} + \eta_i$.

19: **end for**

20: Let final samples be $x_1^{(i)} \leftarrow x_{t_T}^{(i)}$.

---

## 5 EXPERIMENTS AND RESULTS

**Setup** We evaluate three conditional generation settings: **(i)** class-conditional generation on COCO (Lin et al., 2014); **(ii)** text-to-image (T2I) using a pretrained Stable Diffusion backbone in Flow-matching framework; and **(iii)** attribute-conditioned generation, where we assess the quality and diversity of images generated for specific attributes. All methods are applied at inference time to the same pretrained and frozen backbone model, ensuring a fair comparison.

**Fidelity and Alignment** To quantify performance across these setups, our evaluation is structured around three core axes. We first assess the fidelity and alignment of generated samples. This includes the standard Fréchet Inception Distance (FID) (Heusel et al., 2017) for perceptual realism against the reference distribution. For semantic and human-preference alignment, we measure CLIP-Score for text-prompt agreement and Image Reward (Xu et al., 2023) for alignment with human aesthetic preferences. Finally, to assess perceptual quality and detect artifacts, we employ the no-reference Blind/Referenceless Image Spatial Quality Evaluator (BRISQUE) (Mittal et al., 2012) as well as the learned CLIP-IQA (Wang et al., 2023) metric.

**Distributional Coverage** Beyond the quality of individual images, we measure distributional coverage to understand how well the model captures the breadth of the true data manifold. We plot Improved Precision-Recall (IPR) curves (Kynkäänniemi et al., 2019) and report the iso-precision $\Delta$Recall, which directly quantifies coverage expansion at a comparable fidelity level. This is complemented by Coverage@$\tau$ on pre-clustered real-data features to evaluate discrete mode discovery. The Does-It Metric (DIM) (Teotia et al., 2025) further probes coverage by measuring the balanced generation of attributes from coarse prompts.

**Intra-Set Diversity** Finally, for the crucial user-facing scenario of generating multiple candidates from a single prompt, we evaluate intra-set diversity. We report pairwise $1 - $ MS-SSIM (Wang et al.,

2003) to capture perceptual and structural variation. Our primary metric here is the reference-free Vendi Score (Friedman & Dieng, 2022), which reflects the "effective number" of distinct items in a set. We report both Inception and pixels representations to disentangle low-level from semantic diversity. The Can-It Metric (CIM) (Teotia et al., 2025) confirms that this diversity is controllable, assessing the ability to produce distinct outputs when explicitly conditioned on different attributes.

**Implementation Details** To ensure a fair comparison, all evaluated methods, including our own and all baselines, are training-free and applied on the same base model. More implementation details can be found in Appendix C.

## 5.1 MAIN RESULTS

**Fidelity-Coverage Trade-off** Our quantitative evaluation shows that our method improves the trade-off between generation fidelity and diversity, which is illustrated by the Precision-Recall for Distributions (PRD) curves in Figure 3 (see more results in Appendix D.1). While baselines such as DPP and PG suffer from a sharp precision collapse as recall begins to increase, our method maintains a more robust curve, demonstrating a superior ability to expand sample diversity without a severe penalty to fidelity. This overall advantage is confirmed by our method achieving the highest Area Under the Curve (AUC) across all CFG scales. This result underscores our method's ability to effectively explore the data manifold for diverse samples while maintaining high-quality outputs.

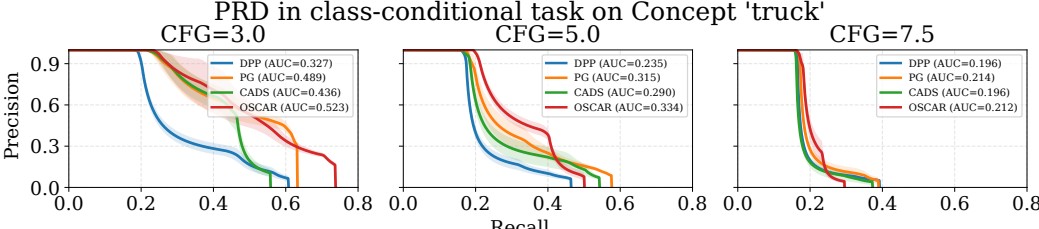

Figure 3: PRD curves on the "truck" concept, comparing methods across different CFG levels. For most CFG settings, our method's curves are shifted toward the top-right, indicating a superior precision–recall trade-off.

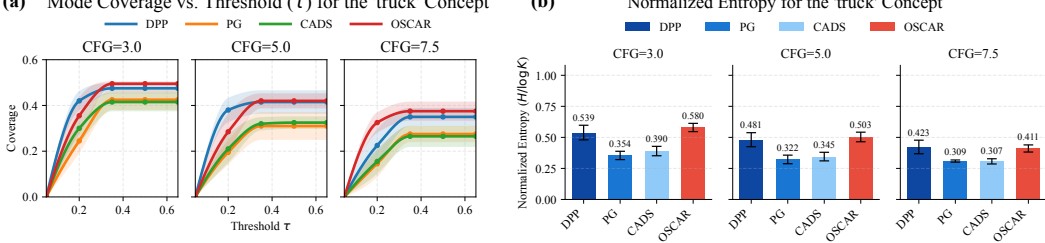

Figure 4: **(a)** Mode Coverage vs. Threshold ($\tau$) for the 'truck' concept, comparing our method against baselines at three fixed CFG levels. Higher curves indicate a larger fraction of real-data clusters are covered, signifying superior mode discovery. Shaded bands denote the $\pm 1$ standard deviation across multiple seeds and prompts. **(b)** Normalized entropy for the 'truck' concept, comparing methods at different CFG levels. Higher bars indicate a more uniform distribution of samples across discovered modes, signifying less redundancy. Error bars denote the $\pm 1$ standard deviation.

**Reference-Free Diversity and Fidelity** As detailed in Table 1, our approach enhances diversity while maintaining state-of-the-art sample quality and prompt adherence. Our method is the clear leader in diversity, achieving the highest scores across all CFG levels for both Vendi Score Pixel($VS_p$), which measures low-level variation, and Vendi Score Inception($VS_I$), which captures semantic differences. Crucially, this diversity is not an artifact of degraded quality. Our method delivers top-tier generation fidelity, evidenced by FID and Brisque scores, while CLIP Scores confirm that text-image alignment is rigorously maintained. To ensure the robustness of these findings, all reported metrics are aggregated over multiple random seeds and synonymous prompts for each concept (see more results and the example of our prompt structure in Appendix D.3).

**Intra-Class Mode Discovery** The robust performance stems from our method's ability to discover and represent fine-grained, intra-class modes more effectively than baselines. This is demonstrated through a two-fold analysis. First, the mode coverage results in Figure 4(a) show that while our method is competitive at stricter thresholds, it consistently converges to a higher final coverage plateau

Table 1: Quantitative comparison of our method against baselines across different CFG levels for the "truck" concept, you can see other concept results in Appendix D. Our method consistently improves diversity metrics while achieving state-of-the-art performance on quality and alignment scores.

| Method | Guidance Scale (CFG) | | | Guidance Scale (CFG) | | |
|---|---|---|---|---|---|---|
| | 3.0 | 5.0 | 7.5 | 3.0 | 5.0 | 7.5 |
| | Brisque ↓ | | | 1 − MS-SSIM(%) ↑ | | |
| FM-SD3.5 | 19.58 ± 1.95 | 23.40 ± 2.27 | 33.11 ± 1.82 | 83.93 ± 2.05 | 82.62 ± 2.63 | 80.46 ± 2.82 |
| PG | 40.11 ± 5.83 | 40.34 ± 6.11 | 47.66 ± 3.92 | 84.60 ± 2.76 | 83.76 ± 2.37 | 81.07 ± 1.22 |
| CADS | 21.15 ± 0.76 | 23.97 ± 1.59 | 32.73 ± 1.96 | **86.12 ± 2.76** | 83.82 ± 3.10 | 82.71 ± 2.53 |
| DPP | 18.73 ± 0.91 | 24.51 ± 1.98 | 33.88 ± 0.98 | 85.75 ± 1.32 | 83.42 ± 1.86 | 81.90 ± 1.73 |
| **Ours** | **18.50 ± 1.62** | **21.72 ± 1.31** | **27.80 ± 0.85** | 84.80 ± 1.03 | **83.91 ± 1.04** | **83.22 ± 1.21** |
| | Vendi Score Pixel ↑ | | | Vendi Score Inception ↑ | | |
| FM-SD3.5 | 2.48 ± 0.19 | 2.31 ± 0.11 | 2.23 ± 0.11 | 4.55 ± 0.28 | 3.86 ± 0.13 | 3.76 ± 0.14 |
| PG | 2.45 ± 0.18 | 2.30 ± 0.13 | 2.25 ± 0.08 | 4.80 ± 0.18 | 4.22 ± 0.14 | 3.81 ± 0.11 |
| CADS | 2.63 ± 0.09 | 2.34 ± 0.12 | 2.25 ± 0.10 | 4.58 ± 0.28 | 3.88 ± 0.14 | 3.71 ± 0.14 |
| DPP | 2.48 ± 0.09 | 2.30 ± 0.09 | 2.17 ± 0.03 | 4.24 ± 0.32 | 3.58 ± 0.23 | 3.15 ± 0.10 |
| **Ours** | **2.66 ± 0.18** | **2.47 ± 0.11** | **2.34 ± 0.10** | **4.93 ± 0.33** | **4.34 ± 0.13** | **4.08 ± 0.04** |
| | CLIP-IQA ↑ | | | CLIP Score ↑ | | |
| FM-SD3.5 | 6.26 ± 0.61 | 6.48 ± 0.44 | 6.52 ± 0.48 | 26.97 ± 0.58 | 26.16 ± 0.52 | 26.02 ± 0.54 |
| PG | 6.22 ± 0.57 | 6.43 ± 0.47 | 6.48 ± 0.42 | 27.15 ± 0.61 | 26.57 ± 0.55 | 26.52 ± 0.53 |
| CADS | **6.61 ± 0.57** | 6.65 ± 0.51 | 6.69 ± 0.46 | 26.89 ± 0.68 | 26.53 ± 0.62 | 26.55 ± 0.53 |
| DPP | 6.29 ± 0.58 | 6.42 ± 0.50 | 6.44 ± 0.46 | 27.06 ± 0.78 | 26.68 ± 0.67 | 26.46 ± 0.59 |
| **Ours** | 6.57 ± 0.62 | **6.76 ± 0.51** | **6.79 ± 0.48** | **27.20 ± 0.55** | **26.76 ± 0.55** | **26.56 ± 0.48** |
| | FID ↓ | | | Image Reward ↑ | | |
| FM-SD3.5 | 166.18 ± 1.10 | 165.51 ± 0.91 | 166.08 ± 0.65 | 0.40 ± 0.32 | 0.52 ± 0.26 | 0.60 ± 0.27 |
| PG | 167.34 ± 2.12 | 166.50 ± 2.39 | 165.39 ± 2.03 | 0.38 ± 0.38 | 0.54 ± 0.30 | 0.60 ± 0.29 |
| CADS | 165.84 ± 0.95 | 166.13 ± 1.22 | 167.34 ± 1.03 | **0.45 ± 0.32** | 0.56 ± 0.27 | 0.64 ± 0.26 |
| DPP | 166.51 ± 1.52 | 165.89 ± 0.72 | 166.21 ± 1.28 | 0.34 ± 0.35 | 0.49 ± 0.28 | 0.55 ± 0.27 |
| **Ours** | **165.40 ± 0.94** | **164.75 ± 0.66** | **165.31 ± 0.79** | 0.43 ± 0.35 | **0.57 ± 0.29** | **0.65 ± 0.28** |

across all CFG scales. This indicates that our method ultimately identifies a broader set of modes. Furthermore, the normalized entropy results in Figure 4(b) highlight our method's clear superiority in distributing samples among these modes. It achieves a substantially and consistently higher entropy, providing strong evidence that our generated samples are more uniformly distributed, leading to a less redundant and more semantically rich output set (see additional results in Appendix D.2).

**Attribute-Level Diversity** At the semantic level, we analyze our method's ability to balance default-mode diversity with explicit controllability. For this evaluation on the *bus* concept, we generated images for each core attribute using the structured prompt system detailed in Appendix D.4. The results in Figure 5(a) show that while baseline methods exhibit strong biases, our method consistently achieves scores closer to zero, signifying a more balanced generation. Crucially, this improved balance does not compromise instruction-following, as Figure 5(b) shows our method maintains high, competitive CIM scores. This demonstrates that our method successfully overcomes the typical trade-off between default-mode diversity and explicit control.

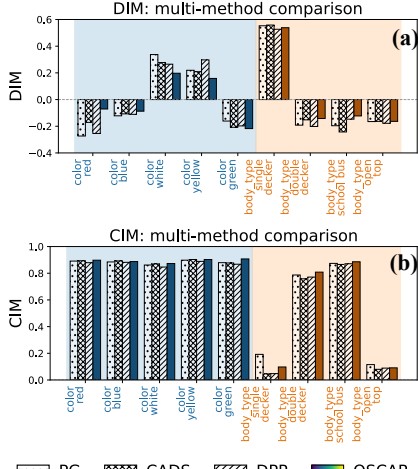

Figure 5: Evaluation of attribute-level diversity and generalization on the *bus* concept. DIM quantifies the balance of attributes generated from coarse prompts (a score closer to 0 indicates better balance), while CIM assesses the ability to follow explicit attribute requests (a higher score is better).

## 5.2 ABLATION STUDY

This section explores the role of the most important safeguards and parameters in our method on the final quality and diversity of generated samples. For a more detailed analysis of our method's robustness to different noise distributions, please see Appendix F.

**The Core Role of Fidelity Safeguards** We validate our fidelity safeguards by comparing the full OSCAR model against three ablated variants: one without

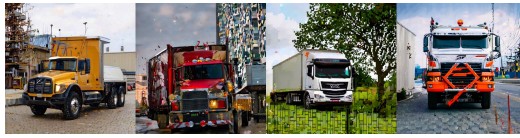

(a) w/o Redundancy-Aware Reweighting      (b) w/o Orthogonal Projection

Figure 6: Ablation study of two fidelity safeguards.

Table 2: Comparison with baselines and step-wise ablation of our fidelity safeguards, at fixed guidance $CFG=5.0$. Our full model outperforms both the foundational baseline and prior methods. The ablation study shows that each component contributes to the final performance.

| Method | CLIP Score↑ | Vendi (Pixel) ↑ | Vendi (Inception) ↑ | FID↓ | BRISQUE↓ |
|---|---|---|---|---|---|
| FM-SD3.5 | $28.24 \pm 0.18$ | $2.45 \pm 0.13$ | $5.37 \pm 0.27$ | $164.4 \pm 1.8$ | $23.4 \pm 1.4$ |
| DPP | $28.84 \pm 0.22$ | $2.55 \pm 0.23$ | $5.49 \pm 0.16$ | $171.2 \pm 1.9$ | $26.1 \pm 3.0$ |
| CADS | $27.32 \pm 0.21$ | $2.54 \pm 0.28$ | $5.58 \pm 0.19$ | $169.1 \pm 2.5$ | $24.0 \pm 1.9$ |
| PG | $28.27 \pm 0.27$ | $2.54 \pm 0.13$ | $5.43 \pm 0.18$ | $163.7 \pm 1.3$ | $24.2 \pm 3.3$ |
| w/o OP & RR | $26.70 \pm 0.23$ | $2.82 \pm 0.20$ | $4.86 \pm 0.24$ | $185.9 \pm 1.8$ | $50.1 \pm 2.8$ |
| w/o RR | $27.26 \pm 0.39$ | $2.77 \pm 0.16$ | $5.23 \pm 0.28$ | $174.4 \pm 3.0$ | $35.0 \pm 1.5$ |
| w/o OP | $26.59 \pm 0.25$ | $2.75 \pm 0.19$ | $5.06 \pm 0.16$ | $177.8 \pm 3.2$ | $38.8 \pm 1.5$ |
| **Oscar** | $\mathbf{28.26 \pm 0.22}$ | $\mathbf{2.86 \pm 0.05}$ | $\mathbf{5.63 \pm 0.22}$ | $\mathbf{163.3 \pm 1.6}$ | $\mathbf{21.2 \pm 1.5}$ |

both Orthogonal Projection and Redundancy-Aware Reweighting (w/o OP & RR), one without OP (w/o OP), and one without RR (w/o RR). The quantitative results in Table 2 show that removing these components consistently harms all metrics, with variants that drop OP suffering the most severe degradation. The visual results in Figure 6 further confirm this degradation, illustrating that these ablated variants are prone to chaotic textures and implausible artifacts, whereas our method maintains both diversity and realism. Additional qualitative comparisons are provided in Appendix G.

**Sensitivity to Key Hyperparameters**     This paragraph will explore the role of the most important parameters in our method on the final quality and diversity of generated samples.

**Orthogonality $\lambda$**    The influence of the orthogonality strictness $\lambda$ is detailed in Figure 7(a). This parameter has a critical impact on generation quality but a minor effect on diversity. As the data shows, relaxing the constraint by decreasing $\lambda$ leads to a noticeable degradation in fidelity, confirming that a strict orthogonal projection ($\lambda \approx 0.9$) is essential for preserving image quality.

**Leverage exponent $\alpha$**    The effect of the leverage exponent $\alpha$ is shown in Figure 7(b). The results reveal that an intermediate value (e.g., $\alpha = 0.5$) is better, as both fully adaptive ($\alpha = 1.0$) and uniform ($\alpha = 0.0$) weighting are found to degrade sample quality and reduce diversity.

**Noise gate $t_{\text{gate}}$**    Figure 7(c) illustrates the impact of the noise injection window by comparing extreme scenarios. While a conservative, late-stage injection is effective, extending the window to the entire generation process degrades sample quality as ex-

Figure 7: Ablation study on key parameters. Each row compares quantitative results with a visual example for a specific parameter.

(a) Influence of $\lambda$

| $\lambda$ | Brisque ↓ | $\text{VS}_p$ ↑ |
|---|---|---|
| 0.9 | 21.56 | 2.61 |
| 0.6 | 21.69 | 2.30 |
| 0.3 | 23.64 | 2.34 |

$\lambda = 0.3$     $\lambda = 0.9$

(b) Influence of $\alpha$

| $\alpha$ | Brisque ↓ | $\text{VS}_p$ ↑ |
|---|---|---|
| 1.0 | 40.39 | 2.50 |
| 0.5 | 20.63 | 2.66 |
| 0.1 | 44.28 | 2.49 |

$\alpha = 0.1$     $\alpha = 1.0$

(c) Influence of $t_{\text{gate}}$

| $t_{\text{gate}}$ | Brisque ↓ | $\text{VS}_p$ ↑ |
|---|---|---|
| 0.15 | 23.13 | 2.60 |
| 0.70 | 33.67 | 2.48 |
| 1.00 | 53.88 | 2.21 |

$t_{\text{gate}}[0.02, 0.70]$     $t_{\text{gate}}[0.02, 1.00]$

pected. However, these extremes belie the method's true stability, as our approach is in fact highly robust across a wide range of reasonable settings for both the noise gate and noise scale, as detailed in Appendix F.4.1.

## 6 CONCLUSION AND DISCUSSION

In this work, we addressed the critical challenge of the fidelity-diversity trade-off in flow-matching models by introducing a training-free, geometry-consistent control framework. Our method combines a deterministic guidance signal with a controllable stochastic noise to enhance semantic diversity. The key innovation is that both of these components are projected to be orthogonal to the model's base velocity field, which decouples the diversity-seeking signal from the quality-generating flow. Our experiments confirmed that this approach yields a superior precision-recall trade-off and enhances a suite of diversity metrics. While effective, extending our geometry-consistent control principles to other guidance tasks and modalities remains a promising avenue for future work.

## ETHICS STATEMENT

This work adheres to the ICLR Code of Ethics and principles of research integrity. The data used in this study are drawn from public sources and contain no private or sensitive information. A fairness-aware assessment of the methodology was conducted, and no significant discriminatory effects were identified. Potential risks related to the proposed applications have been carefully considered. The funding body played no role in the design or outcomes of this research, and the authors affirm the originality of the work submitted.

## REPRODUCIBILITY

To support reproducible research, we provide a detailed algorithm description in our main text (Algo 1), as well as comprehensive implementation and hardware specifications in Appendix C. The complete code will be made publicly available upon acceptance of the paper.

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

# Content of Appendix

## A  USE OF LLMs

The use of LLMs in this study was strictly limited to their function as writing and editing assistants. Their role involved improving the manuscript's grammar, spelling, and stylistic consistency. LLMs did not contribute to any core scientific aspects of the work, such as research ideation, experimental design, data analysis, or interpretation of results. All substantive content and intellectual contributions are solely the work of the authors.

## B  THEORETICAL PROPERTIES: QUALITY-PRESERVING DIVERSITY VIA ORTHOGONAL STOCHASTIC CONTROL

**Notation & Setup.**  Let $\hat{\psi} : \mathcal{X} \times [0,1] \to U$ be an endpoint predictor and $\phi : U \to \mathbb{R}^D$ a semantic feature encoder. For $m$ concurrent trajectories, define the endpoint features

$$z_i(t) \ = \ \phi\big(\hat{\psi}(x_i(t), t)\big), \qquad i = 1, \dots, m,$$

stack rows $Z(t) = [z_1(t); \dots; z_m(t)] \in \mathbb{R}^{m \times D}$, and let the Gram matrix be $K(t) = Z(t) Z(t)^\top \in \mathbb{R}^{m \times m}$. We use the trace–stabilized log–det set energy

$$\mathcal{E}_s(Z) \ = \ -\tfrac{1}{2} \log \det \Big( I + \tau\, K + \varepsilon_{\mathrm{tr}} I \Big), \qquad \varepsilon_{\mathrm{tr}} = \varepsilon\, \frac{\mathrm{tr}(K)}{m}, \quad \tau, \varepsilon > 0. \tag{6}$$

Let $v_\theta(x, t \,|\, c)$ denote the base drift with optional conditioning $c$ and define the regularized unit base direction

$$\hat{q}(x, t) \ = \ \frac{v_\theta(x, t \,|\, c)}{\|v_\theta(x, t \,|\, c)\| + \delta}, \qquad \delta > 0,$$

together with the quality–orthogonal projector

$$\Pi_\perp \ = \ I - \lambda\, \hat{q}\, \hat{q}^\top, \qquad \lambda \in [0, 1].$$

Our feature–space controlled SDE is

$$dZ_t \ = \ -\Pi_\perp \nabla_Z \mathcal{E}_s(Z_t)\, dt \ + \ \sqrt{\beta(t)}\, \Pi_\perp\, dW_t, \tag{7}$$

where $\beta : [0,1] \to \mathbb{R}_+$ is a bounded, nonincreasing noise schedule and $W_t$ is standard Brownian motion in $\mathbb{R}^{m \times D}$.

*Assumption* 1 (Local smoothness and bounded curvature on the orthogonal subspace).  (i) $\phi$ and $\hat{\psi}$ are $C^2$ with bounded Jacobians/Hessians on the sampling region. (ii) $\mathcal{E}_s$ is $C^2$ on $\{Z : I + \tau Z Z^\top + \varepsilon_{\mathrm{tr}} I \succ 0\}$. (iii) There exists $L_\perp > 0$ such that $\mathrm{tr}\big(\Pi_\perp^\top \nabla^2 \mathcal{E}_s(Z)\, \Pi_\perp\big) \leq L_\perp$ for all $Z$ encountered during sampling.

*Lemma* 1 (Pullback identity).  Let $\Phi(x, t) = \phi(\hat{\psi}(x, t))$ and $Z = [\Phi(x^{(1)}, t); \dots; \Phi(x^{(m)}, t)]$. Then for each $i$,

$$\frac{\partial}{\partial x^{(i)}} \mathcal{E}_s(Z) = \Big( J_x \hat{\psi}(x^{(i)}, t) \Big)^\top \Big( J_u \phi(u) \Big)^\top \Big[ \nabla_Z \mathcal{E}_s(Z) \Big]_i \Big|_{u = \hat{\psi}(x^{(i)}, t)}.$$

*Proof of Lemma 1.*  Let $f(Z) := \mathcal{E}_s(Z)$ and define the row-stacking map $G(x^{(1)}, \dots, x^{(m)}; t) = Z = [z_1; \dots; z_m]$ with $z_i = \Phi(x^{(i)}, t) = \phi(\hat{\psi}(x^{(i)}, t))$. Fix $i$ and a direction $v \in \mathbb{R}^{\dim(x)}$. By the chain rule for directional derivatives (Frobenius inner product), we have

$$\frac{d}{d\epsilon}\, f\big( G(x^{(1)}, \dots, x^{(i)} + \epsilon v, \dots, x^{(m)}; t) \big) \Big|_{\epsilon=0} = \big\langle \nabla_Z f(Z),\, \dot{Z} \big\rangle_F,$$

where $\dot{Z}$ is the first-order variation of $Z$ induced by the perturbation $x^{(i)} \mapsto x^{(i)} + \epsilon v$. Only the $i$-th row varies, hence

$$\dot{z}_i = \frac{d}{d\epsilon}\, \Phi(x^{(i)} + \epsilon v, t) \Big|_{\epsilon=0} = J_u \phi(u_i)\, J_x \hat{\psi}(x^{(i)}, t)\, v, \qquad u_i = \hat{\psi}(x^{(i)}, t).$$

Using $\langle A, B \rangle_F = \sum_j \langle A_j, B_j \rangle$ (rowwise),

$$\frac{d}{d\epsilon} f(\cdot) \Big|_{\epsilon=0} = \big\langle [\nabla_Z f(Z)]_i,\, \dot{z}_i \big\rangle = v^\top \Big( J_x \hat{\psi}(x^{(i)}, t) \Big)^\top \Big( J_u \phi(u_i) \Big)^\top [\nabla_Z \mathcal{E}_s(Z)]_i.$$

Since this holds for every $v$, we obtain

$$\frac{\partial}{\partial x^{(i)}} \mathcal{E}_s(Z) = \left( J_x \hat{\psi}(x^{(i)}, t) \right)^\top \left( J_u \phi(u) \right)^\top \left[ \nabla_Z \mathcal{E}_s(Z) \right]_i \Big|_{u = \hat{\psi}(x^{(i)}, t)},$$

which is the desired identity. The $C^2$ conditions in Assumption 1 ensure differentiability and justify exchanging differentiation with the Frobenius inner product. □

*Remark* 1 (Scope of assumptions). We do not assume global boundedness for deep networks. Our analysis is restricted to the operational domain $\mathcal{D}$ traced by the sampler (finite horizon, bounded step sizes, regularization), where neural networks are empirically locally smooth. Such local assumptions are standard in analyses of learned dynamics and control. In practice, weight decay/normalization and gradient clipping further bound the effective Jacobians on $\mathcal{D}$.

*Theorem* 1 (Expected energy descent & marginal quality preservation). Under Assumption 1, along the SDE equation 1 we have

$$\frac{d}{dt} \mathbb{E}\, \mathcal{E}_s(Z_t) \;\le\; -\, \mathbb{E} \left\| \Pi_\perp \nabla_Z \mathcal{E}_s(Z_t) \right\|_F^2 \;+\; \beta(t)\, L_\perp. \tag{8}$$

Consequently, since $\beta(t) \downarrow 0$ as $t \uparrow 1$, there exists $t^\star \in (0, 1)$ such that for all $t \ge t^\star$, $\frac{d}{dt} \mathbb{E}\, \mathcal{E}_s(Z_t) < 0$ (late-stage monotonicity). Moreover, because both the control drift and diffusion act in $\mathrm{range}(\Pi_\perp)$, the alignment coordinate $y = \langle x, \hat{q} \rangle$ has the same marginal evolution as the base FM ODE to first order in the step size: no diffusion and no control drift along $y$.

*Proof.* Apply Itô's lemma to the twice-differentiable energy $\mathcal{E}_s(Z_t)$ under equation **??**. Writing $G(Z) = \nabla_Z \mathcal{E}_s(Z)$ and $H(Z) = \nabla_Z^2 \mathcal{E}_s(Z)$, with drift $b(Z_t) = -\Pi_\perp G(Z_t)$ and diffusion matrix $\Sigma_t = \sqrt{\beta(t)}\, \Pi_\perp$, we obtain

$$\frac{d}{dt} \mathbb{E}\, \mathcal{E}_s(Z_t) = \mathbb{E} \left[ \langle G(Z_t), b(Z_t) \rangle_F \right] + \tfrac{1}{2} \mathbb{E} \left[ \langle H(Z_t), \Sigma_t \Sigma_t^\top \rangle_F \right].$$

The first term equals $-\mathbb{E} \left\| \Pi_\perp G(Z_t) \right\|_F^2$. For the second term, using $\Sigma_t \Sigma_t^\top = \beta(t)\, \Pi_\perp$ and Assumption 1(iii),

$$\tfrac{1}{2} \mathbb{E} \left[ \langle H(Z_t), \Sigma_t \Sigma_t^\top \rangle_F \right] = \tfrac{\beta(t)}{2} \mathbb{E} \left[ \mathrm{tr}(\Pi_\perp^\top H(Z_t) \Pi_\perp) \right] \;\le\; \beta(t)\, L_\perp,$$

absorbing the factor $\tfrac{1}{2}$ into $L_\perp$ if desired. Combining the two gives equation 8. Since $\beta(t)$ is bounded and $\beta(t) \downarrow 0$ as $t \uparrow 1$, there exists $t^\star$ such that the negative first term dominates for all $t \ge t^\star$, hence $\frac{d}{dt} \mathbb{E}\, \mathcal{E}_s(Z_t) < 0$.

For the alignment coordinate $y = \langle x, \hat{q} \rangle$, note that both the control drift $-\Pi_\perp G$ and the diffusion $\sqrt{\beta(t)}\, \Pi_\perp dW_t$ lie in $\mathrm{range}(\Pi_\perp)$, which is orthogonal to $\hat{q}$. Therefore, to first order in the step size, there is neither additional drift nor diffusion along $y$ beyond that of the base FM ODE, establishing marginal quality preservation. □

*Remark* 2 (Discretization and redundancy-aware reweighting).

(i) **Heun-style discretization.** With step size $\Delta t$, the update reads

$$x_{k+1} = x_k + \Delta t \left[ \tfrac{1}{2}(v_k + v_{k+1}) + u_k \right], \qquad u_k = \Pi_\perp g(x_k, t_k) + \sqrt{\beta(t_k)\, \Delta t}\, \Pi_\perp \xi_k, \;\; \xi_k \sim \mathcal{N}(0, I). \tag{9}$$

This inherits the descent certificate equation 8 up to an $O(\Delta t^2)$ truncation error. In practice, we use $\Delta t \le 0.05$, making the discretization error negligible relative to the stochastic variability.

(ii) **Redundancy-aware reweighting.** Let $s_i = \left[ (K + \varepsilon I)^{-1} \right]_{ii}$ and set $w_i \propto s_i^\alpha$ ($\alpha > 0$). With $W = \mathrm{diag}(w_1, \ldots, w_m)$, replace $K$ by $\tilde{K} = W^{1/2} K W^{1/2}$ in the energy equation 6. This preconditioning boosts underrepresented samples and attenuates redundant ones, and it preserves the descent inequality equation 8 with a possibly different constant $L_\perp$, thereby stabilizing the set gradient without an explicit trust region.

*Link to Theorem 2.* Replacing $K$ by $\tilde{K}$ keeps the identity $-\mathcal{E}_s(Z) = \tfrac{1}{2} \log \det \left( I + \tau \tilde{K} + \varepsilon_{\mathrm{tr}} I \right)$, so Theorem 2 applies verbatim to the *weighted* set as a weighted volume/diversity measure.

*Assumption* 2 (Feature regularity for geometric interpretation). (*Approx. centering*) $\sum_{i=1}^m z_i \approx 0$; (*Norm stabilization*) $\|z_i\|^2 \in [1 - \rho,\, 1 + \rho]$ for a small $\rho$.

*Remark* 3 (Practical scope). CLIP-style encoders output $\ell_2$-normalized features and we further apply mini-batch centering. The objective equation 10 does *not* require strict centering; Assumption 2 is only used to interpret log-det as an isotropic "volume" and to link it to diversity measures.

*Theorem* 2 (Energy $\downarrow \Longleftrightarrow$ Volume $\uparrow \Longrightarrow$ Diversity $\uparrow$). Let $K = ZZ^\top$ and define the set volume

$$\mathcal{V}(Z) \;=\; \det\big(I + \tau K + \varepsilon_{\mathrm{tr}}I\big)^{1/2}.$$

Then

$$-\mathcal{E}_s(Z) \;=\; \tfrac{1}{2}\log\det\big(I + \tau K + \varepsilon_{\mathrm{tr}}I\big) \;=\; \log\mathcal{V}(Z). \tag{10}$$

Consequently, by Theorem 1, $\mathbb{E}\log\mathcal{V}(Z_t)$ is (late-stage) increasing. Moreover, under Assumption 2:

(i) (*Angles / correlations*) With approximately unit-norm rows, increasing $\mathcal{V}(Z)$ reduces average pairwise correlations and increases pairwise angles among $\{z_i\}$.

(ii) (*Spectrum uniformity*) Since $\mathrm{tr}(K) = \sum_i \|z_i\|^2 \approx m$ is (approximately) fixed, the concavity of $\log\det(\cdot)$ implies that a larger $\log\det(I + \tau K + \varepsilon_{\mathrm{tr}}I)$ corresponds to a more uniform spectrum of $K$ (equivalently, more balanced singular values of $Z$), indicating higher coverage/diversity.

*Proof.* The identity equation 10 is immediate from the definition of $\mathcal{E}_s$. Late-stage monotonicity of $\mathbb{E}\log\mathcal{V}(Z_t)$ follows from Theorem 1.

For (ii) (*spectrum uniformity*), write $A = \alpha I + \tau K$ with $\alpha = 1 + \varepsilon_{\mathrm{tr}} > 0$. Let $(\lambda_j)$ be the eigenvalues of $K$. Then

$$\log\det A = \sum_{j=1}^m \log(\alpha + \tau\lambda_j).$$

As a symmetric concave function of $(\lambda_j)$, this sum is *Schur-concave*; under the (approximate) trace constraint $\sum_j \lambda_j = \mathrm{tr}(K) \approx m$, it is maximized when $(\lambda_j)$ is more balanced (Jensen/Karamata). Hence larger $\log\det A$ implies a more uniform spectrum of $K$ (and thus more balanced singular values of $Z$).

For (i) (*correlations/angles*), under Assumption 2 we have $\mathrm{diag}(K) \approx \mathbf{1}$, so the diagonals are (approximately) fixed. Using

$$\|K\|_F^2 = \sum_{j=1}^m \lambda_j^2 = \sum_{i=1}^m K_{ii}^2 + 2\sum_{i<j} K_{ij}^2 \;\approx\; m + 2\sum_{i<j} K_{ij}^2,$$

we see that, for fixed diagonals, the average squared off-diagonal magnitude is monotone in $\sum_j \lambda_j^2$. Among spectra with fixed trace, $\sum_j \lambda_j^2$ is minimized when the spectrum is most uniform; thus the same majorization argument used in (ii) implies $\sum_{i<j} K_{ij}^2$ decreases as $\log\det A$ increases. Since, with (unit-norm) rows, $K_{ij} = \langle z_i, z_j \rangle$ are cosines, smaller off-diagonals mean reduced average correlations and hence larger pairwise angles. This establishes (i). $\square$

*Theorem* 3 (Noise robustness via budget and step size). Assume (i) $\hat{q}$ is $L_q$-Lipschitz on the operational domain; (ii) along the trajectory $\|\nabla_Z \mathcal{E}_s(Z)\|_F \leq G$; and (iii) Assumption 1 holds so that equation 8 is valid with curvature constant $L_\perp$. Let $y_t = \langle z_t, \hat{q}(z_t, t) \rangle$ and $y_t^{\mathrm{base}}$ be the alignment coordinate under the base flow (no control, no orthogonal diffusion). For an integrator with step $\Delta t$ and any bounded, nonincreasing schedule $\beta : [0, 1] \to \mathbb{R}_+$,

$$\left| \mathbb{E}\, y_1 - \mathbb{E}\, y_1^{\mathrm{base}} \right| \;\leq\; C_1\, L_q\, G\, \Delta t \;+\; C_2\, L_\perp \int_0^1 \beta(t)\, dt, \tag{11}$$

for universal constants $C_1, C_2 = O(1)$. Hence the deviation is $O(\Delta t) + O(B)$ with $B := \int_0^1 \beta(t)\, dt$.

*Proof.* Work with the feature-space SDE and its discretization. Because both the control drift $-\Pi_\perp \nabla_Z \mathcal{E}_s$ and the diffusion $\sqrt{\beta(t)}\, \Pi_\perp dW_t$ lie in $\mathrm{range}(\Pi_\perp)$, the alignment coordinate $y = \langle z, \hat{q} \rangle$ receives no first-order contribution from the control when $\hat{q}$ is locally frozen; the residual change is controlled by the variation of $\hat{q}$ and by the stochastic term. The Lipschitz property of $\hat{q}$ and the bound

$\|\nabla_Z \mathcal{E}_s\|_F \le G$ yield a discretization remainder bounded by $C_1 L_q G \Delta t$. For the stochastic contribution, apply Itô's isometry together with the descent certificate equation 8: the quadratic variation seen through the curvature bound in Assumption 1(iii) accumulates as $\frac{1}{2} \mathbb{E} \langle \nabla_Z^2 \mathcal{E}_s, \beta(t) \Pi_\perp \rangle_F \le \beta(t) L_\perp$, which integrates to $C_2 L_\perp \int_0^1 \beta(t)\, dt$. Combining the two parts gives equation 11. $\qquad\square$

*Proposition* 1 (Schedule generality: budget controls the bound). Let $\beta_1, \beta_2$ be bounded, nonincreasing schedules with equal budget $B = \int_0^1 \beta_j(t)\, dt$. Under the assumptions of Theorem 3, the bound equation 11 is the same for $\beta_1$ and $\beta_2$ (up to constants), and for any $a > 0$, replacing $\beta$ by $\beta_a(t) = a\, \beta(t)$ scales the bound linearly with $a$ via $B_a = a\, B$.

*Proof.* Immediate from equation 11, which depends on $\beta$ only through the integral $\int_0^1 \beta(t)\, dt$. $\qquad\square$

*Remark* 4 (Schedules and matching a target budget). A convenient closed form is $\beta(t) = \dfrac{1 - t}{1 - t + \varepsilon_\beta}$ with $\varepsilon_\beta \in (0, 1)$, which decreases from $\frac{1}{1 + \varepsilon_\beta} \approx 1$ at $t = 0$ to $0$ at $t = 1$. Common monotone decay choices include:

$$\beta_0^{\cos}(t) = \cos^2\left(\frac{\pi t}{2}\right), \quad \beta_0^{\text{poly}}(t) = (1 - t)^p \ (p \ge 1), \quad \beta_0^{\exp}(t) = \frac{e^{-\kappa t} - e^{-\kappa}}{1 - e^{-\kappa}} \ (\kappa > 0).$$

To match a desired noise budget $B$, set $\tilde{\beta}(t) = a\, \beta_0(t)$ with $a = B \big/ \int_0^1 \beta_0(t)\, dt$. We provide empirical comparisons of these schedules in Appendix F.4.3.

*Theorem* 4 (Girsanov Representation of OSCAR Path Measure). Let $\mathbb{P}$ be the path measure induced by the reference process (with orthogonal noise but no guidance) and $\mathbb{Q}$ be the path measure induced by the full OSCAR process on the time interval $[0, 1]$. Assuming the guidance signal $g(x, t)$ and the noise schedule satisfy Novikov's condition, the Radon-Nikodym derivative (likelihood ratio) of the generated trajectories is given by:

$$\frac{d\mathbb{Q}}{d\mathbb{P}}(X_{[0,1]}) = \exp\left(\int_0^1 \frac{1}{\sqrt{\beta(t)}} \langle g(X_t, t), dW_t \rangle - \frac{1}{2} \int_0^1 \frac{\|g(X_t, t)\|^2}{\beta(t)}\, dt\right)$$

where the inner products are defined in the subspace defined by $\Pi_\perp$.

Furthermore, the KL divergence between the OSCAR trajectory distribution and the reference distribution is exactly the energy cost of the guidance:

$$\mathcal{D}_{KL}(\mathbb{Q} \| \mathbb{P}) = \frac{1}{2} \mathbb{E}_{\mathbb{Q}} \left[ \int_0^1 \frac{\|g(X_t, t)\|^2}{\beta(t)}\, dt \right]$$

*Theorem* 5 (Process-level coupling bound). Let the baseline process follow the probability-flow ODE $dX_t = v_\theta(X_t, t)\, dt$ and the controlled process follow

$$d\tilde{X}_t = \left( v_\theta(\tilde{X}_t, t) - \Pi_\perp(\tilde{X}_t, t)\, g(\tilde{X}_t, t) \right) dt + \sqrt{\beta(t)}\, \Pi_\perp(\tilde{X}_t, t)\, dW_t,$$

driven by the *same* Brownian motion $W_t$ (synchronous coupling). Assume: (i) $v_\theta(\cdot, t)$ is $L_v$-Lipschitz for all $t \in [0, 1]$; (ii) $\Pi_\perp(\cdot, t)$ is $L_\Pi$-Lipschitz and $\|\Pi_\perp\| \le 1$; (iii) $\beta$ is bounded and nonincreasing; and (iv) the drift budget is finite: $M_g := \int_0^1 \|\Pi_\perp(\cdot, t)\, g(\cdot, t)\|_\infty\, dt < \infty$. Let $D_t := \tilde{X}_t - X_t$. Then there exists $C = C(L_v, L_\Pi)$ such that

$$\frac{d}{dt} \mathbb{E}\|D_t\|^2 \le C\, \mathbb{E}\|D_t\|^2 + C\, \|\Pi_\perp g(\cdot, t)\|_\infty^2 + C\, \beta(t), \tag{12}$$

and hence, by Grönwall,

$$\mathbb{E}\|D_1\|^2 \le C\left( \left( \int_0^1 \|\Pi_\perp g(\cdot, t)\|_\infty\, dt \right)^2 + \int_0^1 \beta(t)\, dt \right). \tag{13}$$

*Proof.* By Itô for $D_t = \tilde{X}_t - X_t$,

$$dD_t = \underbrace{\left(v_\theta(\tilde{X}_t, t) - v_\theta(X_t, t)\right)}_{\text{Lipschitz in } D_t} dt - \Pi_\perp(\tilde{X}_t, t)g(\tilde{X}_t, t)\, dt + \sqrt{\beta(t)}\, \Pi_\perp(\tilde{X}_t, t)\, dW_t.$$

Applying Itô to $\|D_t\|^2$ and taking expectations,

$$\frac{d}{dt}\mathbb{E}\|D_t\|^2 = 2\,\mathbb{E}\left\langle D_t,\ v_\theta(\tilde{X}_t, t) - v_\theta(X_t, t)\right\rangle - 2\,\mathbb{E}\left\langle D_t,\ \Pi_\perp g(\tilde{X}_t, t)\right\rangle + \beta(t)\,\mathbb{E}\|\Pi_\perp(\tilde{X}_t, t)\|_F^2.$$

The first term is bounded by $2L_v\,\mathbb{E}\|D_t\|^2$. For the second, use Young's inequality: for any $\varepsilon > 0$, $-2\langle D, \Pi_\perp g\rangle \le \varepsilon\|D\|^2 + \varepsilon^{-1}\|\Pi_\perp g\|^2$. Choosing $\varepsilon$ to absorb into $2L_v$ yields

$$\frac{d}{dt}\mathbb{E}\|D_t\|^2 \ \le\ C\,\mathbb{E}\|D_t\|^2 \ +\ C\,\|\Pi_\perp g(\cdot, t)\|_\infty^2 \ +\ C\,\beta(t),$$

since $\|\Pi_\perp\|_F^2 \le C$ by (ii). Grönwall then gives equation 13. $\qquad\square$

## C  Implementation Details

### C.1  Finite-difference endpoint extrapolation.

Let $z_k$ denote the latent at step $k$ (after applying OSCAR at step $k-1$), and let $\Delta z_{k-1}^{\text{ctrl}}$ be the control displacement added at step $k-1$ with step size $\Delta t_{k-1}$. We estimate a local velocity by a finite difference

$$v_k \ \approx\ \frac{z_k - z_{k-1} - \Delta z_{k-1}^{\text{ctrl}}}{\Delta t_{k-1}},$$

and extrapolate a local endpoint as

$$z_{\text{ep}} \ =\ z_k + \alpha(t_k)\, v_k,$$

where $\alpha(t_k)$ is a scalar schedule. This extrapolation reuses already computed latents and does not require an extra evaluation of $v_\theta$, so the predictor NFE is unchanged. We initially experimented with a classical Heun-style predictor that evaluates $v_\theta$ twice per step, but found that, under matched compute, it did not yield consistent improvements while doubling the predictor NFE (Karras et al., 2022). For this reason, all reported results use the finite-difference extrapolation above, which captures a similar average velocity at no additional predictor cost.

### C.2  General Experimental Setup

All experiments are conducted on the frozen, pretrained Stable Diffusion v-3.5 Medium model (Esser et al., 2024), with all generations performed at a resolution of 512x512 pixels and using bfloat16 precision. All runs use the model's default flow-matching Euler scheduler (`FlowMatchEulerDiscreteScheduler`) with 30 sampling steps. A consistent negative prompt, "low quality, blurry", is used across all experiments. The main prompts used for generation are derived from the captions of the COCO dataset. To ensure robustness, all reported metrics are aggregated over at least four distinct random seeds. For each unique setting (i.e., a specific prompt and CFG scale), we generate a set of 64 images for general class-conditional tasks, and a set of 20 images for the DIM/CIM evaluations. All experiments were performed on a system with two NVIDIA A40 GPUs.

### C.3  Framework and Baseline Implementation Details

Our flow-matching backbone, which serves as the foundation for our method and all baselines, is adapted from the official implementation at `https://github.com/facebookresearch/flow_matching`. For **Particle Guidance** (Corso et al., 2023), which was originally designed for diffusion models, we adapted the official author implementation, publicly available at `https://github.com/gcorso/particle-guidance`, to the flow-matching framework. For **CADS** (Sadat et al., 2023) and **Diverse Flow** (Morshed & Boddeti, 2025), official code was not provided by the authors. Therefore, we carefully re-implemented their methodologies, strictly adhering to the descriptions and formulations presented in their respective papers. For all baselines, we performed a thorough hyperparameter search for their key parameters to ensure each method was evaluated at its strongest possible performance, guaranteeing a fair and rigorous comparison.

Table 3: Hyperparameters used by our sampler, with default values and brief descriptions.

| Name | Symbol | Value | Meaning |
|------|--------|-------|---------|
| gamma0 | $\gamma_0$ | 0.12 | Global strength for the diversity/volume term. |
| gamma-max-ratio | $\gamma_{\mathrm{max}}$ | 0.3 | Trust-region ratio: caps diversity displacement by a fraction of the base flow displacement norm. |
| partial-ortho | $p_\perp$ | 0.95 | Proportion of projection orthogonal to the base velocity. |
| noise-partial-ortho | $p_\perp^{\mathrm{noise}}$ | 0.95 | Proportion of orthogonalization for the noise with respect to the base velocity. |
| t-gate | $t_{\mathrm{gate}}$ | 0.4 | Time gate $[0.05, t_{gate}]$ where the diversity term & noise is active. |
| sched-shape | $noise(t)$ | cos2 | Time schedule shape for $\gamma$ (cos2 or t1mt).[*] |
| tau | $\tau$ | 1.0 | Scale for the volume/log-det related term. |
| eps-logdet | $\varepsilon$ | $1 \times 10^{-3}$ | Numerical stabilizer $\varepsilon$ for log-determinant/volume computation. |
| eta-sde | $\eta$ | 1.0 | Global scale for SDE/Brownian noise. |
| vnorm-threshold | $\delta_v$ | $1 \times 10^{-4}$ | Skip velocity-based projections when the base velocity norm is below this threshold. |

[*] The cosine-squared (cos2) schedule yields a smooth, bell-shaped profile that ramps up from zero, peaks at the midpoint, and then ramps down. The parabolic (t1mt) schedule exhibits a similar rise–fall pattern shaped like an inverted parabola.

### C.4 Evaluation Setup

Our evaluation metrics are computed using standard pretrained models. Specifically, we use OpenAI's ViT-B/32 model for calculating CLIP Scores (Radford et al., 2021), and a standard InceptionV3 model with ImageNet weights for all Inception-based metrics (Szegedy et al., 2015). For perceptual quality and human-preference alignment, we additionally report ImageReward scores using the public ImageReward model (Xu et al., 2023), and CLIP-IQA, a CLIP-based no-reference image quality metric (Wang et al., 2023). All these models are kept fixed and are never finetuned on our generated data. For distributional metrics that require a reference set of real images, such as FID and KID, we construct a concept-specific reference dataset to ensure a fair comparison. Since our prompts are based on concepts from the COCO dataset (Lin et al., 2014), our reference set for a given concept is composed exclusively of all images corresponding to that concept's class from the COCO training set. For example, when evaluating generated "truck" images, the real reference dataset consists solely of the "truck" images from the COCO training data. This domain-aligned setup provides a more accurate measure of distributional fidelity.

### C.5 Hyperparameters Setup

Table 3 details the key hyperparameters for our method, which govern the core components of our diversity guidance and orthogonal noise injection.

## D Additional Main Experiments Results

### D.1 Additional Precision-Recall Distribution Curves

To demonstrate that the superior fidelity-coverage trade-off of our method is not limited to a single class, this section provides additional PRD curves for two more concepts from the COCO dataset: "bus" and "bicycle". As shown in Figure 8, the results are consistent with the findings for the "truck" concept presented in the main paper. Across all concepts and guidance scales, our method's PRD curve consistently lies to the top-right of the baselines, achieving a higher recall for any given level of precision and thus a superior AUC.

Figure 8: Additional PRD curves for the (a) bus, (b) bicycle, (c) apple, (d) pizza and (e) suitcase concepts. The results confirm that our method consistently achieves a better fidelity–coverage trade-off compared to baselines across various semantic classes.

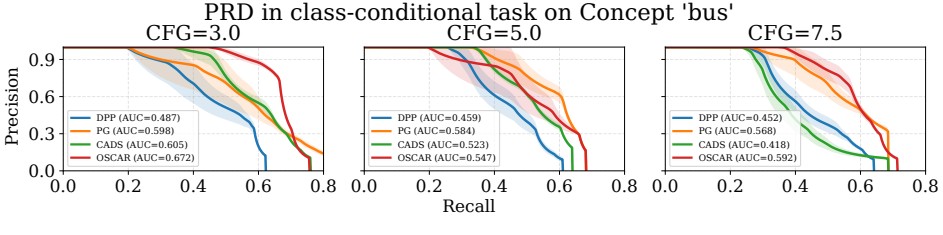

(a) PRD curves for the bus concept

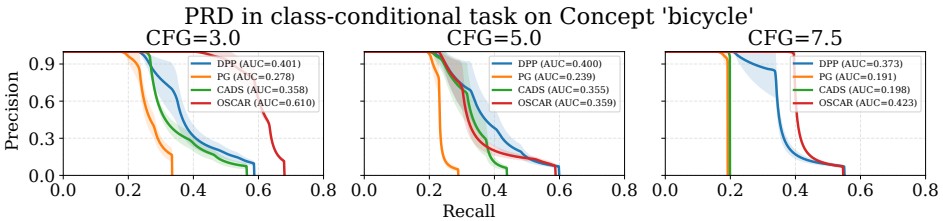

(b) PRD curves for the bicycle concept

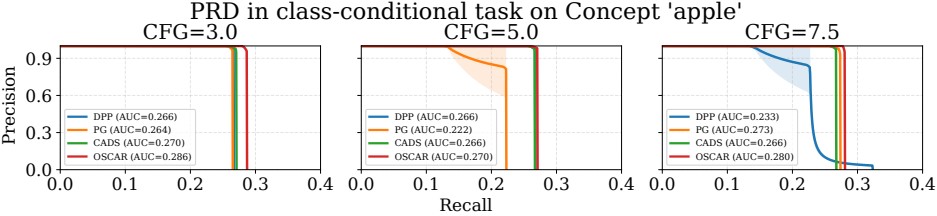

(c) PRD curves for the apple concept

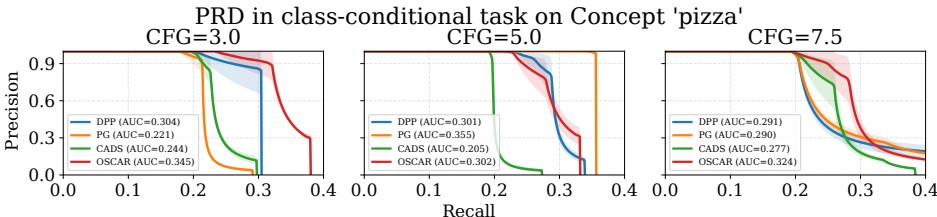

(d) PRD curves for the pizza concept

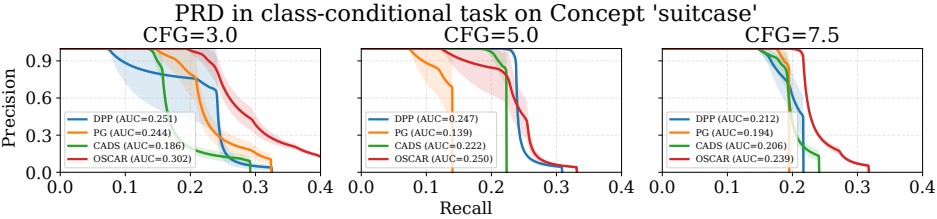

(e) PRD curves for the suitcase concept

## D.2 EXTENDED MODE COVERAGE AND ENTROPY ANALYSIS

To further demonstrate our method's ability to discover and evenly represent fine-grained, intra-class modes, we extend the mode coverage and normalized entropy analysis to the "bus" and "bicycle"

concepts. The results, shown in Figure 9 and Figure 10, are fully consistent with the findings for the "truck" concept presented in the main paper.

For both additional concepts, our method consistently achieves a higher final cluster coverage plateau, indicating that it successfully identifies a broader set of unique sub-types compared to the baselines. Furthermore, the normalized entropy scores for our method are substantially and consistently higher. This provides strong evidence that our generated samples are more uniformly distributed across the discovered modes, leading to a less redundant and more semantically rich output for the end-user.

Figure 9: Mode Coverage and Normalized Entropy for the 'bus' concept. The results confirm our method's superior ability to both discover more intra-class modes and distribute samples more uniformly among them.

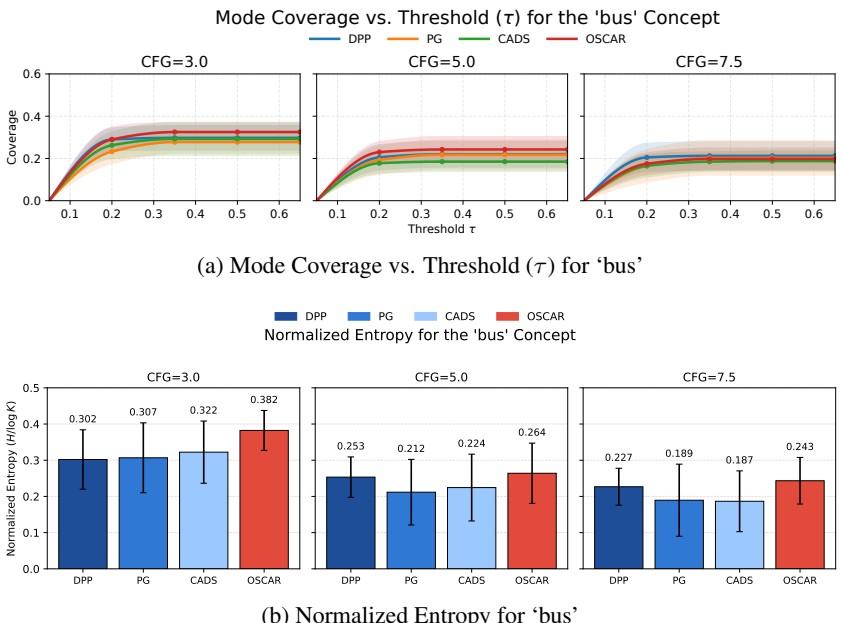

(a) Mode Coverage vs. Threshold ($\tau$) for 'bus'

(b) Normalized Entropy for 'bus'

### D.3 QUANTITATIVE RESULTS FOR ADDITIONAL CONCEPTS

To further validate the generalizability of our method's performance, this section provides a comprehensive extension of the main experimental results. For the sake of brevity and clarity, the quantitative and qualitative comparisons presented in the main body of the paper focused primarily on the "truck" concept. Here, we present the same set of comparisons for several additional concepts from the COCO dataset, with detailed quantitative results for the "bus" and "bicycle" concepts shown in Table 4 and Table 5. These extended results demonstrate that the conclusions drawn in the main paper are not specific to a single class, and confirm that our method's advantages in enhancing diversity while preserving quality hold across a wider range of semantic categories.

#### D.3.1 CLASSIFICATION PROMPTS EXAMPLE

To ensure our evaluation is robust and not biased by specific prompt phrasing, we use a set of synonymous prompts for each tested concept. The following listing shows an excerpt from our prompt file, illustrating the structure used for the "truck" and "bus" concepts discussed in the main paper.

Figure 10: Mode Coverage and Normalized Entropy for the 'bicycle' concept, reinforcing the consistent outperformance of our method.

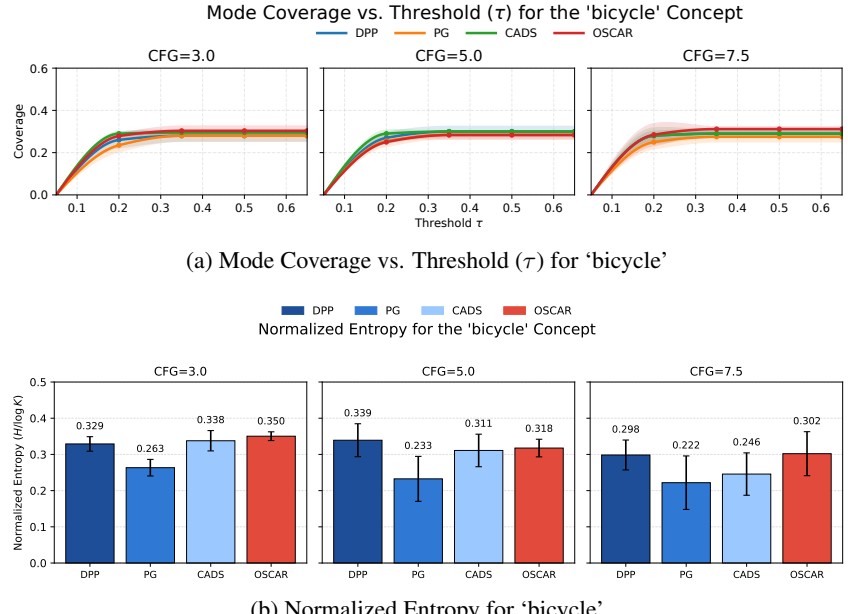

(a) Mode Coverage vs. Threshold ($\tau$) for 'bicycle'

(b) Normalized Entropy for 'bicycle'

---

**Example of the JSON structure used to store prompts for different concepts**

```
"truck": [
    "a truck",
    "a photo of a truck",
    "a photo of one truck"
],
"bus": [
    "a bus",
    "a photo of a bus",
    "a photo of one bus"
]
```

### D.3.2 PROMPT-LEVEL ANALYSIS WITHIN A SINGLE CONCEPT

We also notice that, even within a fixed semantic concept, different textual prompts can induce large variations in standard evaluation metrics. When aggregating over all prompts belonging to the same concept, this high intra-concept variance tends to blur the performance differences between methods and makes the results harder to interpret. To better disentangle these effects, in this part we therefore report metrics for two representative prompts under the "truck" concept, namely "a truck" and "a photo of a truck" (see Tab. 7 and Tab. 6). The former is intentionally short and under-specified, while the latter provides slightly richer semantic context. Comparing methods under these two prompts side by side allows us to more clearly separate the impact of our diversity-enhancing control from the prompt-induced variability, and we observe similar behaviors for other concepts.

In addition, to further demonstrate that our conclusions are not specific to a single concept, we report prompt-level results on three other concepts, "apple", "suitcase", and "pizza", using a single representative prompt for each concept (see Tab. 8, Tab. 9, and Tab. 10). On these concepts, our method achieves consistently larger improvements on the diversity metrics (Vendi Score Pixel and Vendi Score Inception) compared to all baselines, while preserving comparable image quality.

Table 4: Comprehensive quantitative comparison of our method against all baselines for the "bus" concept. The results demonstrate the performance of different methods across various guidance scales and evaluation metrics.

| Method | Guidance Scale (CFG) | | | Guidance Scale (CFG) | | |
|---|---|---|---|---|---|---|
| | 3.0 | 5.0 | 7.5 | 3.0 | 5.0 | 7.5 |
| | **Brisque** $\downarrow$ | | | $1 - \text{MS-SSIM}(\%)$ $\uparrow$ | | |
| FM-SD3.5 | $24.92 \pm 2.41$ | $27.93 \pm 2.18$ | $32.07 \pm 2.03$ | $91.05 \pm 0.58$ | $91.21 \pm 0.67$ | $90.85 \pm 0.99$ |
| PG | $38.02 \pm 6.00$ | $34.76 \pm 3.92$ | $36.18 \pm 2.45$ | $90.89 \pm 1.57$ | $90.96 \pm 2.07$ | $90.96 \pm 2.94$ |
| CADS | $23.49 \pm 2.17$ | $27.91 \pm 2.51$ | $34.93 \pm 3.37$ | $93.11 \pm 0.77$ | $92.80 \pm 1.13$ | $91.79 \pm 1.89$ |
| DPP | $23.00 \pm 3.12$ | $28.31 \pm 2.68$ | $34.86 \pm 2.52$ | $90.82 \pm 1.24$ | $90.96 \pm 1.93$ | $90.10 \pm 2.75$ |
| **Ours** | $22.71 \pm 5.03$ | $26.55 \pm 4.72$ | $33.55 \pm 4.78$ | $92.38 \pm 1.23$ | $91.91 \pm 1.77$ | $91.12 \pm 2.26$ |
| | **Vendi Score Pixel** $\uparrow$ | | | **Vendi Score Inception** $\uparrow$ | | |
| FM-SD3.5 | $3.49 \pm 0.33$ | $3.60 \pm 0.44$ | $3.62 \pm 0.46$ | $3.74 \pm 0.51$ | $3.28 \pm 0.44$ | $3.07 \pm 0.42$ |
| PG | $3.53 \pm 0.74$ | $3.68 \pm 1.06$ | $3.63 \pm 1.12$ | $4.06 \pm 1.08$ | $3.35 \pm 0.89$ | $3.21 \pm 0.81$ |
| CADS | $3.95 \pm 0.81$ | $3.93 \pm 1.13$ | $3.87 \pm 1.24$ | $3.91 \pm 1.11$ | $3.40 \pm 0.96$ | $3.27 \pm 0.85$ |
| DPP | $3.59 \pm 0.80$ | $3.58 \pm 1.09$ | $3.46 \pm 1.10$ | $3.78 \pm 1.40$ | $3.31 \pm 1.31$ | $3.13 \pm 1.15$ |
| **Ours** | $4.08 \pm 0.80$ | $3.98 \pm 1.02$ | $3.78 \pm 1.05$ | $4.03 \pm 1.12$ | $3.47 \pm 0.99$ | $3.32 \pm 0.92$ |
| | **CLIP-IQA** $\uparrow$ | | | **CLIP Score** $\uparrow$ | | |
| FM-SD3.5 | $6.08 \pm 0.32$ | $6.06 \pm 0.37$ | $6.14 \pm 0.37$ | $28.08 \pm 0.17$ | $27.96 \pm 0.18$ | $27.92 \pm 0.17$ |
| PG | $6.11 \pm 0.59$ | $6.20 \pm 0.49$ | $6.20 \pm 0.47$ | $28.05 \pm 0.38$ | $27.68 \pm 0.36$ | $27.72 \pm 0.30$ |
| CADS | $6.23 \pm 0.53$ | $6.21 \pm 0.49$ | $6.19 \pm 0.49$ | $27.81 \pm 0.42$ | $27.65 \pm 0.34$ | $27.76 \pm 0.25$ |
| DPP | $6.09 \pm 0.58$ | $6.09 \pm 0.54$ | $6.04 \pm 0.48$ | $28.28 \pm 0.40$ | $28.12 \pm 0.33$ | $28.17 \pm 0.37$ |
| **Ours** | $6.24 \pm 0.58$ | $6.26 \pm 0.52$ | $6.27 \pm 0.46$ | $27.94 \pm 0.37$ | $27.84 \pm 0.36$ | $27.91 \pm 0.37$ |
| | **FID** $\downarrow$ | | | **Image Reward** $\uparrow$ | | |
| FM-SD3.5 | $113.25 \pm 1.60$ | $111.22 \pm 1.28$ | $110.61 \pm 1.13$ | $0.26 \pm 0.31$ | $0.38 \pm 0.32$ | $0.42 \pm 0.26$ |
| PG | $116.18 \pm 2.93$ | $114.41 \pm 2.35$ | $113.71 \pm 2.07$ | $0.23 \pm 0.48$ | $0.39 \pm 0.31$ | $0.44 \pm 0.29$ |
| CADS | $115.37 \pm 2.53$ | $114.40 \pm 2.12$ | $114.04 \pm 1.75$ | $0.32 \pm 0.32$ | $0.43 \pm 0.24$ | $0.48 \pm 0.23$ |
| DPP | $114.65 \pm 3.87$ | $113.39 \pm 3.18$ | $112.66 \pm 2.61$ | $0.22 \pm 0.44$ | $0.31 \pm 0.35$ | $0.35 \pm 0.31$ |
| **Ours** | $114.33 \pm 3.50$ | $112.44 \pm 2.38$ | $111.98 \pm 2.50$ | $0.31 \pm 0.34$ | $0.42 \pm 0.30$ | $0.46 \pm 0.30$ |

### D.4 DETAILED ANALYSIS OF ATTRIBUTE-LEVEL DIVERSITY

While global metrics like FID and Vendi Score are useful, they do not fully capture two critical aspects of a generative model's behavior: its inherent biases and its fine-grained controllability. To address this, we adopt the DIM/CIM framework proposed by Teotia et al. (2025). This framework is designed to evaluate the crucial trade-off between a model's spontaneous creativity and its ability to follow specific instructions.

To analyze the trade-off between spontaneous diversity and explicit control, we employ two complementary metrics: Default-mode Diversity (DIM) and Conditional Generalization (CIM). The DIM metric quantifies a model's default-mode bias when given a coarse, underspecified prompt (e.g., "a photo of a bus"), where a score closer to zero indicates a more balanced and diverse output. In contrast, the CIM metric evaluates controllability by measuring whether a model can reliably produce a specific, requested attribute from a dense, explicit prompt (e.g., "a photo of a red bus"), where a high score is better. By evaluating both metrics, we can comprehensively assess whether our method's improvement in diversity comes at the cost of controllability, directly addressing the core trade-off.

**Prompt Generation Methodology** To evaluate attribute-level performance using the DIM and CIM metrics, we created a structured set of prompts for each concept. The process begins with a real-world caption from the COCO dataset, which we term the `coco_seed_caption`. From this, we derive a generic `coarse_prompt` by removing specific attributes (e.g., "yellow"). This coarse prompt is used to evaluate the model's default-mode bias. Subsequently, we programmatically insert a variety of attributes (e.g., different colors and body types) back into the coarse prompt template to generate a list of `dense_prompts`. This set of specific prompts is used to evaluate the model's conditional generalization and instruction-following ability. Listing D.4 shows one complete example of this structure for the 'bus' concept, demonstrating how one seed caption yields an entire set of evaluation prompts.

Table 5: Comprehensive quantitative comparison of our method against all baselines for the "bicycle" concept. The results demonstrate our method's performance in enhancing diversity while maintaining fidelity and prompt alignment across different guidance scales.

| Method | Guidance Scale (CFG) | | | Guidance Scale (CFG) | | |
|---|---|---|---|---|---|---|
| | 3.0 | 5.0 | 7.5 | 3.0 | 5.0 | 7.5 |
| | Brisque ↓ | | | $1 - $ MS-SSIM(%) ↑ | | |
| FM-SD3.5 | $25.84 \pm 1.97$ | $27.03 \pm 2.04$ | $30.76 \pm 1.89$ | $88.40 \pm 1.06$ | $89.33 \pm 1.34$ | $89.57 \pm 1.51$ |
| PG | $49.33 \pm 2.20$ | $63.71 \pm 1.86$ | $69.05 \pm 4.21$ | $86.83 \pm 2.12$ | $88.93 \pm 2.30$ | $89.26 \pm 2.90$ |
| CADS | $28.48 \pm 3.58$ | $32.54 \pm 2.67$ | $34.00 \pm 2.58$ | $91.48 \pm 2.59$ | $92.99 \pm 2.24$ | $92.51 \pm 2.35$ |
| DPP | $21.19 \pm 2.77$ | $23.75 \pm 2.40$ | $26.44 \pm 3.99$ | $87.02 \pm 4.41$ | $89.19 \pm 3.95$ | $90.11 \pm 3.67$ |
| **Ours** | $20.84 \pm 3.25$ | $24.84 \pm 3.09$ | $26.14 \pm 2.80$ | $90.74 \pm 2.25$ | $91.81 \pm 2.27$ | $91.48 \pm 1.49$ |
| | Vendi Score Pixel ↑ | | | Vendi Score Inception ↑ | | |
| FM-SD3.5 | $2.26 \pm 0.24$ | $2.38 \pm 0.25$ | $2.46 \pm 0.25$ | $3.59 \pm 0.32$ | $3.10 \pm 0.30$ | $2.90 \pm 0.25$ |
| PG | $2.06 \pm 0.11$ | $2.18 \pm 0.07$ | $2.29 \pm 0.05$ | $3.50 \pm 0.07$ | $3.06 \pm 0.10$ | $2.97 \pm 0.07$ |
| CADS | $2.30 \pm 0.04$ | $2.37 \pm 0.06$ | $2.48 \pm 0.04$ | $3.76 \pm 0.28$ | $3.19 \pm 0.10$ | $2.94 \pm 0.07$ |
| DPP | $2.28 \pm 0.02$ | $2.34 \pm 0.01$ | $2.43 \pm 0.03$ | $3.83 \pm 0.12$ | $3.15 \pm 0.18$ | $2.86 \pm 0.06$ |
| **Ours** | $2.55 \pm 0.17$ | $2.51 \pm 0.07$ | $2.61 \pm 0.03$ | $3.72 \pm 0.13$ | $3.24 \pm 0.15$ | $2.99 \pm 0.11$ |
| | CLIP-IQA ↑ | | | CLIP Score ↑ | | |
| FM-SD3.5 | $0.43 \pm 0.41$ | $0.51 \pm 0.38$ | $0.48 \pm 0.38$ | $29.53 \pm 0.30$ | $29.26 \pm 0.29$ | $29.06 \pm 0.26$ |
| PG | $0.42 \pm 0.42$ | $0.51 \pm 0.33$ | $0.49 \pm 0.31$ | $29.86 \pm 0.22$ | $29.09 \pm 0.24$ | $28.77 \pm 0.13$ |
| CADS | $0.47 \pm 0.35$ | $0.52 \pm 0.32$ | $0.53 \pm 0.34$ | $29.43 \pm 0.26$ | $29.10 \pm 0.27$ | $28.85 \pm 0.15$ |
| DPP | $0.42 \pm 0.38$ | $0.55 \pm 0.32$ | $0.62 \pm 0.29$ | $29.85 \pm 0.16$ | $29.56 \pm 0.15$ | $29.45 \pm 0.11$ |
| **Ours** | $0.48 \pm 0.38$ | $0.55 \pm 0.33$ | $0.58 \pm 0.31$ | $29.81 \pm 0.17$ | $29.31 \pm 0.22$ | $28.97 \pm 0.06$ |
| | FID ↓ | | | Image Reward ↑ | | |
| FM-SD3.5 | $149.21 \pm 1.85$ | $148.28 \pm 1.45$ | $147.06 \pm 3.90$ | $6.25 \pm 0.70$ | $6.35 \pm 0.58$ | $6.29 \pm 0.66$ |
| PG | $149.34 \pm 3.05$ | $151.45 \pm 1.68$ | $149.44 \pm 4.26$ | $6.18 \pm 0.70$ | $6.27 \pm 0.68$ | $6.22 \pm 0.68$ |
| CADS | $149.27 \pm 1.59$ | $149.06 \pm 1.20$ | $149.24 \pm 3.90$ | $6.28 \pm 0.67$ | $6.31 \pm 0.27$ | $6.25 \pm 0.70$ |
| DPP | $146.59 \pm 2.06$ | $148.14 \pm 0.93$ | $145.49 \pm 3.87$ | $6.14 \pm 0.69$ | $6.27 \pm 0.69$ | $6.29 \pm 0.68$ |
| **Ours** | $146.36 \pm 3.48$ | $148.50 \pm 1.33$ | $146.99 \pm 4.80$ | $6.30 \pm 0.54$ | $6.44 \pm 0.51$ | $6.42 \pm 0.63$ |

Table 6: Quantitative comparison of our method against baselines across different CFG levels for the "a photo of a truck" prompt under the "truck" concept. Our method consistently improves diversity metrics while maintaining competitive or better image quality and alignment.

| Method | Guidance Scale (CFG) | | | Guidance Scale (CFG) | | |
|---|---|---|---|---|---|---|
| | 3.0 | 5.0 | 7.5 | 3.0 | 5.0 | 7.5 |
| | Brisque ↓ | | | $1 - $ MS-SSIM(%) ↑ | | |
| PG | $34.45 \pm 5.69$ | $34.18 \pm 1.05$ | $35.77 \pm 3.94$ | $89.29 \pm 1.48$ | $91.64 \pm 1.31$ | $91.60 \pm 1.34$ |
| CADS | $31.10 \pm 2.25$ | $33.65 \pm 1.12$ | $37.59 \pm 1.42$ | $88.25 \pm 0.77$ | $90.19 \pm 0.55$ | $92.01 \pm 0.55$ |
| DPP | $23.84 \pm 1.24$ | $29.56 \pm 1.63$ | $36.76 \pm 1.11$ | $89.82 \pm 1.19$ | $91.11 \pm 1.34$ | $91.26 \pm 1.15$ |
| **Ours** | $26.98 \pm 0.95$ | $29.85 \pm 0.85$ | $37.57 \pm 1.51$ | $90.75 \pm 2.15$ | $91.80 \pm 2.51$ | $91.80 \pm 1.79$ |
| | Vendi Score Pixel ↑ | | | Vendi Score Inception ↑ | | |
| PG | $3.33 \pm 0.25$ | $3.56 \pm 0.27$ | $3.64 \pm 0.24$ | $5.53 \pm 0.13$ | $4.92 \pm 0.20$ | $4.68 \pm 0.12$ |
| CADS | $3.71 \pm 0.24$ | $3.72 \pm 0.17$ | $3.73 \pm 0.11$ | $5.90 \pm 0.17$ | $4.72 \pm 0.24$ | $4.43 \pm 0.18$ |
| DPP | $3.65 \pm 0.14$ | $3.83 \pm 0.17$ | $3.79 \pm 0.13$ | $5.86 \pm 0.34$ | $4.93 \pm 0.16$ | $4.30 \pm 0.29$ |
| **Ours** | $3.79 \pm 0.11$ | $3.86 \pm 0.15$ | $4.01 \pm 0.15$ | $5.89 \pm 0.34$ | $5.12 \pm 0.35$ | $4.77 \pm 0.19$ |
| | FID ↓ | | | CLIP Score ↑ | | |
| PG | $143.51 \pm 3.35$ | $141.83 \pm 1.83$ | $143.63 \pm 2.18$ | $27.82 \pm 0.21$ | $27.45 \pm 0.20$ | $27.46 \pm 0.13$ |
| CADS | $130.65 \pm 1.24$ | $129.41 \pm 1.58$ | $129.38 \pm 0.88$ | $27.54 \pm 0.15$ | $27.22 \pm 0.09$ | $27.04 \pm 0.11$ |
| DPP | $143.87 \pm 3.69$ | $139.43 \pm 1.63$ | $141.05 \pm 3.98$ | $27.63 \pm 0.09$ | $27.14 \pm 0.10$ | $27.01 \pm 0.12$ |
| **Ours** | $130.13 \pm 0.96$ | $128.18 \pm 0.20$ | $126.89 \pm 1.58$ | $27.75 \pm 0.04$ | $27.50 \pm 0.11$ | $27.49 \pm 0.12$ |

---

**Appendix Listing: Example of DIM/CIM Prompt Structure**

```
{
    "coco_seed_caption": "A yellow bus stopping at a bus stop in a
    city.",
```

Table 7: Quantitative comparison of our method against baselines across different CFG levels for the "a truck" prompt under the "truck" concept. Our method consistently improves diversity metrics while maintaining competitive or better image quality and alignment.

| Method | Guidance Scale (CFG) | | | Guidance Scale (CFG) | | |
|---|---|---|---|---|---|---|
| | 3.0 | 5.0 | 7.5 | 3.0 | 5.0 | 7.5 |
| | Brisque ↓ | | | $1 - $ MS-SSIM(%) ↑ | | |
| PG | $43.88 \pm 3.47$ | $36.40 \pm 3.20$ | $40.30 \pm 2.48$ | $86.44 \pm 2.63$ | $86.08 \pm 1.96$ | $83.81 \pm 1.97$ |
| CADS | $22.52 \pm 1.61$ | $23.59 \pm 1.61$ | $28.37 \pm 1.48$ | $86.29 \pm 2.15$ | $85.71 \pm 2.26$ | $84.46 \pm 2.65$ |
| DPP | $22.18 \pm 1.57$ | $25.77 \pm 1.33$ | $32.05 \pm 0.92$ | $88.33 \pm 0.91$ | $87.47 \pm 1.86$ | $85.58 \pm 1.32$ |
| **Ours** | $19.50 \pm 2.04$ | $22.31 \pm 1.62$ | $27.65 \pm 1.72$ | $90.20 \pm 1.94$ | $88.60 \pm 1.24$ | $87.73 \pm 1.38$ |
| | Vendi Score Pixel ↑ | | | Vendi Score Inception ↑ | | |
| PG | $4.63 \pm 0.27$ | $4.21 \pm 0.13$ | $4.09 \pm 0.07$ | $2.49 \pm 0.17$ | $2.42 \pm 0.14$ | $2.38 \pm 0.11$ |
| CADS | $4.63 \pm 0.41$ | $3.95 \pm 0.24$ | $3.63 \pm 0.11$ | $2.69 \pm 0.16$ | $2.53 \pm 0.09$ | $2.47 \pm 0.08$ |
| DPP | $4.61 \pm 0.26$ | $3.97 \pm 0.20$ | $3.72 \pm 0.07$ | $2.59 \pm 0.07$ | $2.42 \pm 0.06$ | $2.31 \pm 0.06$ |
| **Ours** | $4.79 \pm 0.28$ | $4.29 \pm 0.16$ | $4.14 \pm 0.19$ | $2.82 \pm 0.11$ | $2.66 \pm 0.11$ | $2.51 \pm 0.06$ |
| | FID ↓ | | | CLIP Score ↑ | | |
| PG | $142.82 \pm 3.12$ | $142.88 \pm 2.71$ | $142.07 \pm 5.60$ | $27.65 \pm 0.15$ | $26.70 \pm 0.12$ | $26.48 \pm 0.17$ |
| CADS | $126.75 \pm 1.23$ | $125.96 \pm 0.52$ | $125.17 \pm 0.64$ | $27.19 \pm 0.09$ | $26.57 \pm 0.07$ | $26.57 \pm 0.14$ |
| DPP | $139.64 \pm 4.07$ | $138.31 \pm 2.76$ | $138.62 \pm 4.79$ | $27.82 \pm 0.10$ | $27.30 \pm 0.10$ | $26.88 \pm 0.09$ |
| **Ours** | $128.80 \pm 1.27$ | $127.64 \pm 1.30$ | $126.17 \pm 1.45$ | $27.65 \pm 0.12$ | $27.18 \pm 0.11$ | $26.80 \pm 0.10$ |

Table 8: Quantitative comparison of our method against baselines across different CFG levels for the "a photo of a apple" prompt under the "apple" concept.

| Method | Guidance Scale (CFG) | | | Guidance Scale (CFG) | | |
|---|---|---|---|---|---|---|
| | 3.0 | 5.0 | 7.5 | 3.0 | 5.0 | 7.5 |
| | Brisque ↓ | | | $1 - $ MS-SSIM(%) ↑ | | |
| PG | $20.50 \pm 0.65$ | $25.12 \pm 1.63$ | $30.74 \pm 3.28$ | $56.00 \pm 2.08$ | $55.55 \pm 2.53$ | $56.60 \pm 1.89$ |
| CADS | $23.76 \pm 4.87$ | $25.64 \pm 5.68$ | $30.82 \pm 2.27$ | $58.16 \pm 3.29$ | $58.92 \pm 3.01$ | $59.89 \pm 3.40$ |
| DPP | $19.69 \pm 2.54$ | $26.36 \pm 2.72$ | $31.08 \pm 4.26$ | $56.97 \pm 3.76$ | $57.45 \pm 4.96$ | $56.51 \pm 5.69$ |
| **Ours** | $17.71 \pm 1.81$ | $25.53 \pm 0.51$ | $27.96 \pm 1.89$ | $57.97 \pm 3.56$ | $55.79 \pm 3.00$ | $57.56 \pm 2.90$ |
| | Vendi Score Pixel ↑ | | | Vendi Score Inception ↑ | | |
| PG | $2.06 \pm 0.06$ | $2.04 \pm 0.09$ | $2.12 \pm 0.11$ | $2.12 \pm 0.08$ | $1.91 \pm 0.02$ | $1.99 \pm 0.18$ |
| CADS | $1.87 \pm 0.30$ | $1.78 \pm 0.22$ | $1.77 \pm 0.22$ | $2.19 \pm 0.17$ | $1.96 \pm 0.03$ | $1.93 \pm 0.12$ |
| DPP | $1.87 \pm 0.27$ | $1.85 \pm 0.22$ | $1.86 \pm 0.25$ | $2.06 \pm 0.09$ | $2.03 \pm 0.03$ | $1.99 \pm 0.13$ |
| **Ours** | $2.19 \pm 0.06$ | $2.13 \pm 0.08$ | $2.22 \pm 0.08$ | $2.27 \pm 0.04$ | $2.07 \pm 0.08$ | $2.07 \pm 0.13$ |
| | FID ↓ | | | CLIP Score ↑ | | |
| PG | $152.83 \pm 0.83$ | $149.68 \pm 0.12$ | $151.91 \pm 0.70$ | $31.84 \pm 0.09$ | $31.79 \pm 0.07$ | $31.83 \pm 0.23$ |
| CADS | $154.05 \pm 3.25$ | $151.76 \pm 1.39$ | $150.88 \pm 2.44$ | $31.79 \pm 0.01$ | $31.81 \pm 0.15$ | $31.69 \pm 0.07$ |
| DPP | $151.97 \pm 1.72$ | $149.80 \pm 1.30$ | $149.99 \pm 2.25$ | $31.87 \pm 0.09$ | $31.84 \pm 0.12$ | $31.88 \pm 0.14$ |
| **Ours** | $152.95 \pm 1.00$ | $150.35 \pm 2.84$ | $151.53 \pm 1.48$ | $31.88 \pm 0.12$ | $31.81 \pm 0.14$ | $31.87 \pm 0.19$ |

```
"coarse_prompt": "A bus stopping at a bus stop in a city.",
"dense_prompts": [
    {
        "attribute_type": "color",
        "attribute": "red",
        "dense_prompt": "A red bus stopping at a bus stop in a
city."
    },
    {
        "attribute_type": "body_type",
        "attribute": "double-decker",
        "dense_prompt": "A double-decker bus stopping at a bus
stop in a city."
    }
]
```

Table 9: Quantitative comparison of our method against baselines across different CFG levels for the "a photo of a suitcase" prompt under the "suitcase" concept.

| | Guidance Scale (CFG) | | | Guidance Scale (CFG) | | |
|---|---|---|---|---|---|---|
| **Method** | 3.0 | 5.0 | 7.5 | 3.0 | 5.0 | 7.5 |
| | **Brisque** ↓ | | | **1 − MS-SSIM(%)** ↑ | | |
| PG | $58.07 \pm 4.75$ | $43.32 \pm 2.48$ | $36.44 \pm 3.20$ | $74.11 \pm 4.74$ | $75.69 \pm 1.39$ | $77.33 \pm 1.73$ |
| CADS | $54.98 \pm 8.24$ | $83.71 \pm 2.57$ | $100.60 \pm 7.59$ | $84.36 \pm 1.97$ | $87.02 \pm 5.40$ | $86.12 \pm 3.97$ |
| DPP | $27.66 \pm 6.52$ | $29.07 \pm 1.85$ | $33.48 \pm 5.31$ | $76.76 \pm 5.98$ | $78.10 \pm 5.06$ | $80.45 \pm 2.46$ |
| **Ours** | $20.38 \pm 1.83$ | $17.35 \pm 1.51$ | $14.97 \pm 1.55$ | $82.99 \pm 3.49$ | $83.36 \pm 2.41$ | $84.77 \pm 1.85$ |
| | **Vendi Score Pixel** ↑ | | | **Vendi Score Inception** ↑ | | |
| PG | $2.72 \pm 0.20$ | $2.83 \pm 0.08$ | $2.84 \pm 0.14$ | $5.53 \pm 0.28$ | $5.06 \pm 0.30$ | $4.79 \pm 0.49$ |
| CADS | $2.75 \pm 0.29$ | $2.93 \pm 0.12$ | $3.01 \pm 0.11$ | $5.50 \pm 0.13$ | $5.04 \pm 0.51$ | $4.46 \pm 0.39$ |
| DPP | $3.01 \pm 0.29$ | $3.26 \pm 0.23$ | $3.18 \pm 0.10$ | $5.17 \pm 0.13$ | $4.57 \pm 0.13$ | $4.24 \pm 0.29$ |
| **Ours** | $3.18 \pm 0.36$ | $3.43 \pm 0.06$ | $3.61 \pm 0.04$ | $5.73 \pm 0.13$ | $5.45 \pm 0.27$ | $4.87 \pm 0.42$ |
| | **FID** ↓ | | | **CLIP Score** ↑ | | |
| PG | $158.69 \pm 3.64$ | $156.19 \pm 2.06$ | $154.75 \pm 1.88$ | $30.12 \pm 0.11$ | $29.85 \pm 0.34$ | $29.76 \pm 0.14$ |
| CADS | $166.04 \pm 4.10$ | $161.58 \pm 0.62$ | $158.54 \pm 4.39$ | $29.37 \pm 0.82$ | $29.07 \pm 0.36$ | $29.10 \pm 0.20$ |
| DPP | $156.42 \pm 2.06$ | $156.26 \pm 1.94$ | $155.99 \pm 3.09$ | $30.33 \pm 0.28$ | $29.84 \pm 0.15$ | $29.72 \pm 0.12$ |
| **Ours** | $159.20 \pm 2.90$ | $153.91 \pm 3.99$ | $151.90 \pm 1.63$ | $29.54 \pm 0.62$ | $29.51 \pm 0.19$ | $29.59 \pm 0.05$ |

Table 10: Quantitative comparison of our method against baselines across different CFG levels for the "a photo of a pizza" prompt under the "pizza" concept.

| | Guidance Scale (CFG) | | | Guidance Scale (CFG) | | |
|---|---|---|---|---|---|---|
| **Method** | 3.0 | 5.0 | 7.5 | 3.0 | 5.0 | 7.5 |
| | **Brisque** ↓ | | | **1 − MS-SSIM(%)** ↑ | | |
| PG | $31.07 \pm 4.46$ | $19.28 \pm 3.16$ | $21.19 \pm 1.74$ | $90.90 \pm 0.95$ | $91.04 \pm 1.75$ | $92.11 \pm 1.14$ |
| CADS | $18.23 \pm 0.35$ | $18.24 \pm 1.18$ | $25.06 \pm 8.83$ | $93.29 \pm 0.73$ | $91.18 \pm 1.74$ | $90.22 \pm 2.64$ |
| DPP | $20.36 \pm 0.76$ | $20.46 \pm 0.41$ | $32.35 \pm 7.69$ | $93.61 \pm 0.50$ | $92.77 \pm 0.39$ | $93.54 \pm 1.04$ |
| **Ours** | $18.13 \pm 0.81$ | $17.39 \pm 0.20$ | $27.25 \pm 1.58$ | $94.45 \pm 0.69$ | $93.76 \pm 1.37$ | $94.05 \pm 1.69$ |
| | **Vendi Score Pixel** ↑ | | | **Vendi Score Inception** ↑ | | |
| PG | $1.79 \pm 0.01$ | $1.99 \pm 0.04$ | $2.44 \pm 0.10$ | $2.31 \pm 0.05$ | $2.18 \pm 0.11$ | $2.66 \pm 0.11$ |
| CADS | $1.84 \pm 0.04$ | $1.86 \pm 0.05$ | $1.92 \pm 0.08$ | $2.30 \pm 0.04$ | $2.16 \pm 0.08$ | $2.33 \pm 0.46$ |
| DPP | $1.91 \pm 0.04$ | $2.11 \pm 0.08$ | $2.35 \pm 0.13$ | $2.36 \pm 0.08$ | $2.19 \pm 0.11$ | $2.60 \pm 0.31$ |
| **Ours** | $2.05 \pm 0.02$ | $2.24 \pm 0.05$ | $2.60 \pm 0.06$ | $2.88 \pm 0.09$ | $2.41 \pm 0.10$ | $2.70 \pm 0.22$ |
| | **FID** ↓ | | | **CLIP Score** ↑ | | |
| PG | $136.71 \pm 2.75$ | $136.36 \pm 2.40$ | $141.19 \pm 2.80$ | $30.32 \pm 0.16$ | $30.29 \pm 0.07$ | $29.80 \pm 0.17$ |
| CADS | $134.44 \pm 0.66$ | $135.62 \pm 0.49$ | $138.73 \pm 3.63$ | $29.69 \pm 0.09$ | $29.58 \pm 0.34$ | $29.32 \pm 0.07$ |
| DPP | $134.22 \pm 3.54$ | $135.69 \pm 2.56$ | $140.73 \pm 4.90$ | $29.86 \pm 0.18$ | $29.86 \pm 0.11$ | $29.27 \pm 0.27$ |
| **Ours** | $131.93 \pm 1.74$ | $132.46 \pm 3.04$ | $139.88 \pm 2.13$ | $29.89 \pm 0.25$ | $29.81 \pm 0.16$ | $29.44 \pm 0.23$ |

**Qualitative Comparison.** To provide a qualitative sense of the performance differences, we present a visual comparison against all baselines in Figure 11. The images in this figure were generated using the dense prompt "A single-decked bus is parked on the street" for the first row, and other representative prompts for the subsequent rows. The results visually confirm our method's superior ability to generate a more diverse set of outputs while maintaining high fidelity, in line with the quantitative findings. You can see more results in Sec G.

Figure 11: Qualitative comparison of default-mode diversity using **coarse prompts**.

| (a) CADS | (b) DPP | (c) PG | (d) Our Method |
|---|---|---|---|

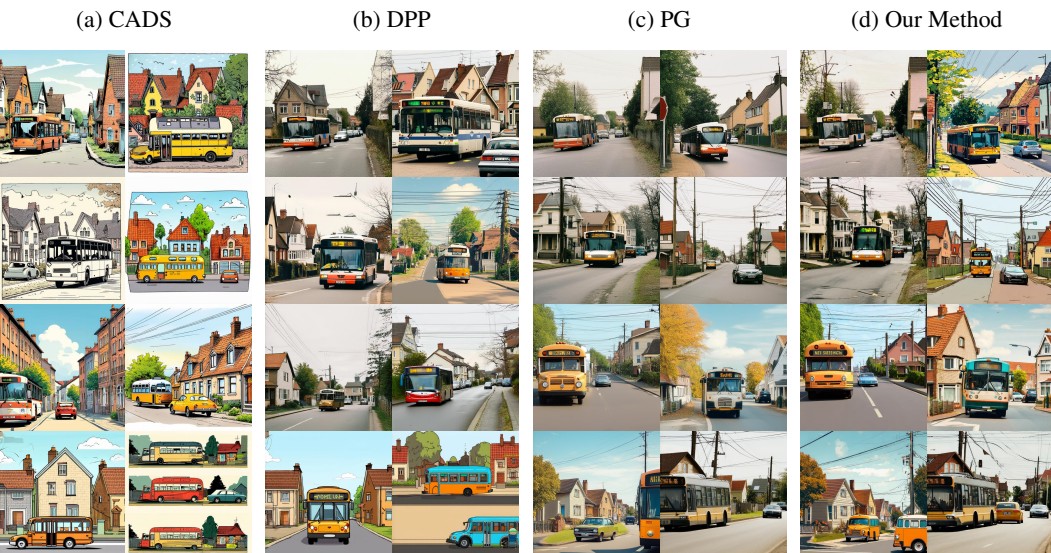

# E ADDITIONAL EXPERIMENTS

## E.1 TOY EXAMPLE 1: BEHAVIOR IN HIGH-PARTICLE REGIMES

In the main text Fig 2, we demonstrated OSCAR's ability to improve mode coverage using a moderate number of particles ($N = 200$). Here, we analyze the behavior of our method in the high-particle regime ($N = 2000$) to illustrate the inherent trade-off between support coverage and density matching. As shown in Fig 12, when the number of particles is significantly increased without scaling down the guidance strength $\gamma(t)$, the generated samples tend to form a uniform distribution over the support of the target distribution, rather than strictly adhering to the local Gaussian density peaks. In the case of the 2D Mixture of Gaussians, this results in "square-like" patches that fully cover the modes but deviate from the Gaussian bell shape.

This phenomenon is a direct consequence of our core objective: maximizing the Feature Volume. The control signal $g(x, t)$ acts effectively as a repulsive force between trajectories. From a physics perspective, when a large number of mutually repulsive particles are confined within a potential well, the configuration that minimizes total potential energy is a uniform distribution rather than a concentrated cluster. Consequently, in high-density regimes, OSCAR prioritizes covering the support, exploring the tails and boundaries of the valid region, over matching the exact probability density of the mode centers.

Rigorously, this behavior is predicted by the Girsanov Representation derived in Appendix B Theorem 4. The KL divergence between the OSCAR sampling distribution $\mathbb{Q}$ and the baseline distribution $\mathbb{P}$ is governed by the energy cost of the guidance: $\mathcal{D}_{KL}(\mathbb{Q}\|\mathbb{P}) = \frac{1}{2}\mathbb{E}_{\mathbb{Q}}[\int \|g\|^2/\beta dt]$. As $N$ increases, the magnitude of the collective repulsive gradients $\|g\|^2$ grows significantly. This implies a larger upper bound on the divergence, theoretically predicting that the sampled distribution will deviate more significantly from the baseline density to achieve maximum entropy.

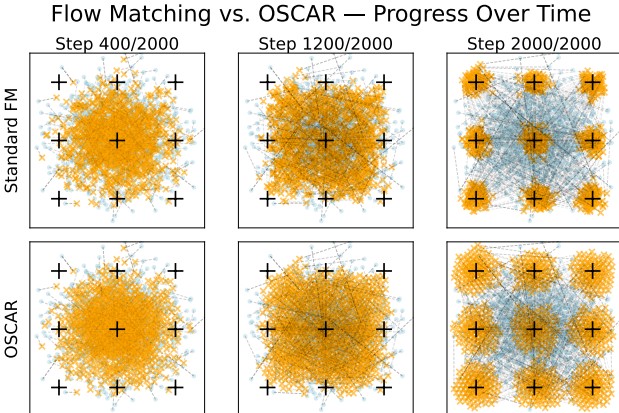

Figure 12: When the number of particles is significantly increased without a corresponding reduction in guidance strength, the cumulative repulsive force dominates the base flow. This causes the samples to arrange themselves into a uniform distribution to minimize system energy, effectively prioritizing maximum support coverage over precise density matching.

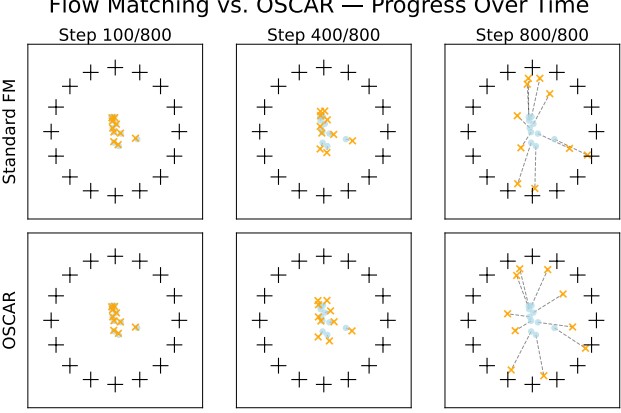

Figure 13: Toy example on a ring of Gaussian modes. Rows: Standard Flow Matching (top) vs. OSCAR (bottom). Columns: three snapshots (early, mid, final) under the same NFE. Black plus signs mark the mode centers on the ring; light-blue circles are the shared initial particles sampled uniformly from an inner disk; orange x marks are the particles at each snapshot, with dashed lines connecting initial and current positions. Compared with standard FM, OSCAR spreads particles tangentially and covers more modes earlier and more uniformly by the final step, under the same sampling budget.

### E.2 Toy Example 2: Intuitive Visualization on a 2D

To illustrate the diversity gains afforded by our method, we design a two-dimensional toy setting where the target distribution is a ring-structured Gaussian mixture with equally spaced modes. Initial particles are drawn uniformly from a smaller concentric disk. We compare standard FM and OSCAR two samplers, using the same set of initial particles and the same NFE. As shown in the Fig. 13, at early and intermediate stages, FM primarily induces radial contraction and tends to cluster particles within a few angular sectors. In contrast, OSCAR maintains radial convergence while introducing tangential diversification, which drives particles to spread across sectors more quickly and more uniformly. By the final snapshot, OSCAR covers substantially more modes than FM and exhibits a more uniform angular distribution of particles, demonstrating superior mode coverage and overall diversity under an identical computational budget.

### E.3 IMAGENET-400 SINGLE-SAMPLE EVALUATION

As an additional sanity check that our diversity-aware control does not degrade per-sample fidelity, we conduct a class-conditional experiment on ImageNet. We randomly sample 400 classes from ImageNet-1K and, for each class, construct a single class-name prompt and generate one image per method. We then evaluate all methods on the resulting set of 400 images. As reported in Tab. 11, our method achieves the best performance on both variants of the Vendi Score (pixel- and Inception-based), indicating that the induced stochastic control does not collapse to a narrow subset of visual patterns. At the same time, OSCAR attains the lowest FID among all competitors, and matches baselines on CLIP score and other fidelity-oriented metrics. Taken together, these results confirm that our method preserves, and can even slightly improve, overall image quality while maintaining strong diversity characteristics even in this challenging single-sample-per-class regime.

Table 11: Evaluation on 400 single-image ImageNet-style prompts (1 per class). This unbiased setting tests for any systematic degradation of single-image fidelity.

| Method | FID $\downarrow$ | Vendi (Pixel) $\uparrow$ | Vendi (Inception) $\uparrow$ | CLIP Score $\uparrow$ |
|---|---|---|---|---|
| FM-SD3.5 | 100.4 | 3.38 | 35.70 | $18.14 \pm 0.25$ |
| DPP | 100.3 | 3.34 | 36.03 | $18.11 \pm 0.25$ |
| CADS | 101.1 | 3.52 | 35.91 | $17.96 \pm 0.25$ |
| PG | 99.7 | 3.66 | 34.34 | $18.13 \pm 0.24$ |
| **Oscar** | **99.3** | **3.85** | **36.40** | $\mathbf{18.08 \pm 0.24}$ |

### E.4 FAIR-BUDGET COMPARISON

We first compare the computational cost of Oscar against various baselines under the same generation configuration, as shown in Table 12. In terms of FLOPs and runtime, Oscar introduces only a modest computational overhead compared to the baseline FM-SD3.5. However, this overhead is significantly lower than that of DPP, which requires more than double the computation and time of the baseline. This result empirically validates our claim in Section 2 that our method significantly reduces the computational complexity inherent in DPP. Regarding peak GPU VRAM usage, all methods exhibit roughly comparable memory consumption.

Table 12: Computational cost comparison under fixed generation settings (NFE=30, CFG=5.0, batch size=32). We report total theoretical computation, wall-clock time, and peak VRAM usage.

| Variant | FLOPs(G) $\downarrow$ | Time (seconds/run) $\downarrow$ | Peak VRAM(GB) $\downarrow$ |
|---|---|---|---|
| FM-SD3.5 | 4093.4 | 237.8 | 19.2 |
| DPP | 9045.1 | 990.2 | 19.5 |
| CADS | 4093.4 | 231.2 | 20.0 |
| PG | 4093.4 | 229.6 | 26.4 |
| Oscar (Ours) | 5534.6 | 451.4 | 18.2 |

Costs measured on an NVIDIA A6000 GPU, 512x512 resolution, batch size=32.

To conduct a fairer comparison, we further evaluate Oscar against FM-SD3.5 under a matched computational budget, as detailed in Table 13. We match the cost of Oscar against two enhanced FM-SD3.5 variants: one with increased NFE to 40 and another with an increased number of particles to 48. The results show that even at a matched computational cost, our method comprehensively outperforms the baselines on most metrics. Although the FM-SD3.5 variant with increased particles achieves a better FID score, we note that the FID metric is known to be highly sensitive to the number of samples when the evaluation set is small.

Table 13: Compute-matched comparison with standard Flow Matching model under matched FLOPs.

| Method | Compute budget | | | Quality & diversity metrics | | | |
| | NFE | Particles | FLOPs (T/run) | CLIP↑ | Vendi (Pixel)↑ | FID↓ | BRISQUE↓ |
| --- | --- | --- | --- | --- | --- | --- | --- |
| FM-SD3.5 | 30 | 32 | 4.09 | $28.24 \pm 0.18$ | $2.45 \pm 0.13$ | $164.4 \pm 1.8$ | $23.4 \pm 1.4$ |
| +K particles | 30 | 48 | 6.14 | $28.15 \pm 0.23$ | $2.60 \pm 0.21$ | $\mathbf{149.7 \pm 1.3}$ | $23.5 \pm 1.7$ |
| +N NFE | 40 | 32 | 5.43 | $28.15 \pm 0.21$ | $2.40 \pm 0.15$ | $165.3 \pm 1.8$ | $24.6 \pm 1.7$ |
| **OSCAR** | 30 | 32 | 5.53 | $\mathbf{28.26 \pm 0.22}$ | $\mathbf{2.86 \pm 0.05}$ | $163.3 \pm 1.6$ | $\mathbf{21.2 \pm 1.5}$ |

## E.5 PERFORMANCE ACROSS DIFFERENT SAMPLING STEPS

To demonstrate the robustness and efficiency of our method across various computational budgets, we first analyze its performance as a function of the NFE. The quantitative results, presented in Table 14, confirm that our method consistently outperforms all baselines across the tested NFE settings of 10, 20, and 40. Specifically, our method achieves a significant and stable advantage in diversity, leading in both Vendi Score Pixel and Vendi Score Inception scores at every step count. This is accomplished without sacrificing quality; our method attains the best Brisque score at every NFE level and maintains a highly competitive FID score, confirming a high degree of overall distributional fidelity.

Table 14: Quantitative comparison of our method against baselines across different NFE. Our method consistently improves diversity metrics while achieving state-of-the-art performance on quality and alignment scores.

| Method | Number of Function Evaluations (NFE) | | | Number of Function Evaluations (NFE) | | |
| | 40 | 20 | 10 | 40 | 20 | 10 |
| --- | --- | --- | --- | --- | --- | --- |
| | Brisque ↓ | | | 1 - MS-SSIM(%) ↑ | | |
| PG | $48.75 \pm 2.85$ | $71.26 \pm 3.24$ | $105.04 \pm 3.94$ | $79.40 \pm 0.46$ | $79.27 \pm 0.73$ | $79.07 \pm 0.66$ |
| CADS | $15.74 \pm 0.73$ | $27.51 \pm 0.95$ | $56.84 \pm 1.81$ | $81.61 \pm 0.65$ | $80.81 \pm 0.61$ | $80.49 \pm 0.83$ |
| DPP | $13.80 \pm 1.41$ | $26.51 \pm 1.98$ | $61.20 \pm 2.23$ | $81.05 \pm 0.58$ | $80.62 \pm 0.71$ | $80.00 \pm 0.41$ |
| **Ours** | $\mathbf{12.30 \pm 1.25}$ | $\mathbf{17.73 \pm 1.21}$ | $\mathbf{44.04 \pm 1.38}$ | $\mathbf{83.47 \pm 0.42}$ | $\mathbf{82.02 \pm 0.72}$ | $\mathbf{82.00 \pm 0.73}$ |
| | Vendi Score Pixel ↑ | | | Vendi Score Inception ↑ | | |
| PG | $1.74 \pm 0.06$ | $1.72 \pm 0.06$ | $1.64 \pm 0.05$ | $2.78 \pm 0.15$ | $2.77 \pm 0.07$ | $2.73 \pm 0.05$ |
| CADS | $1.97 \pm 0.04$ | $1.94 \pm 0.05$ | $1.77 \pm 0.06$ | $2.84 \pm 0.40$ | $2.81 \pm 0.07$ | $2.83 \pm 0.06$ |
| DPP | $1.88 \pm 0.06$ | $1.85 \pm 0.06$ | $1.75 \pm 0.05$ | $2.87 \pm 0.07$ | $2.86 \pm 0.12$ | $2.77 \pm 0.10$ |
| **Ours** | $\mathbf{2.07 \pm 0.07}$ | $\mathbf{2.06 \pm 0.06}$ | $\mathbf{1.93 \pm 0.05}$ | $\mathbf{2.93 \pm 0.08}$ | $\mathbf{2.88 \pm 0.05}$ | $\mathbf{2.87 \pm 0.12}$ |
| | FID ↓ | | | CLIP Score ↑ | | |
| PG | $167.34 \pm 2.12$ | $166.50 \pm 2.39$ | $165.39 \pm 2.03$ | $\mathbf{19.11 \pm 0.40}$ | $\mathbf{18.93 \pm 0.36}$ | $18.23 \pm 0.76$ |
| CADS | $165.84 \pm 0.95$ | $166.13 \pm 1.22$ | $167.34 \pm 1.03$ | $18.97 \pm 0.38$ | $18.90 \pm 0.46$ | $\mathbf{18.35 \pm 0.63}$ |
| DPP | $166.51 \pm 1.52$ | $165.89 \pm 0.72$ | $166.21 \pm 1.28$ | $18.97 \pm 0.42$ | $18.63 \pm 0.46$ | $17.91 \pm 3.94$ |
| **Ours** | $\mathbf{165.40 \pm 0.94}$ | $\mathbf{164.75 \pm 0.66}$ | $165.31 \pm 0.79$ | $18.77 \pm 0.46$ | $18.58 \pm 0.58$ | $18.11 \pm 0.69$ |
| | CLIP-IQA ↑ | | | Image Reward ↑ | | |
| PG | $7.57 \pm 0.24$ | $7.52 \pm 0.23$ | $7.41 \pm 0.25$ | $1.26 \pm 0.20$ | $1.17 \pm 0.23$ | $1.03 \pm 0.25$ |
| CADS | $7.67 \pm 0.23$ | $7.65 \pm 0.23$ | $7.59 \pm 0.21$ | $1.42 \pm 0.12$ | $1.37 \pm 0.13$ | $1.29 \pm 0.16$ |
| DPP | $7.61 \pm 0.25$ | $7.64 \pm 0.24$ | $7.61 \pm 0.24$ | $1.42 \pm 0.17$ | $1.36 \pm 0.15$ | $1.23 \pm 0.18$ |
| **Ours** | $\mathbf{7.66 \pm 0.22}$ | $\mathbf{7.65 \pm 0.24}$ | $\mathbf{7.63 \pm 0.23}$ | $\mathbf{1.49 \pm 0.11}$ | $\mathbf{1.43 \pm 0.12}$ | $\mathbf{1.36 \pm 0.14}$ |

This quantitative superiority is particularly evident in low-step scenarios. For a direct visual comparison, Figure 15 showcases the qualitative differences against all baselines at a fixed, low computational budget of NFE=20 for the prompt "A photo of a truck". The visual results corroborate our quantitative findings, highlighting that our method produces a significantly more diverse and visually coherent set of images compared to the baselines in efficient generation scenarios.

Having established our method's advantage over baselines, we also analyze its internal trade-off between computational cost and image fidelity in Figure 14. While our approach is effective even at NFE=10, the generations can exhibit minor artifacts like blurry edges and localized distortions. Quality progressively improves with a larger budget, with NFE=20 yielding sharper results and

NFE=40 producing the most natural and artifact-free images. Taken together, these results confirm that our method is superior to baselines at all tested budgets, and its output quality can be further enhanced with a higher NFE.

Figure 14: Visual analysis of our method's sensitivity to the NFE for the prompt "A cozy cabin in the snowy forest." A low NFE of 10 results in images with blurry edges and localized artifacts, while a higher NFE of 40 yields the most natural and artifact-free results.

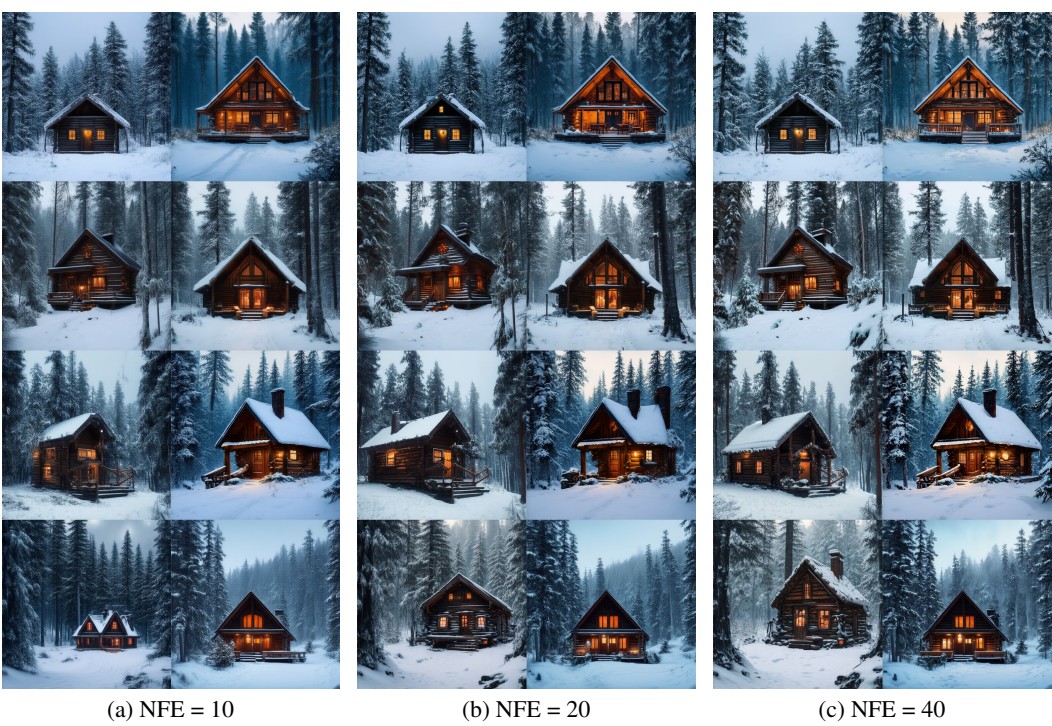

(a) NFE = 10          (b) NFE = 20          (c) NFE = 40

Figure 15: Visual comparison of all methods at a fixed low computational budget (NFE=20) for the prompt "A cozy cabin in the snowy forest". Our method generates a visibly more diverse set of images.

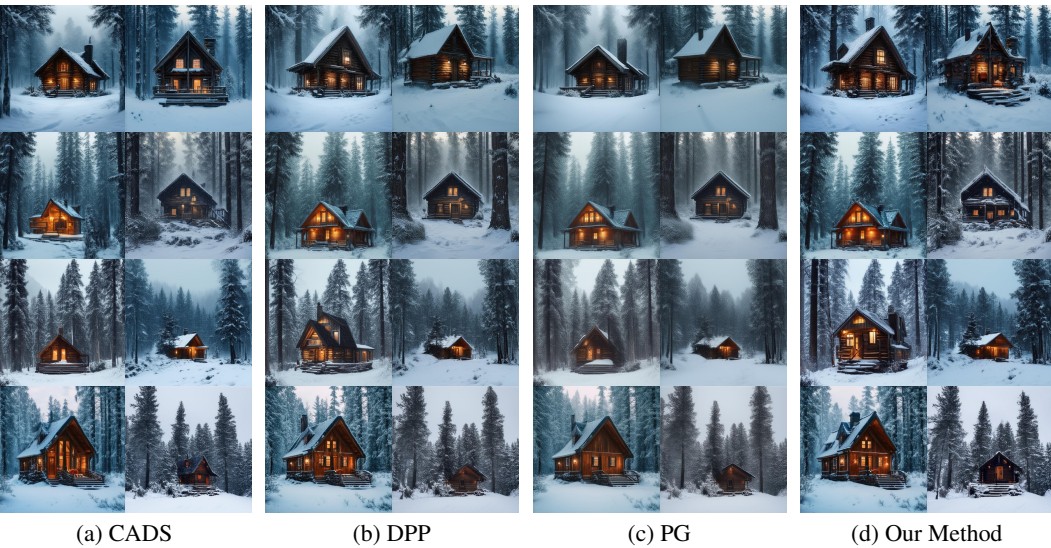

(a) CADS          (b) DPP          (c) PG          (d) Our Method

# F    ROBUST STUDIES

## F.1    NUMERICAL ROBUSTNESS: OUR EXTRAPOLATION VS. ALTERNATIVE INTEGRATORS

The final step of our framework utilizes a predictor to estimate the final vector. To evaluate the robustness of this component, we compared several methods, with the results detailed in Table 8. These methods include Euler, Midpoint, no predictor (w/o Predictor), and our method.

Table 15 reveals a clear trade-off between performance and stability. Completely removing the predictor, while yielding the lowest volatility, significantly sacrifices image quality, leading to notably worse performance. In contrast, the Euler and Midpoint methods show improved average performance, but at the cost of greater instability across multiple runs. Our method achieves the joint-best average performance across all metrics. More importantly, our method accomplishes this superior performance while maintaining low volatility, striking the optimal balance between high-quality output and high stability. It was therefore selected as our default configuration.

Table 15: Robustness and ablation study on the predictor component. Different predictors are used to estimate the final vector with the same NFE. Our default choice demonstrates superior performance.

| Predictor | CLIP↑ | Vendi (Pixel) ↑ | Vendi (Inception) ↑ | FID↓ | BRISQUE↓ |
|---|---|---|---|---|---|
| w/o Predictor | $27.77 \pm 0.16$ | $2.78 \pm 0.04$ | $5.37 \pm 0.11$ | $165.5 \pm 1.2$ | $24.9 \pm 1.5$ |
| Euler | $28.19 \pm 0.30$ | $\mathbf{2.87 \pm 0.16}$ | $5.52 \pm 0.28$ | $164.6 \pm 1.6$ | $21.9 \pm 1.8$ |
| Midpoint | $28.21 \pm 0.32$ | $2.86 \pm 0.12$ | $5.43 \pm 0.30$ | $164.4 \pm 1.6$ | $22.4 \pm 2.2$ |
| **Ours** | $\mathbf{28.26 \pm 0.22}$ | $2.86 \pm 0.05$ | $\mathbf{5.63 \pm 0.20}$ | $\mathbf{163.3 \pm 1.6}$ | $\mathbf{21.2 \pm 1.5}$ |

6 seeds; mean±95% CI. "w/o Predictor" directly uses the current vector without a correction step.

## F.2    CROSS-BACKBONE GENERALIZATION

To validate the portability of our OSCAR framework, we extended its application beyond FM-SD3.5 to two other prominent foundational models: SDXL-Turbo (Sauer et al., 2024) and SD1.5 (Rombach et al., 2022a). SDXL-Turbo is a well-known distilled model designed for high-speed, real-time generation, while SD1.5 is a widely-adopted and influential early latent diffusion model.

As detailed in Table 16, we first transferred the hyperparameters tuned on FM-SD3.5 directly to these new models without modification denoted "OSCAR (default params)". Notably, even this direct transfer demonstrates strong robustness: compared to the original baselines, this configuration shows slight improvements in diversity metrics while, critically, incurring no performance degradation in key image quality metrics.

Furthermore, OSCAR's comprehensive performance is further enhanced with simple, model-specific hyperparameter tuning. This strongly indicates that OSCAR is not an architecture-specific solution but rather a general and robust "plug-and-play" framework that can be easily ported to diverse diffusion models.

## F.3    ENCODER CHOICE ROBUSTNESS

To test the robustness of our framework and its dependency on different feature extractors, we conducted an ablation study as shown in Table 17. We replaced our default CLIP encoder with two other prevalent encoders, Inception and DINO, while keeping all other components and parameters unchanged.

The results clearly indicate that our framework is highly robust to the choice of encoder. All three variants demonstrate highly comparable performance across all key metrics; our metrics remained almost entirely unperturbed by the change in encoder. These findings prove that the efficacy of our framework does not stem from an over-reliance on a specific feature space. On the contrary, the framework exhibits strong universality, and its core mechanisms operate stably across different feature spaces. We selected CLIP as our default configuration because it showed a slight advantage in diversity metrics while maintaining highly competitive fidelity.

Table 16: Portability of our OSCAR framework across different foundational models. OSCAR consistently improves fidelity and alignment over the respective baselines for FM-SD3.5, SDXL-Turbo, and SD1.5. Further gains are achieved by tuning OSCAR's hyperparameters for each specific model. Higher is better for CLIP/Vendi; lower is better for FID/BRISQUE.

| Model | Variant | CLIP↑ | Vendi (Pixel) ↑ | Vendi (Inception) ↑ | FID↓ | BRISQUE↓ |
|---|---|---|---|---|---|---|
| **FM-SD3.5** | Baseline | $28.24 \pm 0.18$ | $2.45 \pm 0.13$ | $5.37 \pm 0.27$ | $164.4 \pm 1.8$ | $23.4 \pm 1.4$ |
| | **OSCAR** | $\mathbf{28.26 \pm 0.22}$ | $\mathbf{2.86 \pm 0.05}$ | $\mathbf{5.63 \pm 0.22}$ | $\mathbf{163.3 \pm 1.6}$ | $\mathbf{21.2 \pm 1.5}$ |
| **SDXL-Turbo** | Baseline | $30.98 \pm 0.15$ | $5.31 \pm 0.25$ | $4.24 \pm 0.13$ | $150.4 \pm 1.0$ | $24.6 \pm 0.3$ |
| | **OSCAR** (default params) | $30.77 \pm 0.24$ | $5.41 \pm 0.25$ | $4.29 \pm 0.23$ | $152.8 \pm 0.6$ | $25.1 \pm 1.0$ |
| | **OSCAR** (tuned params) | $\mathbf{30.94 \pm 0.22}$ | $\mathbf{5.48 \pm 0.34}$ | $\mathbf{4.42 \pm 0.17}$ | $\mathbf{151.6 \pm 1.3}$ | $\mathbf{25.0 \pm 0.9}$ |
| **SD1.5** | Baseline | $29.93 \pm 0.35$ | $2.37 \pm 0.12$ | $7.02 \pm 0.27$ | $174.7 \pm 1.9$ | $12.1 \pm 3.3$ |
| | **OSCAR** (default params) | $29.88 \pm 0.35$ | $2.45 \pm 0.12$ | $7.09 \pm 0.16$ | $175.1 \pm 1.6$ | $12.8 \pm 3.2$ |
| | **OSCAR** (tuned params) | $\mathbf{29.93 \pm 0.37}$ | $\mathbf{2.46 \pm 0.12}$ | $\mathbf{7.11 \pm 0.12}$ | $\mathbf{174.6 \pm 1.6}$ | $\mathbf{12.5 \pm 3.1}$ |

6 seeds; mean±95% CI. "default params" uses hyperparameters from FM-SD3.5; "tuned params" are optimized for each model.

Table 17: Ablation study on the feature encoder used within our framework. Our default model shows the best performance compared to other common encoders like Inception and DINO.

| Encoder | CLIP Score ↑ | Vendi (Pixel) ↑ | Vendi (Inception) ↑ | FID↓ | BRISQUE↓ |
|---|---|---|---|---|---|
| Inception | $28.32 \pm 0.22$ | $2.85 \pm 0.08$ | $5.60 \pm 0.22$ | $163.2 \pm 1.2$ | $21.7 \pm 1.6$ |
| DINO | $28.35 \pm 0.17$ | $2.82 \pm 0.05$ | $5.57 \pm 0.24$ | $164.2 \pm 2.0$ | $21.1 \pm 1.8$ |
| **CLIP (Ours)** | $\mathbf{28.26 \pm 0.22}$ | $\mathbf{2.86 \pm 0.05}$ | $\mathbf{5.63 \pm 0.20}$ | $\mathbf{163.3 \pm 1.6}$ | $\mathbf{21.2 \pm 1.5}$ |

6 seeds; mean±95% CI. All variants use the full OSCAR framework.

## F.4 NOISE ROBUSTNESS

### F.4.1 ROBUSTNESS TO NOISE SCHEDULE ($t_{gate}$)

We analyze the sensitivity of our method to the noise application schedule, controlled by the *Noise Gate* parameter. This parameter defines the time interval $[0.05, t_{gate}]$ during which stochastic noise is active. The results, presented quantitatively in Table 18 and visually in Figure 16, demonstrate our method's remarkable robustness. Across a wide range of end times, from $t_{gate} = 0.5$ to $t_{gate} = 0.1$, all quality and diversity metrics remain exceptionally stable, without any sign of performance collapse. This indicates that our method is not sensitive to the precise duration of noise injection, making it easy to use without extensive hyperparameter tuning.

Table 18: Robustness analysis for the noise injection schedule, controlled by the end time $t_{gate}$ of the noise gate $[0.05, t_{gate}]$. The performance across all metrics remains highly stable, demonstrating that our method is not sensitive to this hyperparameter.

| Noise Gate | 1-MS-SSIM % ↑ | Vendi Score Pixel ↑ | Vendi Score Inception ↑ | FID ↓ | Brisque ↓ |
|---|---|---|---|---|---|
| 0.50 | $87.92 \pm 2.11$ | $2.32 \pm 0.15$ | $5.12 \pm 0.07$ | $131.31 \pm 1.31$ | $25.50 \pm 1.71$ |
| 0.40 | $87.92 \pm 1.38$ | $2.36 \pm 0.12$ | $5.21 \pm 0.17$ | $132.31 \pm 2.10$ | $23.78 \pm 1.66$ |
| 0.30 | $88.73 \pm 1.61$ | $2.36 \pm 0.17$ | $5.21 \pm 0.11$ | $132.80 \pm 1.96$ | $23.87 \pm 2.22$ |
| 0.20 | $88.53 \pm 1.44$ | $2.37 \pm 0.18$ | $5.24 \pm 0.12$ | $131.33 \pm 1.15$ | $23.71 \pm 1.87$ |
| 0.10 | $87.97 \pm 1.49$ | $2.34 \pm 0.14$ | $5.16 \pm 0.07$ | $130.93 \pm 1.42$ | $23.54 \pm 2.05$ |

### F.4.2 ROBUSTNESS TO NOISE SCALE ($s$)

Similarly, we evaluate the method's robustness to the magnitude of the injected noise, controlled by the *Noise Scale* parameter. As shown quantitatively in Table 19 and visually in Figure 17, our method's performance remains highly consistent across a wide range of noise scales, from $s = 0.25$ to $s = 5.0$. While an excessively large scale ($s = 10.0$) begins to degrade perceptual quality, the overall stability of metrics like FID and Vendi Score Inception across nearly two orders of magnitude highlights the robustness of our approach. The results suggest a broad optimal operating region around $s = 2.0$, where multiple metrics are jointly optimized.

Figure 16: Visual comparison of generated images across different noise gates $[0.05, t_{gate}]$.

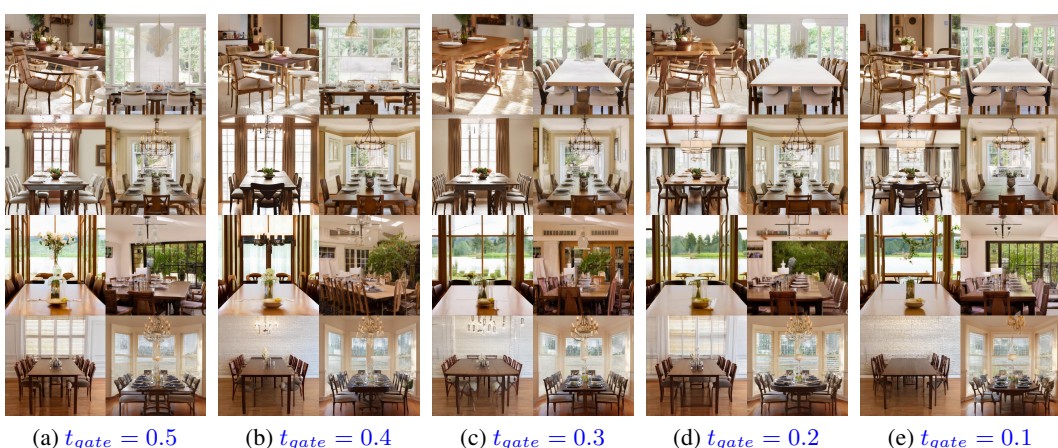

(a) $t_{gate} = 0.5$  (b) $t_{gate} = 0.4$  (c) $t_{gate} = 0.3$  (d) $t_{gate} = 0.2$  (e) $t_{gate} = 0.1$

Table 19: Robustness analysis for the noise scale ($s$). Performance is highly stable for $s \in [0.25, 5.0]$, with a clear optimal region around $s = 2.0$ where multiple quality and diversity metrics are maximized. An excessive scale of $s = 10.0$ leads to a degradation in perceptual quality.

| Noise Scale | CLIP Score ↑ | Vendi Score Pixel ↑ | Vendi Score Inception ↑ | FID ↓ | Brisque ↓ |
|---|---|---|---|---|---|
| 0.25 | $14.69 \pm 2.63$ | $2.05 \pm 0.15$ | $3.70 \pm 0.17$ | $134.54 \pm 2.77$ | $23.59 \pm 4.64$ |
| 0.5 | $14.11 \pm 1.43$ | $2.08 \pm 0.19$ | $3.74 \pm 0.16$ | $135.24 \pm 1.45$ | $24.82 \pm 4.88$ |
| 1.0 | $15.93 \pm 0.91$ | $2.07 \pm 0.16$ | $3.81 \pm 0.13$ | $134.14 \pm 1.77$ | $22.64 \pm 4.54$ |
| 2.0 | $16.11 \pm 0.74$ | $2.12 \pm 0.19$ | $3.84 \pm 0.14$ | $134.38 \pm 2.83$ | $22.18 \pm 3.28$ |
| 5.0 | $15.39 \pm 0.83$ | $2.06 \pm 0.18$ | $3.81 \pm 0.06$ | $133.71 \pm 2.70$ | $24.06 \pm 4.95$ |
| 10.0 | $12.29 \pm 0.60$ | $2.01 \pm 0.16$ | $3.58 \pm 0.22$ | $134.55 \pm 1.75$ | $28.51 \pm 5.55$ |

### F.4.3 ROBUSTNESS TO NOISE SCHEDULE (COS2 VS T1MT)

We compare two time schedules for the exploration term used in our sampler: (i) a cosine-squared schedule, $\gamma(t) = \cos^2(\pi s(t))$, and (ii) a parabolic schedule, $\gamma(t) = s(t)(1 - s(t))$, where $s(t) \in [0, 1]$ is the normalized time. Both profiles are bell-shaped; `cos2` ramps up smoothly from zero, peaks at mid-trajectory, and then decays smoothly, while `t1mt` follows an inverted-parabola rise–fall pattern. Unless otherwise stated, all settings are kept identical across schedules.

**Quantitative results.** Table 20 summarizes an ablation at a fixed guidance of CFG = 3.0. We observe a small trade-off: `cos2` yields slightly higher CLIP Score and Vendi Score Pixel, which means more low-level variation, whereas `t1mt` marginally improves Vendi Score Inception and achieves lower FID/Brisque. Both schedules are viable; `cos2` mildly favors exploration, while `t1mt` mildly favors fidelity.

Table 20: Ablation study on our fidelity safeguards at a fixed guidance of $CFG = 3.0$. The results confirm that our method is robust to different $\beta_t$ schedules.

| $\beta(t)$ | CLIP Score ↑ | Vendi Score Pixel ↑ | Vendi Score Inception ↑ | FID ↓ | Brisque ↓ |
|---|---|---|---|---|---|
| cos2 | $26.61 \pm 0.07$ | $2.66 \pm 0.21$ | $4.81 \pm 0.09$ | $126.52 \pm 0.60$ | $20.26 \pm 2.07$ |
| t1mt | $25.26 \pm 0.30$ | $2.59 \pm 0.02$ | $5.05 \pm 0.01$ | $125.45 \pm 2.06$ | $18.53 \pm 6.61$ |

**Qualitative results.** Figure 18 provides side-by-side visualizations at the same guidance and NFE. Consistent with the quantitative trends, `cos2` tends to distribute samples more broadly (more visible pixel-level variation), while `t1mt` produces slightly cleaner images with comparable semantic coverage.

Figure 17: Visual comparison of generated images across different noise scales ($s$).


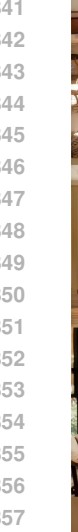
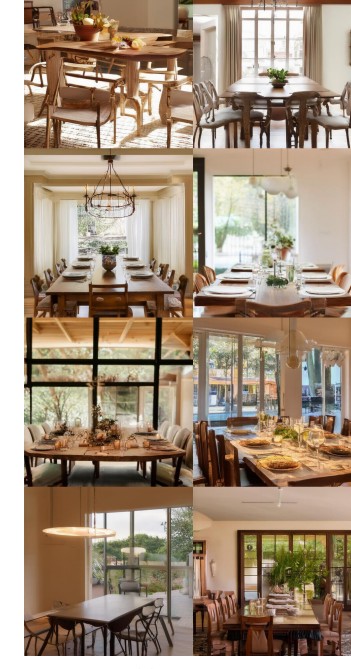
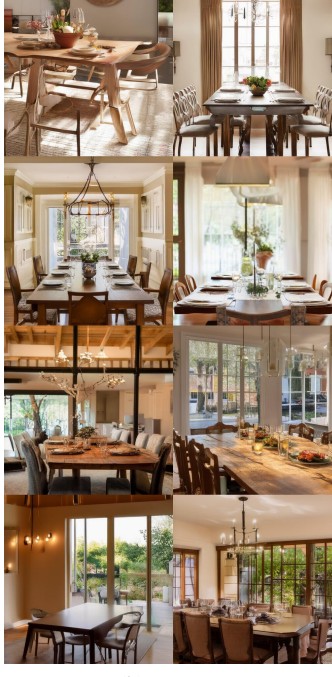
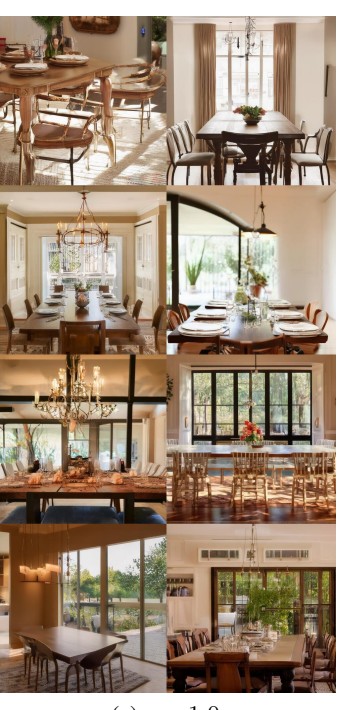

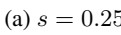

(a) $s = 0.25$      (b) $s = 0.5$      (c) $s = 1.0$

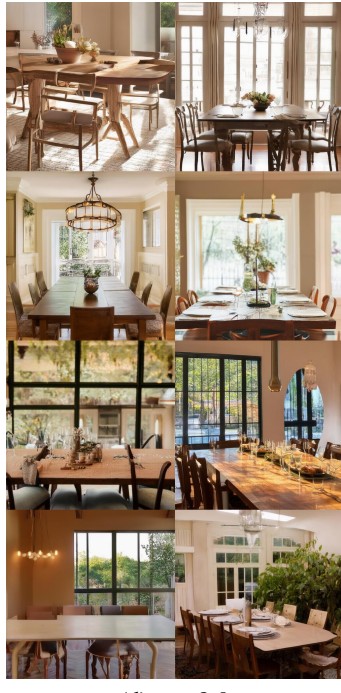
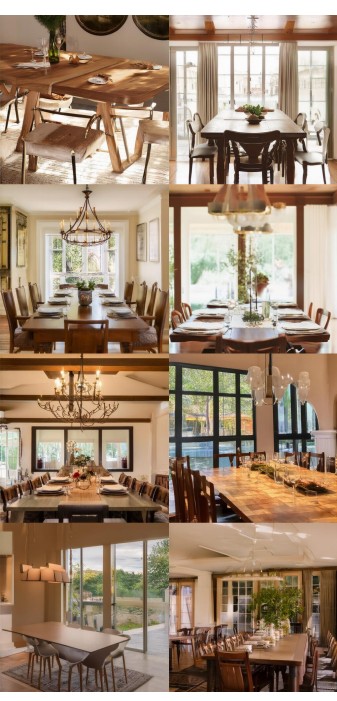
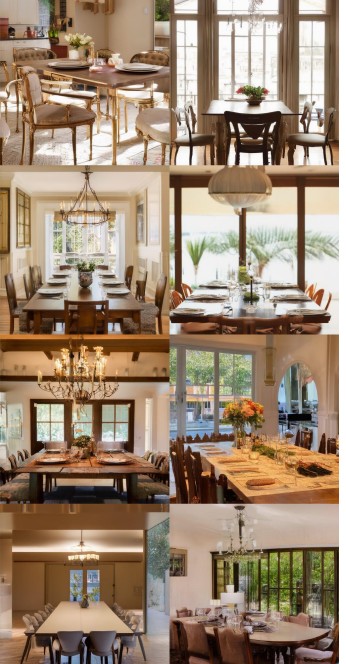

(d) $s = 2.0$      (e) $s = 5.0$      (f) $s = 10.0$

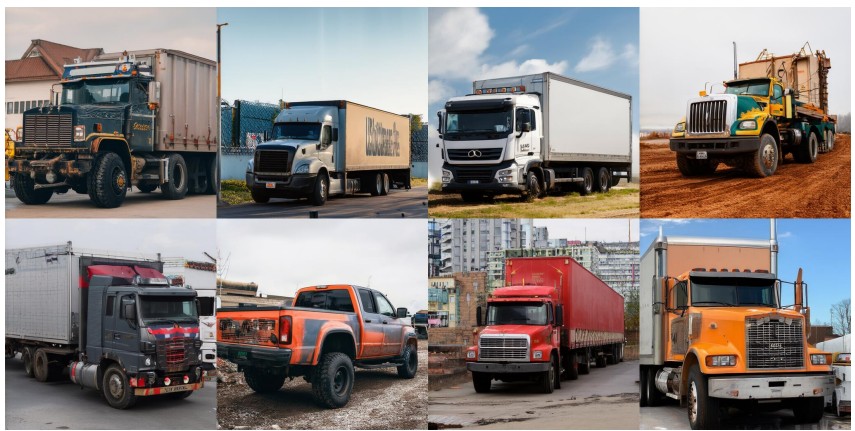

(a) `cos2`

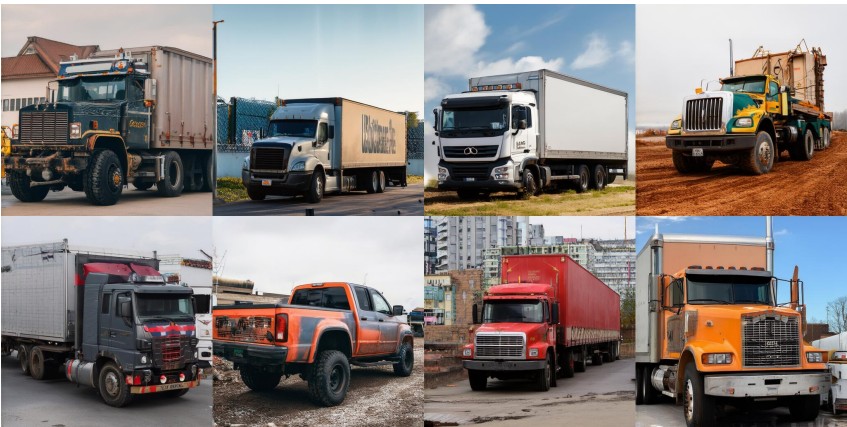

(b) `t1mt`

Figure 18: Qualitative comparison of the two time schedules at the same guidance and NFE.

## G   MORE VISUAL RESULTS

This section provides additional qualitative results to supplement our main findings. We first present extended comparisons for the 512×512 class-conditional generation task on COCO concepts, as shown in Figures 20 and 21. We further showcase visual results from our DIM and CIM evaluations on both coarse and dense prompts, as illustrated in Figures 22 and 23. These comparisons visually demonstrate the effectiveness of our method in enhancing sample diversity compared to baselines.

Figure 19: Comprehensive visual ablation of our fidelity safeguards. Each column corresponds to one variant: (a) w/o OP, (b) w/o RR, (c) w/o OP & RR, and (d) full OSCAR. Each row shows results for a different prompt, from top to bottom: "A photo of a bowl", "A photo of a truck".

(a) w/o OP        (b) w/o RR        (c) w/o OP & RR        (d) Our Method

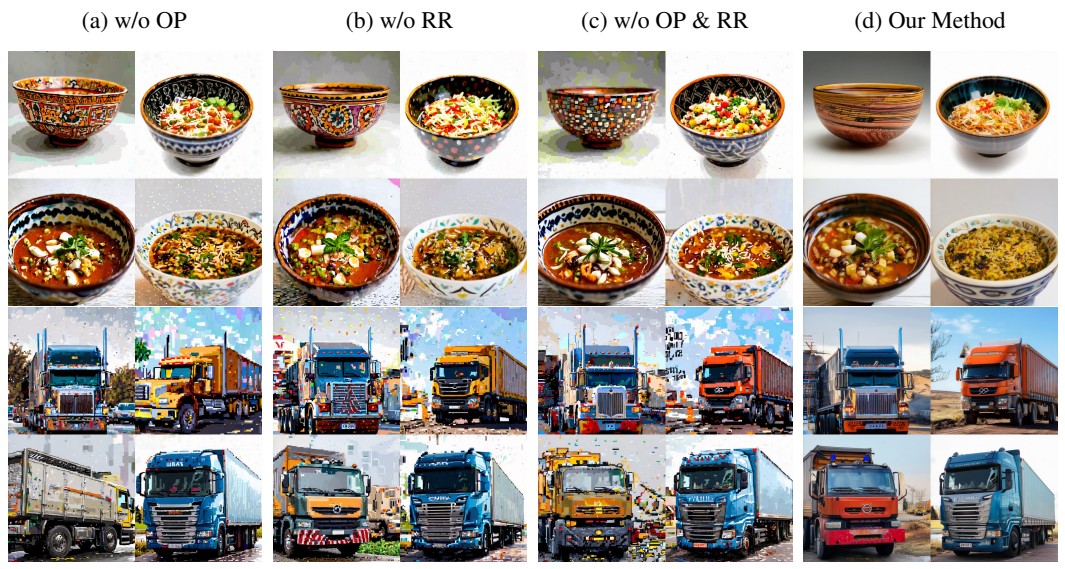

Figure 20: Comprehensive visual comparison across all methods. Each column corresponds to a single method: (a) CADS, (b) DPP, (c) PG, and (d) Our Method. Each row shows results for a different prompt, in the following order from top to bottom: "A photo of a bus", "A photo of a truck", and "A photo of a bicycle".

| (a) CADS | (b) DPP | (c) PG | (d) Our Method |

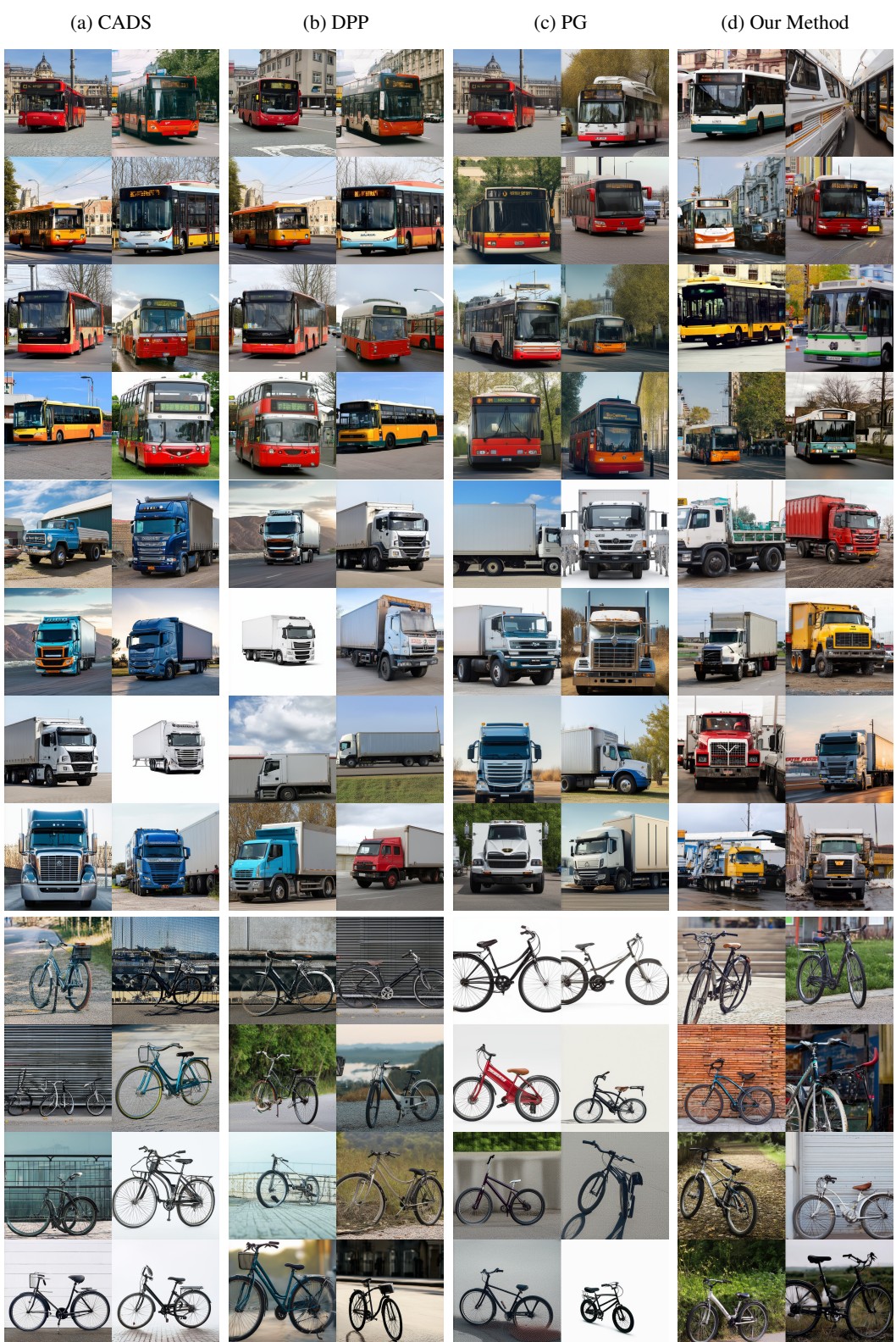

Figure 21: Comprehensive visual comparison across all methods. Each column corresponds to a single method: (a) CADS, (b) DPP, (c) PG, and (d) Our Method. Each row shows results for a different prompt, in the following order from top to bottom: "A photo of a potted plant", "A photo of a dog", and "A close-up photo of a dining table".

(a) CADS            (b) DPP            (c) PG            (d) Our Method

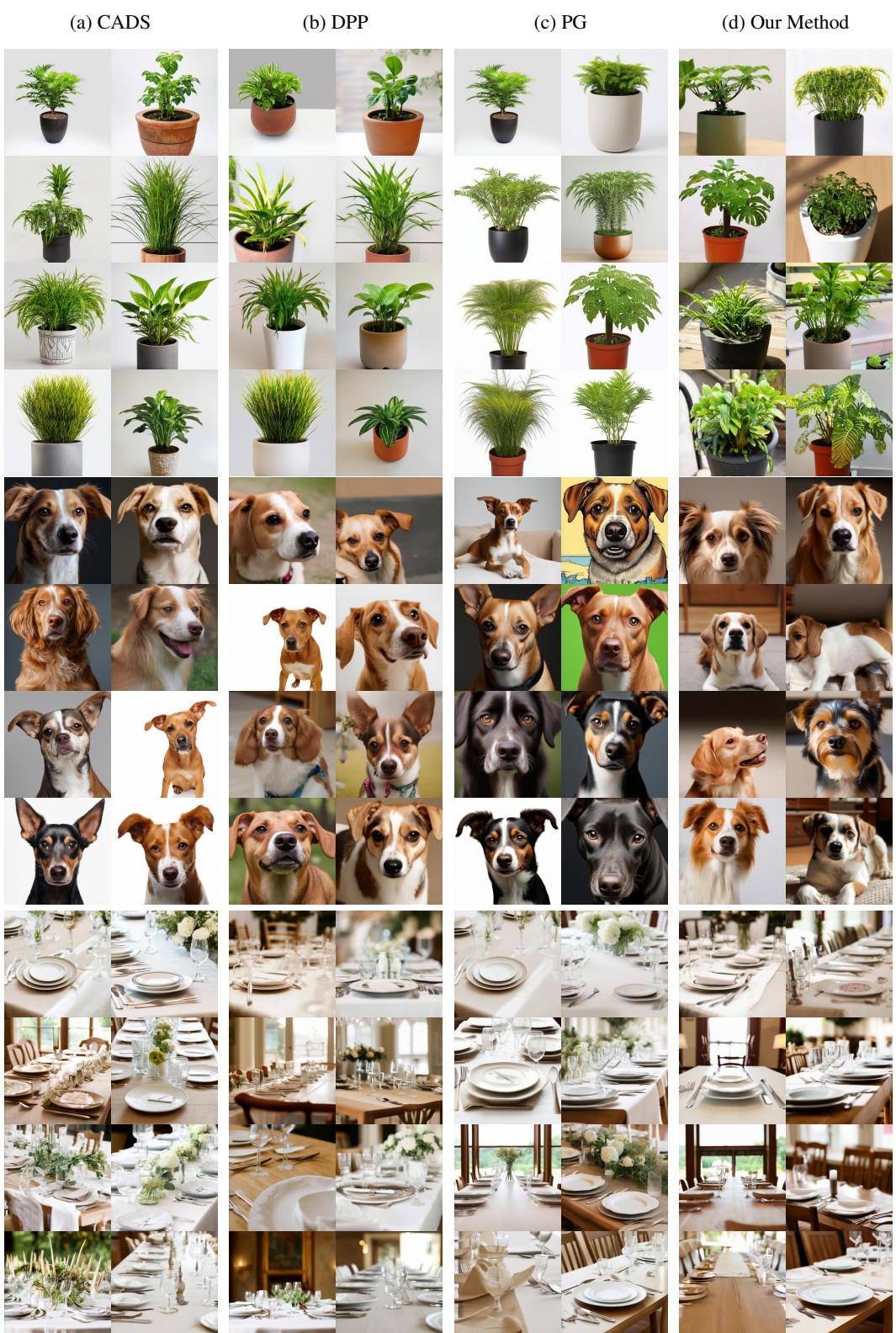

Figure 22: Qualitative comparison of conditional generalization (CIM) across all methods. These images were generated using *dense prompts*, where specific attributes were explicitly requested, such as "a blue bus on a city street" or "a single-decked bus drives down the street". This setting evaluates each model's ability to follow precise instructions.

(a) CADS  (b) DPP  (c) PG  (d) Our Method

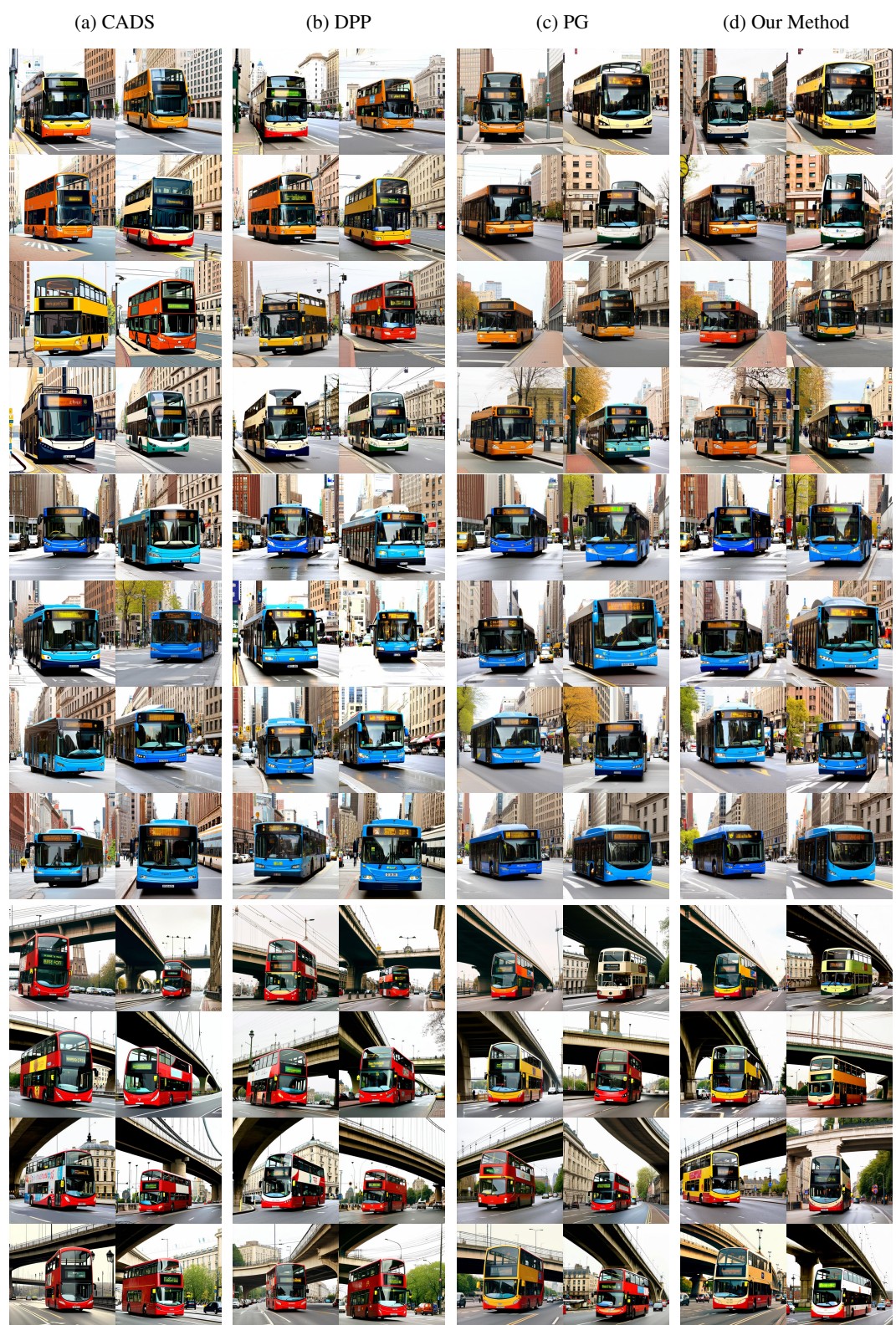

Figure 23: Qualitative comparison of default-mode diversity (DIM) across all methods. These images were generated using underspecified *coarse prompts*, such as "a bus by the side of a building" or "a bus stopping at a bus stop in a city". This experiment evaluates each model's ability to generate a diverse and balanced set of attributes spontaneously.

(a) CADS      (b) DPP      (c) PG      (d) Our Method

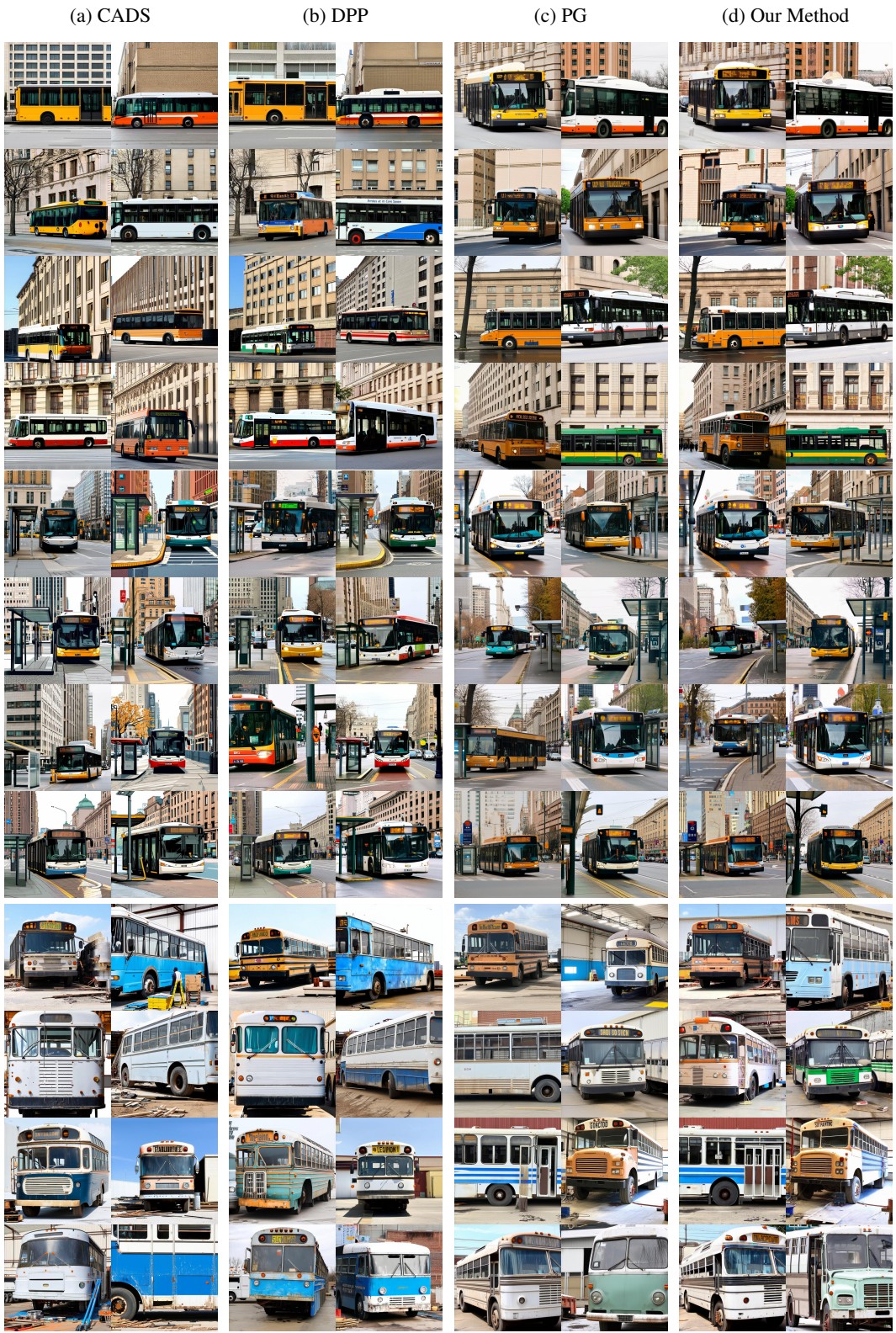

