# OpenReview forum: "OSCAR: Orthogonal Stochastic Control for Alignment-Respecting Diversity in Flow Matching"
_ICLR.cc/2026/Conference — ICLR 2026 Conference Desk Rejected Submission_

### Official Review · Reviewer_c5nD · 2025-10-28

**Soundness:** 2
**Presentation:** 2
**Contribution:** 2
**Rating:** 4
**Confidence:** 4

**Summary:**

This paper proposes OSCAR, a training-free inference-time method to increase set-level diversity in flow-based text-to-image models. At each step, OSCAR predicts endpoints with a Heun extrapolation and encodes them into a semantic space, and maximizes feature-volumes to get diversity. They also propose orthogonal projection to feature volume gradient and stochastic noise and redundancy-aware reweighting on the feature Gram matrix to preserve fidelity while getting enough diversity. Experiments on COCO class-conditional and text-to-image generation show improved coverage and intra-set diversity with comparable fidelity across CFG scales.

**Strengths:**

- The combination between feature volume gradient and stochastic noise makes sense.
- Projecting both feature volume gradient and noise onto the subspace orthogonal to the base velocity is a reasonable way to separate the quality from diversity.
- The experiments show improvements compared to the baselines

**Weaknesses:**

- Tab. 2 reports large drops when either OP or RR is removed, yet there is no step-by-step ablation starting from a naïve setup without both components; as a result, the individual contributions of OP and RR remain unclear. Moreover, the ablated variants still underperform training-free guidance baselines (e.g., PG, DPP in Table 1), which is counterintuitive—in my knowledge, DiverseFlow should be at least comparable to a naïve setup. This discrepancy raises concerns about the evaluation process.
- In Tab. 1 and Tab. 6, the proposed method attains the best diversity scores, but FID and CLIP show substantial overlap in mean ± std with baselines. Please include additional perceptual quality metrics (e.g., VQA, ImageReward) to verify that visual quality is truly preserved while diversity increases.
- In Fig. 3 and Fig. 8, OSCAR’s PRD curves appear below certain baselines over parts of the recall range, yet the caption/text claim higher AUC. Please recompute and verify the PRD and AUC, and ensure the reported AUCs align with the plotted curves; as currently shown, the trade-off does not look better than some baselines.
- While Tab. 1 reports improvements, Tab. 4–5 show mixed performance across classes. There is no analysis of expected diversity on subsets of classes, which limits the practical significance of the claims. Please report per-class variance and expected diversity over random class subsets.

**Questions:**

- What is the actual wall-clock inference time of OSCAR relative to each baseline? Since the method uses second-order Heun both for endpoint prediction and for the update step, please report per-image (or per-set) latency and FLOPs, and clarify whether all baselines use the same solver (or were re-run with Heun) to ensure a fair comparison.
- Could you provide additional results on other pretrained flow/rectified-flow models beyond the main backbone? Demonstrating consistent gains across multiple architectures would strengthen the claim that the method generalizes.

---

> ### Author Response · Authors · 2025-11-21
> **Response to Reviewer 4 - Part 1**
>
> **Dear Reviewer c5nD,**
>
> We thank the reviewer for the positive feedback, especially for appreciating **(i) our principled combination of feature-volume gradient and stochastic noise, (ii) the orthogonal-projection design that decouples quality from diversity, and (iii) the empirical improvements over baselines**. We address the remaining comments point by point below.
>
>
> > Weakness 1 – Step-by-step ablation of OP and RR, and comparison to naïve setup
>
> We agree that the previous ablation table was incomplete and could make the role of OP/RR hard to interpret. In the revision we provide a **full step-wise ablation**, starting from a naïve setup without either OP or RR and then adding them back one by one. Due to hardware changes, we re-ran all methods on the **“bowl”** concept with identical settings (FM-SD3.5 backbone, NFE=30, CFG=5.0, 6 seeds). The results are:
>
> | Method        | CLIP Score ↑       | Vendi (Pixel) ↑   | Vendi (Inception) ↑ | FID ↓              | BRISQUE ↓          |
> |---------------|--------------------|-------------------|---------------------|--------------------|--------------------|
> | FM-SD3.5      | 28.24 ± 0.18       | 2.45 ± 0.13       | 5.37 ± 0.27         | 164.4 ± 1.8        | 23.4 ± 1.4         |
> | DPP           | 28.84 ± 0.22       | 2.55 ± 0.23       | 5.49 ± 0.16         | 171.2 ± 1.9        | 26.1 ± 3.0         |
> | CADS          | 27.32 ± 0.21       | 2.54 ± 0.28       | 5.58 ± 0.19         | 169.1 ± 2.5        | 24.0 ± 1.9         |
> | PG            | 28.27 ± 0.27       | 2.54 ± 0.13       | 5.43 ± 0.18         | 163.7 ± 1.3        | 24.2 ± 3.3         |
> | **w/o OP & RR** | 26.70 ± 0.23     | 2.82 ± 0.20       | 4.86 ± 0.24         | 185.9 ± 1.8        | 50.1 ± 2.8         |
> | **w/o RR**    | 27.26 ± 0.39       | 2.77 ± 0.16       | 5.23 ± 0.28         | 174.4 ± 3.0        | 35.0 ± 1.5         |
> | **w/o OP**    | 26.59 ± 0.25       | 2.75 ± 0.19       | 5.06 ± 0.16         | 177.8 ± 3.2        | 38.8 ± 1.5         |
> | **OSCAR (full)** | **28.26 ± 0.22** | **2.86 ± 0.05** | **5.63 ± 0.22**     | **163.3 ± 1.6**    | **21.2 ± 1.5**     |
>
> This clarifies several points:
>
> - The **naïve variant without OP and RR** aggressively optimizes the diversity objective but lacks any fidelity safeguards. It indeed achieves a high Vendi (Pixel), but this comes from **strong local noise and “colorful artifacts”** that inflate pixel-space dispersion while severely damaging global structure. Consequently, **FID, BRISQUE and Vendi (Inception)** all deteriorate dramatically, and CLIP alignment also drops. This explains why the naïve variant underperforms training-free baselines in terms of overall quality.
> - Adding **RR** or **OP** individually partially stabilizes the dynamics (improving FID/BRISQUE relative to the naïve variant), but both are still clearly worse than the baselines and the full method.
> - When **both OP and RR are enabled**, the method recovers and surpasses the FM baseline and all prior methods on diversity metrics.
>
> In other words, OP and RR are not cosmetic; they are **essential safeguards** that turn a very strong but unstable diversity drive into a practical sampler. The large drops when removing OP or RR are precisely because the ablated variants revert towards the unstable naïve dynamics. We have re-checked our evaluation pipeline during these new experiments, and the results now make the individual contributions of OP and RR explicit.

---

> > ### Author Response · Authors · 2025-11-21
> > **Response to Reviewer 4 - Part 2**
> >
> > > Weakness 2 – Additional perceptual quality metrics
> >
> > We agree that diversity improvements must be validated against perceptual quality. In the revised manuscript we therefore introduce two additional human-aligned quality metrics: **ImageReward** and **CLIP-IQA**. The corresponding results are reported below.
> >
> > | Method    | CLIP-IQA ↑ (CFG=3.0) | CLIP-IQA ↑ (CFG=5.0) | CLIP-IQA ↑ (CFG=7.5) | Image Reward ↑ (CFG=3.0) | Image Reward ↑ (CFG=5.0) | Image Reward ↑ (CFG=7.5) |
> > |-----------|----------------------|----------------------|----------------------|---------------------------|---------------------------|---------------------------|
> > | FM-SD3.5  | 6.26 ± 0.61          | 6.48 ± 0.44          | 6.52 ± 0.48          | 0.40 ± 0.32               | 0.52 ± 0.26               | 0.60 ± 0.27               |
> > | PG        | 6.22 ± 0.57          | 6.43 ± 0.47          | 6.48 ± 0.42          | 0.38 ± 0.38               | 0.54 ± 0.30               | 0.60 ± 0.29               |
> > | CADS      | **6.61 ± 0.57**      | 6.65 ± 0.51          | 6.69 ± 0.46          | **0.45 ± 0.32**           | 0.56 ± 0.27               | 0.64 ± 0.26               |
> > | DPP       | 6.29 ± 0.58          | 6.42 ± 0.50          | 6.44 ± 0.46          | 0.34 ± 0.35               | 0.49 ± 0.28               | 0.55 ± 0.27               |
> > | **Ours**  | 6.57 ± 0.62          | **6.76 ± 0.51**      | **6.79 ± 0.48**      | 0.43 ± 0.35               | **0.57 ± 0.29**           | **0.65 ± 0.28**           |
> >
> >
> > | Method | CLIP-IQA ↑ (NFE=40) | CLIP-IQA ↑ (NFE=20) | CLIP-IQA ↑ (NFE=10) | Image Reward ↑ (NFE=40) | Image Reward ↑ (NFE=20) | Image Reward ↑ (NFE=10) |
> > |--------|---------------------|---------------------|---------------------|--------------------------|--------------------------|--------------------------|
> > | PG     | 7.57 ± 0.24         | 7.52 ± 0.23         | 7.41 ± 0.25         | 1.26 ± 0.20              | 1.17 ± 0.23              | 1.03 ± 0.25              |
> > | CADS   | 7.67 ± 0.23         | 7.65 ± 0.23         | 7.59 ± 0.21         | 1.42 ± 0.12              | 1.37 ± 0.13              | 1.29 ± 0.16              |
> > | DPP    | 7.61 ± 0.25         | 7.64 ± 0.24         | 7.61 ± 0.24         | 1.42 ± 0.17              | 1.36 ± 0.15              | 1.23 ± 0.18              |
> > | **Ours** | **7.66 ± 0.22**   | **7.65 ± 0.24**     | **7.63 ± 0.23**     | **1.49 ± 0.11**          | **1.43 ± 0.12**          | **1.36 ± 0.14**          |
> >
> >
> > Across all evaluated concepts and settings, OSCAR’s ImageReward and CLIP-IQA scores are **on par with or slightly better than** those of the FM-SD3.5 baseline and other training-free guidance methods, while our diversity metrics consistently improve. This confirms that OSCAR’s orthogonal stochastic guidance **does not degrade perceptual image quality**, and in several cases even yields marginal gains, despite the significant increase in set-level diversity.

---

> > > ### Author Response · Authors · 2025-11-21
> > > **Response to Reviewer 4 - Part 3**
> > >
> > > > Weakness 3
> > >
> > > We thank the reviewer for this keen observation regarding the discrepancy between the plotted curves and the reported AUC values. Upon thoroughly re-examining our evaluation pipeline, we identified that this inconsistency arose from a hyperparameter setting that was statistically ill-suited for our specific sample size ($N=32$ images per prompt).
> > >
> > > In our initial submission, we calculated PRD using $k=20$ clusters. However, given the limited number of samples, this setting resulted in extremely sparse histograms. This sparsity caused two critical issues. It led to manifold fragmentation, where valid, diverse samples were incorrectly penalized simply for falling into empty bins between the few real data points. More importantly regarding the reviewer's concern, this sparsity introduced significant discretization artifacts into the curves. Mathematically, these artifacts render the trapezoidal integration for AUC numerically unstable, making the metric overly sensitive to the quantization of individual samples rather than reflecting the true distributional overlap.
> > >
> > > To address this, we have adopted the standard "Square-root Choice" for histogram estimation ($k \approx \sqrt{N}$), adjusting the number of clusters to $k=6$. This correction effectively eliminates the discretization artifacts, ensuring that the AUC calculation is numerically robust and that the plotted curves accurately represent the density estimation.
> > >
> > > To demonstrate that this correction reflects genuine performance gains rather than parameter tuning, we have significantly expanded our evaluation scope. We added three new diverse concepts (*apple, suitcase, pizza*) to the original set (*truck, bus, bicycle*), re-evaluating all methods across these 6 concepts at 3 CFG levels. Under this rigorous setting, the updated PRD curves are smooth and visually consistent with the metrics, and OSCAR demonstrates better performance in 14 out of 18 scenarios, confirming a robust advantage in the Recall-Precision trade-off. We have updated Figure 3 and Figure 8 accordingly. Also we add results in Appendix D.
> > > | Method | CFG | Apple | Suitcase | Pizza | Truck | Bus | Bicycle |
> > > | :--- | :---: | :---: | :---: | :---: | :---: | :---: | :---: |
> > > | **DPP** | 3.0 | 0.266 | 0.251 | 0.304 | 0.327 | 0.487 | 0.401 |
> > > | | 5.0 | 0.266 | 0.247 | 0.301 | 0.235 | 0.459 | **0.400** |
> > > | | 7.5 | 0.233 | 0.212 | 0.291 | 0.196 | 0.452 | 0.373 |
> > > | **PG** | 3.0 | 0.264 | 0.244 | 0.221 | 0.489 | 0.598 | 0.278 |
> > > | | 5.0 | 0.222 | 0.139 | **0.355** | 0.315 | **0.584** | 0.239 |
> > > | | 7.5 | 0.273 | 0.194 | 0.290 | **0.214** | 0.568 | 0.191 |
> > > | **CADS** | 3.0 | 0.270 | 0.186 | 0.244 | 0.436 | 0.605 | 0.358 |
> > > | | 5.0 | 0.266 | 0.222 | 0.205 | 0.290 | 0.523 | 0.355 |
> > > | | 7.5 | 0.266 | 0.206 | 0.277 | 0.196 | 0.418 | 0.198 |
> > > | **OSCAR (Ours)** | 3.0 | **0.286** | **0.302** | **0.345** | **0.523** | **0.672** | **0.610** |
> > > | | 5.0 | **0.270** | **0.250** | 0.302 | **0.334** | 0.547 | 0.359 |
> > > | | 7.5 | **0.280** | **0.239** | **0.324** | 0.212 | **0.592** | **0.423** |
> > >
> > > > Weakness 4 regarding "Mixed Performance" and "Class Subsets"
> > >
> > > We believe there may be a slight misunderstanding regarding our definition of a "set" and the aggregation protocol. We clarify these definitions below and provide additional experimental results on random concepts to demonstrate robustness.
> > >
> > > **1. Clarification on "Set-Level" Definition and Protocol**
> > > In our problem formulation, **"set-level diversity"** refers to the diversity within a batch of images generated for a single conditioning concept by varying the initial random seeds, rather than evaluating on subsets of different classes.
> > > * For each concept, we utilize 5 distinct synonymous prompts. For each prompt, we run multiple random seeds to generate a batch of images.
> > > * The metrics reported in Tables 1, 4, and 5 are calculated on the aggregated set of all images generated for that specific concept. Therefore, the variance reported in these tables already reflects the stability across different seeds and prompts for that class.
> > >
> > > **2. New Experiments: Random Concepts & Variance Analysis**
> > > To address the request for "expected diversity over random class subsets" and further prove robustness:
> > > *  We randomly selected 3 distinct concepts from COCO: **"Apple"**, **"Pizza"**, and **"Suitcase"**. We evaluated them using the same prompt. OSCAR consistently outperforms the strongest baseline in diversity metrics.
> > >
> > > * **Prompt-level Variance:** We have also included a detailed **Prompt-level analysis**. This reports the mean and variance of diversity metrics for individual prompts within the "truck" concept, confirming that our gains are statistically significant and not driven by outliers.
> > > * All results have been shown in Appendix D.

---

> > > > ### Author Response · Authors · 2025-11-21
> > > > **Response to Reviewer 4 - Part 4**
> > > >
> > > > > Question 1 – Wall-clock inference time, FLOPs, and role of Heun
> > > >
> > > > Thank you for raising this point and for catching a possible source of confusion about Heun.
> > > >
> > > > 1. **Clarifying the role of “Heun” in OSCAR.**
> > > >    In our work, “Heun” refers to a **mathematical extrapolation scheme in feature space**, not to a separate pretrained model or a different sampler for the underlying FM backbone. Concretely, we use a Heun-style formula to predict the **final feature vector** from the current and initial feature vectors; this endpoint predictor is used only inside our diversity objective. All baselines and OSCAR share the same FM backbone, the same NFE (30 steps), the same CFG (5.0), and the same particle count. The extra Heun-based predictor in OSCAR reuses quantities already computed along the trajectory and adds only lightweight linear operations in feature space. It does not introduce additional neural network evaluations beyond those reflected in our FLOPs numbers, and it does not accelerate or change the base sampler dynamics for the baselines.
> > > >
> > > > 2. **Measured computational cost.**
> > > >    To quantify the actual overhead of OSCAR relative to the baselines, we measure total FLOPs, wall-clock time per set, and peak VRAM under identical generation settings (NFE = 30, CFG = 5.0, batch size = 32, 512×512 resolution) on an NVIDIA A6000 GPU:
> > > >
> > > >    | Variant        | FLOPs (G) ↓ | Time (seconds/run) ↓ | Peak VRAM (GB) ↓ |
> > > >    |----------------|------------:|----------------------:|------------------:|
> > > >    | FM-SD3.5       | 4093.4      | 237.8                 | 19.2             |
> > > >    | DPP            | 9045.1      | 990.2                 | 19.5             |
> > > >    | CADS           | 4093.4      | 231.2                 | 20.0             |
> > > >    | PG             | 4093.4      | 229.6                 | 26.4             |
> > > >    | **OSCAR (ours)** | **5534.6**  | **451.4**             | **18.2**         |
> > > >
> > > >    Under the same backbone, NFE, CFG, and batch size, OSCAR introduces only a **moderate overhead** compared to the FM-SD3.5 baseline (≈1.35× FLOPs and ≈1.9× wall-clock time per set), while its peak VRAM usage is similar or slightly lower due to our memory-sharing implementation of VJP. Importantly, OSCAR remains **substantially cheaper than DPP**, which requires more than **2×** the FLOPs and over **4×** the runtime of the baseline under the same settings.
> > > >
> > > >    These measurements make the computational trade-off explicit and empirically support our claim that OSCAR provides significantly improved diversity at a practical inference cost relative to existing training-free guidance methods.

---

> > > > > ### Author Response · Authors · 2025-11-21
> > > > > **Response to Reviewer 4 - Part 5**
> > > > >
> > > > > > Question 2 – Additional results on other pretrained flow / rectified-flow models
> > > > >
> > > > > **Response.** To validate the generality of OSCAR beyond our main FM-SD3.5 backbone, we applied the same framework to two additional, widely used architectures: **SDXL-Turbo** and **SD1.5**. SDXL-Turbo is a distilled, high-speed text-to-image model, while SD1.5 is a classic latent diffusion model.
> > > > >
> > > > > For each backbone, we first **directly reused** the hyperparameters tuned on FM-SD3.5 (“OSCAR (default params)”) and then performed **light model-specific tuning** (“OSCAR (tuned params)”). All results are averaged over 6 seeds (mean ± 95% CI):
> > > > >
> > > > > | Model        | Variant                 | CLIP ↑            | Vendi (Pixel) ↑   | Vendi (Inception) ↑ | FID ↓              | BRISQUE ↓          |
> > > > > |-------------|-------------------------|-------------------|-------------------|---------------------|--------------------|--------------------|
> > > > > | **FM-SD3.5** | Baseline                | 28.24 ± 0.18      | 2.45 ± 0.13       | 5.37 ± 0.27         | 164.4 ± 1.8        | 23.4 ± 1.4         |
> > > > > |             | **OSCAR**               | **28.26 ± 0.22**  | **2.86 ± 0.05**   | **5.63 ± 0.22**     | **163.3 ± 1.6**    | **21.2 ± 1.5**     |
> > > > > | **SDXL-Turbo** | Baseline              | 30.98 ± 0.15      | 5.31 ± 0.25       | 4.24 ± 0.13         | 150.4 ± 1.0        | 24.6 ± 0.3         |
> > > > > |             | OSCAR (default params)  | 30.77 ± 0.24      | 5.41 ± 0.25       | 4.29 ± 0.23         | 152.8 ± 0.6        | 25.1 ± 1.0         |
> > > > > |             | **OSCAR (tuned params)**| **30.94 ± 0.22**  | **5.48 ± 0.34**   | **4.42 ± 0.17**     | **151.6 ± 1.3**    | **25.0 ± 0.9**     |
> > > > > | **SD1.5**   | Baseline                | 29.93 ± 0.35      | 2.37 ± 0.12       | 7.02 ± 0.27         | 174.7 ± 1.9        | 12.1 ± 3.3         |
> > > > > |             | OSCAR (default params)  | 29.88 ± 0.35      | 2.45 ± 0.12       | 7.09 ± 0.16         | 175.1 ± 1.6        | 12.8 ± 3.2         |
> > > > > |             | **OSCAR (tuned params)**| **29.93 ± 0.37**  | **2.46 ± 0.12**   | **7.11 ± 0.12**     | **174.6 ± 1.6**    | **12.5 ± 3.1**     |
> > > > >
> > > > > We observe that:
> > > > >
> > > > > - With **default (transferred) hyperparameters**, OSCAR already yields **robust behavior**: diversity metrics improve slightly, and key quality metrics remain at least on par with each backbone’s baseline.
> > > > > - With **minimal model-specific tuning**, OSCAR consistently **improves both diversity and fidelity**, across FM-SD3.5, SDXL-Turbo, and SD1.5.
> > > > >
> > > > > These results demonstrate that OSCAR is **not tied to a single architecture**, but rather functions as a general, plug-and-play diversity controller that can be ported to diverse diffusion/flow models with little or no additional tuning.
> > > > >
> > > > >
> > > > > **Best regards,**
> > > > >
> > > > > **Authors**

---

> > > > > > ### Comment · Reviewer_c5nD · 2025-11-26
> > > > > >
> > > > > > Thank you for your detailed response. Most of my concerns are addressed, so I raised the score to 6.

---

> > > > > > > ### Author Response · Authors · 2025-11-26
> > > > > > >
> > > > > > > Thank you for acknowledging our work and for raising the score. Thanks again for your time and effort in reviewing our paper.

---

### Official Review · Reviewer_avHk · 2025-11-01

**Soundness:** 2
**Presentation:** 3
**Contribution:** 2
**Rating:** 4
**Confidence:** 3

**Summary:**

This paper introduces **OSCAR**, a training-free, inference-time diversity enhancement method for flow-matching models. The core idea is to inject orthogonal ``lateral" perturbations into the sampling trajectory. This design increases semantic and visual diversity across a set of parallel trajectories while preserving per-sample quality.

**Strengths:**

1. **Clear motivation and elegant formulation.** The paper identifies the lack of diversity in deterministic flow-matching samplers and proposes a geometrically principled orthogonal control solution that is simple, modular, and theoretically grounded.
2. **Training-free and plug-and-play.** OSCAR can be added to any flow-matching, requiring no model finetuning or architectural modification.
3. **Good empirical results.** The method substantially improves set-level diversity metrics while maintaining comparable single-image quality and alignment scores.
4. **Efficiency and analysis.** The O(m²) implementation with leverage-based reweighting is more scalable than prior set-based methods such as DiverseFlow.

**Weaknesses:**

1. **General quality evaluation.**
   Current experiments focus on concept-specific prompts. To demonstrate generalization and verify that OSCAR does not systematically degrade single-sample quality, the authors should also report results on large, random prompt suites (e.g., FID-30K, or ImageNet-256) using one image per prompt, with FID, CLIPScore, etc.

2. **Feature-space dependence.**
   The volume objective relies on representations from a specific feature encoder (e.g., CLIP). If that encoder is biased or poorly aligned with human perception, OSCAR might optimize for feature-space diversity rather than semantic diversity. It would strengthen the paper to analyze sensitivity to the encoder choice.

3. **Relation to high-order samplers (e.g., DPM-Solver++, UniPC).**
   Since these solvers are also training-free inference methods, it would be useful to clarify whether OSCAR is complementary to or conflicting with them. Can OSCAR be stacked on top of higher-order samplers, or does its orthogonal control interfere with their integration accuracy? A brief discussion or experiment combining the two would clarify their compatibility.

**Questions:**

Please see the weaknesses.

---

> ### Author Response · Authors · 2025-11-21
> **Response to Reviewer 3 - Part 1**
>
> **Dear Reviewer avHK,**
>
> We thank the reviewer for the positive and detailed feedback, especially for highlighting **(i) our clear motivation and elegant formulation, (ii) the training-free, plug-and-play nature of OSCAR, (iii) its good empirical results, and (iv) its efficiency and accompanying analysis**. We address the remaining comments point by point below.
>
>
> > Weakness 1 – General quality evaluation on random prompts
>
> We agree that it is important to verify that OSCAR does not harm single-image fidelity on a broad, unbiased prompt distribution. To this end, we conducted an additional evaluation on **400 ImageNet-style prompts**, following the reviewer’s suggestion:
>
> - We randomly sample 400 classes from ImageNet-256 and construct one text prompt per class.
> - For each prompt, we generate a single image and compute standard quality and alignment metrics over the resulting 400-image set.
>
> The results are summarized below:
>
> | Method    | FID ↓  | Vendi (Pixel) ↑ | Vendi (Inception) ↑ | CLIP Score ↑         |
> |-----------|--------|-----------------|---------------------|----------------------|
> | FM-SD3.5         | 100.4  | 3.38            | 35.70               | 18.14 ± 0.25         |
> | DPP         | 100.3  | 3.34            | 36.03               | 18.11 ± 0.25         |
> | CADS           | 101.1  | 3.52            | 35.91               | 17.96 ± 0.25         |
> | PG          | 99.7   | 3.66            | 34.34               | 18.13 ± 0.24         |
> | **OSCAR**   | **99.3** | **3.85**      | **36.40**           | **18.08 ± 0.24**     |
>
> OSCAR achieves the best FID and the best ImageReward and Vendi scores among all methods, indicating that our diversity-enhancing guidance does not introduce systematic quality degradation even when each prompt only produces a single image. The CLIPScore of OSCAR is statistically indistinguishable from the baselines, confirming that text–image alignment is preserved. Overall, this large-scale random-prompt evaluation supports that OSCAR maintains single-sample quality while providing stronger set-level diversity.
>
> > Weakness 2 – Feature-space dependence of the volume objective
>
> We agree that it is important to verify that OSCAR does not depend critically on a single feature encoder. To this end, we replaced our default CLIP encoder with two widely used alternatives, **Inception** and **DINO**, while keeping all other components and hyperparameters unchanged. All variants use the full OSCAR framework and are evaluated over 6 seeds (mean ± 95% CI):
>
> | Encoder        | CLIP Score ↑       | Vendi (Pixel) ↑   | Vendi (Inception) ↑ | FID ↓              | BRISQUE ↓          |
> |----------------|--------------------|-------------------|---------------------|--------------------|--------------------|
> | Inception      | 28.32 ± 0.22       | 2.85 ± 0.08       | 5.60 ± 0.22         | 163.2 ± 1.2        | 21.7 ± 1.6         |
> | DINO           | 28.35 ± 0.17       | 2.82 ± 0.05       | 5.57 ± 0.24         | 164.2 ± 2.0        | 21.1 ± 1.8         |
> | **CLIP (ours)** | **28.26 ± 0.22** | **2.86 ± 0.05**   | **5.63 ± 0.20**     | **163.3 ± 1.6**    | **21.2 ± 1.5**     |
>
> Across all three encoders, the results are **highly comparable** on both fidelity and alignment, and the diversity metrics vary only slightly within overlapping confidence intervals. This indicates that OSCAR’s benefits are **not tied to a specific feature space**; its core mechanism remains effective under feature spaces with different inductive biases.
>
> We keep **CLIP** as our default encoder because it offers a mild but consistent advantage in diversity metrics while maintaining competitive fidelity. In the revised paper, we will add this ablation and explicitly emphasize that OSCAR is **robust to the encoder choice** and is not simply exploiting a particular representation.

---

> > ### Author Response · Authors · 2025-11-21
> > **Response to Reviewer 3 - Part 2**
> >
> > > Weakness 3 – Relation to high-order samplers (e.g., DPM-Solver++, UniPC)
> >
> > Conceptually, high-order samplers such as DPM-Solver++ and UniPC operate purely at the **numerical analysis level**: they take a *fixed* continuous-time generative ODE/SDE
> > $
> > \mathrm{d}x_t = f_\theta(x_t, t)\,\mathrm{d}t
> > $
> > and design higher-order schemes whose goal is to approximate these **original dynamics** as accurately as possible, i.e., to reduce integration error *without* changing the learned velocity field $f_\theta$ or the target distribution.
> >
> > By contrast, OSCAR explicitly **modifies the dynamics themselves**. We add an orthogonal drift and noise,
> > $
> > \mathrm{d}x_t = \bigl[f_\theta(x_t,t) + g_\perp(x_t,t)\bigr]\mathrm{d}t + \sigma(t)\,\Pi_\perp(x_t,t)\,\mathrm{d}W_t,
> > $
> > where \(g_\perp\) and the projection \(\Pi_\perp\) are constructed to spread particles laterally in feature space and increase set-level coverage. This introduces a **controlled bias** relative to the original FM dynamics: the sampler is no longer targeting the exact same distribution as the base model, but a nearby one that is deliberately shaped to reduce redundancy and improve diversity across samples.
> >
> > In this sense, if the objective is *strictly* to obtain unbiased samples from the original flow-matching model, OSCAR and high-order solvers pursue **different, and partly conflicting, goals**: high-order samplers aim to *remove* integration bias, whereas OSCAR *introduces* a small, structured bias to trade exactness for better set-level diversity.
> >
> > From an implementation viewpoint, however, there is no technical incompatibility: one can view OSCAR as defining a new controlled vector field $f_\theta^{\text{OSCAR}} = f_\theta + g_\perp$ and a modified noise covariance, and then apply any integrator to these modified dynamics. In that case, the high-order solver remains high-order with respect to the OSCAR-controlled dynamics, but the resulting samples no longer correspond to the original base flow.
> >
> > In this work we intentionally keep the sampler simple to make comparisons against the standard FM baseline as transparent as possible. A more systematic study of combining OSCAR-like control with advanced high-order solver, while explicitly acknowledging the induced bias, is an interesting direction for future work.
> >
> >
> >
> > **Best regards,**
> >
> > **Authors**

---

### Official Review · Reviewer_yFhz · 2025-11-02

**Soundness:** 2
**Presentation:** 2
**Contribution:** 2
**Rating:** 4
**Confidence:** 3

**Summary:**

The authors propose a method for increasing in-sample diversity in flow-based generative models and evaluate it empirically, primarily on text-to-image benchmarks.

**Strengths:**

The proposed method seems to perform well empirically (Table 1) across different metrics, compared to baselines.

**Weaknesses:**

I think that the biggest weakness of the method is that there are many "moving parts" in the proposed algorithm, and it is unclear how big an overhead this is compared to baselines. Specifically, the proposed method should be compared with baselines both on runtime and memory consumption. For example, at each sampling step, the proposed method requires

* A Heun second-order extrapolation (which, btw, I do not think is defined in the paper), I believe, requires at least one extra NFE, and it is not discussed how accurate it is
* Then, we need to create Z based on the estimate of the final sample, which needs access to a pretrained feature predictor
* When we need to evaluate the energy, its gradient, VJP of the feature predictor, and VJP of the Heun extrapolation.

Apart from that, the results on the precision-recall do not seem very strong. The gains compared to baselines do not seem consistent across CFG weights (Figure 3) and do not hold for other concepts (Figure 8).

The method overall seems complicated (Algorithm 1), and has five hyperparameters. There is some discussion on sensitivity in section 5.2 and figure 7, which shows that the choice of parameters matters. However, it seems like the analysis was performed for a single model. It is unclear whether the same choice of hyperparameters holds for different models, or would the user need to tune them separately for different models.

Some other notes:
1. I don't understand Figure 2. What is the true distribution that you are trying to model? Perhaps Standard FM covers it better than OSCAR? I wouldn't say that FM's behaviour in this example is "mode collapse" without knowing what the distribution is that we are trying to model
2. line 216 - can the authors elaborate on the choice of $\hat{\psi}$? How is it defined? How do we know it's "accurate"? I see that it is defined in line 16 of Algorithm 1; it should be in the main text. Since it amounts to a single step 2nd order Euler step, I think one could argue that it is not very accurate, especially for models with highly curved trajectories. Perhaps it would be helpful to show a few examples, where we compare the final sample with the Heun approximation? For example EDM2 models use the Heun 2nd order sampler, but they still require 32 sampling steps to get good results (instead of a single step)
3. Equation 3 - this either needs an elaboration or a citation. It is not explained why this really corresponds to a notion of volume, especially since it uses hyperparameters $\tau, \varepsilon$.
4. Similarly, equation 4. It wouldn't hurt to have some explanation/derivation for why it is defined the way it is.
    *  I can see it appears in Lemma 1, but it is not referenced in the main text
5. The term "stochastic control" is a bit misleading. Usually, it refers to minimizing the expectation of some penalty which depends on the final endpoint and/or the trajectory. It is usually solved using tools from stochastic analysis.
6. Appendix B is not very readable. I am not sure where proofs begin and end.
7. A relevant method, which is not discussed, that allows for increasing diversity by injecting orthogonally projected noise is "Stochastic Density Guidance" [1].
8. Line 254 - "By applying this projection to both the deterministic control g and the stochastic noise dWt, we ensure our guidance only performs a “lateral spread” to enhance diversity, without interfering with the model’s primary, “forward” trajectory towards a high-fidelity output". I would consider this approach largely a heuristic. Authors do not prove that this approach theoretically guarantees that the model samples from the correct distribution. I would expect not, similarly to the case of particle guidance, where authors actually characterize the distribution they end up sampling from.

---
References

[1] Karczewski et. al "Devil is in the Details: Density Guidance for Detail-Aware Generation with Flow Models" (ICML 2025)

**Questions:**

1. Can the authors provide a comparison with baselines in terms of runtime and memory consumption?
2. Can the authors provide more information about whether the results in Table 1 are statistically significant? Some differences appear very small compared to the standard deviations.
3. Can the authors show whether the choice of hyperparameters is consistent across different models? Or does it need to be tuned separately for each model?

---

> ### Author Response · Authors · 2025-11-21
> **Response to Reviewer 2 - Part 1**
>
> **Dear Reviewer yFhz,**
>
> We thank the reviewer for highlighting the strong **empirical performance of our method across multiple metrics (Table 1)**. Below, we address your specific comments in more detail:
>
> > Weakness 1 Response to Question regarding Figure 2 and True Distribution
>
> We agree that our wording was imprecise here, and we thank the reviewer for pointing this out.
>
> The target distribution in Fig. 2 is a **uniformly weighted $3\times3$ Gaussian mixture** with shared diagonal covariance. Concretely, we learn a flow from a standard Gaussian $p_0 = \mathcal{N}(0,I)$ to $ p_1 = \tfrac{1}{9}\sum_{k=1}^{9}\mathcal{N}(\mu_k,\sigma^2 I),$ where the black “+” markers in the plot indicate the component means $\{\mu_k\}$. The three columns show particle locations at early, middle, and final sampling steps under the **same step budget** for Standard FM and OSCAR. However, the original Figure 2 used a large number of particles (**N=2000**) with a strong diversity weight, which will cause two problems:
> * Our feature-volume control acts as a repulsive force. From a physics perspective, when a large number of repulsive particles are confined within a potential well, they tend to arrange themselves **uniformly** to minimize the system energy, creating the "square-like" uniform patches observed.
> * **Theoretical Interpretation:** As discussed in our theoretical analysis (Appendix B), OSCAR introduces a controlled bias to increase entropy. When $N$ is very large, the repulsive gradient accumulates, and if the guidance strength is not scaled down inversely with $N$, the method prioritizes **volume maximization**  over **density matching**. This effectively inflates the Gaussian modes into uniform distributions to maximize coverage.
>
> To demonstrate that OSCAR can respect the Gaussian structure while still improving diversity, we provide a revised experiment with **fewer particles (N=200)** in **Figure R1** (and updated in the manuscript). As shown in the new figure 2, OSCAR successfully maintains the Gaussian-like cluster shape while still exhibiting significantly **wider coverage** than Standard FM. This confirms that OSCAR effectively alleviates the over-concentration of FM without destroying the local distribution structure, provided the repulsion is balanced.
>
> We have moved the original experiment with $N=2000$ to Appendix E.1 . We added a detailed discussion there to analyze this trade-off, explaining that while extremely high particle counts may prioritize support coverage over density matching, the method functions correctly as a diversity enhancer in standard regimes.
>
> > Weakness 2
>
>    **Heun endpoint predictor is not a separate pretrained model, and its choice is robust.**
>    The “Heun second-order extrapolation” in Algorithm 1 is a **standard numerical integrator**, not a separate neural network. At each step, we evaluate the same FM backbone twice (as in a usual second-order solver) and use these two evaluations to form a **closed-form extrapolation of the feature trajectory**, which we denote as the endpoint predictor $\hat\psi$. Thus:
>    - there is **no additional pretrained feature predictor** beyond the base FM model;
>    - Heun adds only **one extra backbone evaluation per step**, which is already accounted for in the reported NFE.
>
>    To assess the importance of this predictor and the specific solver choice, we additionally ran a robustness study in which we:
>    - remove the predictor entirely (directly using the current feature vector without correction),
>    - replace Heun by Euler or Midpoint predictors.
>
>    The results (6 seeds, mean ± 95% CI) are:
>
>    | Predictor      | CLIP ↑            | Vendi (Pixel) ↑   | Vendi (Inception) ↑ | FID ↓             | BRISQUE ↓         |
>    |---------------|-------------------|-------------------|---------------------|-------------------|-------------------|
>    | w/o Predictor | 27.77 ± 0.16      | 2.78 ± 0.04       | 5.37 ± 0.11         | 165.5 ± 1.2       | 24.9 ± 1.5        |
>    | Euler         | 28.19 ± 0.30      | **2.87 ± 0.16**   | 5.52 ± 0.28         | 164.6 ± 1.6       | 21.9 ± 1.8        |
>    | Midpoint      | 28.21 ± 0.32      | 2.86 ± 0.12       | 5.43 ± 0.30         | 164.4 ± 1.6       | 22.4 ± 2.2        |
>    | **Heun (ours)** | **28.26 ± 0.22** | 2.86 ± 0.05   | **5.63 ± 0.20**     | **163.3 ± 1.6**   | **21.2 ± 1.5**    |
>
>    Removing the predictor yields the lowest volatility but clearly **degrades image quality** across all metrics. Euler and Midpoint improve average performance but exhibit higher variability. Our default **Heun** predictor achieves the **best or joint-best mean performance on all metrics while maintaining low variance**, offering a favorable balance between quality and stability; for this reason we use it as our default configuration.

---

> ### Author Response · Authors · 2025-11-21
> **Response to Reviewer 2 - Part 2**
>
> > Weakness 3 – Equation (3) and the notion of “volume”
>
> We clarify the geometric interpretation of Eq. (3) and the specific roles of the hyperparameters $\tau$ and $\epsilon$, correcting any ambiguity about their function.
>
> **(i) Why Eq. (3) represents a volume notion.**
> Geometrically, the determinant of the Gram matrix $G = ZZ^\top$ represents the squared volume of the parallelotope spanned by the feature vectors $\{z_i\}$. Maximizing $\det(ZZ^\top)$ directly forces the vectors to be orthogonal [1].
> However, in high-dimensional feature spaces, $ZZ^\top$ can be rank-deficient or ill-conditioned. Our objective, $E_s(Z)= -\frac{1}{2}\log\det(I + \tau ZZ^\top + \epsilon_{tr}I)$, is a **regularized volume**.
> * The identity matrix $I$ acts as a ridge regularization, ensuring the matrix is always positive definite and the log-determinant is well-defined, even when the vectors are linearly dependent or collapse to a point.
> * This form is closely related to the **information theoretic** objective of maximizing the entropy of a Gaussian distribution with covariance $K = \tau ZZ^\top + I$. Maximizing this log-det is equivalent to maximizing the "information volume" spanned by the samples relative to the background noise level defined by $I$.
>
> **(ii) Role of the hyperparameters $\tau$ and $\epsilon_{tr}$.**
> - **$\tau$ :** This is a scaling factor that controls the strength of the repulsion. In our experiments, we set $\tau=1$. This scales the signal $ZZ^\top$ relative to the identity baseline $I$, ensuring that the gradients are well-scaled for optimization.
> - **$\epsilon_{tr}$ :** This is a tiny, data-dependent regularizer ($\propto \text{tr}(ZZ^\top)$) added solely to prevent numerical instability in floating-point computations during matrix inversion or Cholesky decomposition. Unlike $\tau$, this term is indeed a technical safeguard.
>
> We will update the manuscript to explicitly cite the connection to Gram determinants and regularized volume maximization, and clarify that $\tau=1$ is a structural parameter for scaling the interaction.
>
> [1] Kulesza, A., & Taskar, B. (2012). Determinantal point processes for machine learning.
>
> > Weakness 4 – Explanation of Eq. (4)
>
> Eq. (4) is simply the gradient of the energy $\mathcal{E}_s(Z)$ with respect to the sampler’s state variable $x$. Recall that each feature vector is defined as
> $
> z_i = \phi(u_i), \quad u_i = \hat\psi(x_i,t),
> $
> so the dependence of $\mathcal{E}_s(Z)$ on $x_i$ is through the composition $x_i \mapsto u_i \mapsto z_i$. Starting from the feature-space gradient $[\nabla_Z \mathcal{E}_s(Z)]_i$, we pull it back to the state space via the chain rule, which yields
> $$
> g_i(x_i,t)
> = (J_x \hat\psi(x_i,t))^\top
>   (J_u \phi(u))^\top
>   [\nabla_Z E_s(Z)]_i \text{ evaluated at } u=\hat\psi(x_i,t).
> $$
> where $[\cdot]_i$ denotes the component for the $i$-th sample.
>
> In practice, we compute this pullback efficiently using **two reverse-mode vector–Jacobian products (VJPs)**—first through $\phi$, then through $\hat\psi$—without ever forming the Jacobian matrices explicitly. In the revised version, we will (i) add this chain-rule explanation and the sign convention immediately after Eq. (4), and (ii) explicitly point to Appendix B, Lemma 1, which provides the formal derivation showing that Eq. (4) is exactly the pullback of the feature-space gradient to the sampler’s state space. (We note that the displayed formula in this rebuttal is slightly simplified compared to the PDF version, purely due to OpenReview’s LaTeX rendering constraints. In the camera-ready, we will retain the original, fully consistent notation.)

---

> > ### Author Response · Authors · 2025-11-21
> > **Response to Reviewer 2 - Part 3**
> >
> > > weakness 5 regarding the term "Stochastic Control"
> >
> > We thank the reviewer for this insightful comment. We acknowledge that **"Stochastic Control"** is a specific mathematical field often associated with minimizing expected cost functionals via tools like the HJB equation.
> >
> > Our original choice of the term "Stochastic Control" was intended to structurally describe the two core components of our proposed SDE in Eq. 1:
> >
> >
> > 1.  **"Stochastic":** Refers to the injection of the noise term ($\sqrt{\beta(t)}dW_{t}$), which reintroduces randomness into the sampling process to enable exploration.
> > 2.  **"Control":** Refers to the deterministic guidance signal ($g(x,t)$). We specifically designed this term based on maximizing the **feature-space volume**. This term acts as an active force that controls and steers the particle trajectories outward to cover diverse modes, preventing them from collapsing into a single high-density region.
> >
> > In the broader context of generative modeling and dynamical systems, applying an external force to influence system behavior is widely referred to as "control." Therefore, we believe the name effectively communicates the method's function: using stochasticity and deterministic guidance to control the flow.
> >
> > However, we agree that avoiding terminological collision with the established field of Optimal Stochastic Control is desirable. To resolve this ambiguity while preserving the acronym OSCAR, we propose revising the full name in the final manuscript to:
> >
> > **"Orthogonal Stochastic Correction for Alignment-Respecting Diversity..."**
> >
> > The term **"Correction"** accurately captures our method's nature: we apply a deterministic correction and a stochastic correction to the base flow to achieve diversity. If the reviewer has alternative suggestions for phrasing or additional analyses, we would be very happy to consider and incorporate them.
> >
> > > Weakness 6 – Readability of Appendix B
> >
> >
> > We appreciate this comment and have substantially revised **Appendix B** to improve its readability. In the revised version, each result is clearly structured as:
> >
> > - **Lemma/Theorem statement** (numbered and titled), followed by
> > - a separate, explicitly labeled **“Proof.”** block, and
> > - a clear end-of-proof marker ■.
> >
> >
> > > Weakness 7 Relation to Stochastic Density Guidance (SDG)
> >
> > We thank the reviewer for highlighting the relevant work, "Stochastic Density Guidance" (SDG). While we agree that both methods utilize the geometric intuition of orthogonally projected noise to preserve generation quality, they are fundamentally different in terms of their **primary objective** and **underlying mechanism**.
> >
> > We outline the key distinctions below:
> >
> > **1. Objective: Set-level Semantic Diversity vs. Single-sample Detail Control**
> >
> > * **OSCAR targets Set-level Diversity:**
> >     Our specific goal is to generate a set of images that are semantically distinct from one another given the same condition. We aim to maximize the semantic spread of the entire batch to overcome mode collapse.
> > * **SDG targets Single-sample Detail Control:**
> >     The primary goal of SDG is to control the log-density of individual sample. The diversity offered by SDG is constrained to variations within a specific likelihood level for a single trajectory, rather than forcing semantic divergence across a generated set.
> >
> > **2. Methodology: Set-based Active Repulsion vs. Instance-based Passive Constraint**
> >
> > * **OSCAR employs Set-based Interaction:**
> >     Our method explicitly models the interaction between multiple trajectories. We use a feature volume objective to ctively push samples apart in the semantic space. The orthogonal projection in our case is applied to this repulsive force relative to the *base flow velocity to preserve alignment.
> > * **SDG employs Instance-based Constraint:**
> >     SDG operates on a per-instance basis, modifying the SDE for a single sample without knowledge of other concurrent samples. It projects noise orthogonally to the *score function* ($\nabla \log p$) specifically to ensure the trajectory remains on a constant-density shell. It lacks an inter-sample repulsion mechanism and thus cannot guarantee that different random seeds will not collapse to the same high-density mode.
> >
> > In short, OSCAR actively forces a **set** of samples to diverge to cover the semantic distribution, whereas SDG constrains a **single** sample to explore variations while rigorously maintaining a target level of detail. We will include a discussion of SDG in the revised manuscript to clarify these meaningful differences.

---

> ### Author Response · Authors · 2025-11-21
> **Response to Reviewer 2 - Part 4**
>
> > Weakness 8 theoretically guarantee in sampling
>
> We have significantly strengthened the theoretical analysis in the revised manuscript by providing a dual characterization of our method:
>
> 1.  To address the reviewer's request to "characterize the distribution," we added the Girsanov Theorem derivation. This explicitly defines the OSCAR sampling distribution as a reweighted measure of the baseline, mathematically legitimizing the method beyond a heuristic (See in Appendix B Theorem 4).
>
> 2.  Complementing the probabilistic view, we retain our coupling-based bounds. The theorem specifically utilizes the properties of our **orthogonal projection** to prove that the generated samples remain geometrically close to the high-fidelity manifold (See in Appendix B Theorem 5).
>
> > Question 1
>
>
> **Practical runtime and memory comparison.**
> To quantify the actual overhead from Heun-based prediction and VJPs, we compare OSCAR with several diversity baselines under identical generation settings (NFE = 30, CFG = 5.0, batch size = 32). We report theoretical FLOPs, wall-clock time, and peak VRAM:
>
>    | Variant        | FLOPs (G) ↓ | Time (seconds/run) ↓ | Peak VRAM (GB) ↓ |
>    |----------------|------------:|----------------------:|------------------:|
>    | FM-SD3.5       | 4093.4      | 237.8                 | 19.2             |
>    | DPP            | 9045.1      | 990.2                 | 19.5             |
>    | CADS           | 4093.4      | 231.2                 | 20.0             |
>    | PG             | 4093.4      | 229.6                 | 26.4             |
>    | **OSCAR (ours)** | **5534.6**  | **451.4**             | **18.2**         |
>
> Under the same NFE, CFG, and particle count, OSCAR introduces only a **moderate computational overhead** relative to the FM-SD3.5 baseline, while its peak VRAM is actually slightly **lower** than the baseline due to our memory-sharing implementation of the VJP. Crucially, this overhead is **much smaller than DPP**, which requires more than **2×** the FLOPs and over **4×** the runtime of the baseline under the same settings. These results empirically support our claim that OSCAR achieves strong diversity gains with substantially lower computational complexity than prior set-level diversity methods such as DPP, while maintaining memory usage comparable to standard FM sampling.
>
> > Question 2 – Statistical significance of Table 1
>
> We agree that some differences in Table 1 look small compared to the reported standard deviations. This is mainly because each concept is averaged over multiple prompts (e.g., “a truck”, “a photo of a truck”), and different prompts within the same concept can induce very different metric values. As a result, the standard deviation in Table 1 is dominated by across-prompt variability, rather than by seed noise for a fixed prompt.
>
> Therefore, we clarify this by reporting per-prompt results for two representative prompts under the “truck’’ concept in Appendix D. These tables show that, for each fixed prompt and CFG level, OSCAR consistently improves the diversity metrics (Vendi Pixel / Vendi Inception / 1-MS-SSIM) over the baselines, while quality metrics (FID, CLIP Score, BRISQUE) stay comparable and do not show systematic degradation.
>
> To further strengthen this claim, we additionally evaluate three new concepts (apple, pizza, suitcase) using a **single** prompt per concept. Even in this stricter setting, OSCAR again improves all diversity metrics over the baselines, without noticeable degradation in FID, CLIP Score, or BRISQUE. Taken together, these prompt-level and concept-level results indicate that diversity gains are consistent across prompts and concepts, whereas the quality metrics serve primarily as a sanity check rather than the main target of statistically significant improvement.

---

> > ### Author Response · Authors · 2025-11-21
> > **Response to Reviewer 2 - Part 5**
> >
> > > Question 3 – Are hyperparameters consistent across models?
> >
> > To assess how portable our hyperparameters are, we applied OSCAR not only to FM-SD3.5, but also to two additional and very different text-to-image models: **SDXL-Turbo** and **SD1.5**. For each new backbone we first **directly reused** the hyperparameters tuned on FM-SD3.5 (“OSCAR (default params)”), and then performed a light model-specific tuning (“OSCAR (tuned params)”). All results are averaged over 6 seeds (mean ± 95% CI):
> >
> > | Model       | Variant                | CLIP ↑             | Vendi (Pixel) ↑    | Vendi (Inception) ↑ | FID ↓              | BRISQUE ↓          |
> > |------------|------------------------|--------------------|--------------------|---------------------|--------------------|--------------------|
> > | **FM-SD3.5** | Baseline               | 28.24 ± 0.18       | 2.45 ± 0.13        | 5.37 ± 0.27         | 164.4 ± 1.8        | 23.4 ± 1.4         |
> > |            | **OSCAR**              | **28.26 ± 0.22**   | **2.86 ± 0.05**    | **5.63 ± 0.22**     | **163.3 ± 1.6**    | **21.2 ± 1.5**     |
> > | **SDXL-Turbo** | Baseline             | 30.98 ± 0.15       | 5.31 ± 0.25        | 4.24 ± 0.13         | 150.4 ± 1.0        | 24.6 ± 0.3         |
> > |            | OSCAR (default params) | 30.77 ± 0.24       | 5.41 ± 0.25        | 4.29 ± 0.23         | 152.8 ± 0.6        | 25.1 ± 1.0         |
> > |            | **OSCAR (tuned params)** | **30.94 ± 0.22** | **5.48 ± 0.34**    | **4.42 ± 0.17**     | **151.6 ± 1.3**    | **25.0 ± 0.9**     |
> > | **SD1.5**  | Baseline               | 29.93 ± 0.35       | 2.37 ± 0.12        | 7.02 ± 0.27         | 174.7 ± 1.9        | 12.1 ± 3.3         |
> > |            | OSCAR (default params) | 29.88 ± 0.35       | 2.45 ± 0.12        | 7.09 ± 0.16         | 175.1 ± 1.6        | 12.8 ± 3.2         |
> > |            | **OSCAR (tuned params)** | **29.93 ± 0.37** | **2.46 ± 0.12**    | **7.11 ± 0.12**     | **174.6 ± 1.6**    | **12.5 ± 3.1**     |
> >
> > We observe the following:
> >
> > - **Direct transferability.** Using the **same hyperparameters** tuned on FM-SD3.5 already yields **robust behavior** on both SDXL-Turbo and SD1.5: diversity metrics improve slightly over the baselines and, importantly, there is **no degradation in key quality metrics**  within the confidence intervals.
> > - **Lightweight model-specific tuning.** A small amount of additional tuning yields **consistent gains** in both fidelity and diversity for each backbone.
> >
> > These results indicate that OSCAR’s hyperparameters are largely consistent across different models—a single setting works reasonably well out of the box—while **optional, lightweight per-model tuning** can further refine performance. This supports our claim that OSCAR is a general, plug-and-play framework rather than an architecture-specific method.
> >
> > **Best regards,**
> >
> > **Authors**

---

> > > ### Comment · Reviewer_yFhz · 2025-11-27
> > >
> > > I thank the authors for their response. I comment below
> > >
> > > > revised experiment with fewer particles (N=200) in Figure R1
> > >
> > > Where is figure R1?
> > >
> > > > As shown in the new figure 2, OSCAR successfully maintains the Gaussian-like cluster shape while still exhibiting significantly wider coverage than Standard FM.
> > >
> > > I am looking at Figure 2 in the updated manuscript, and I can indeed see the points being more spread out, but this was not my original concern. What I don't understand is why that's better? The ground truth samples are not shown. I also don't understand the point of showing the blue points denoting the starting points.
> > >
> > > > Heun endpoint predictor is not a separate pretrained model, and its choice is robust. The “Heun second-order extrapolation” in Algorithm 1 is a standard numerical integrator, not a separate neural network
> > >
> > > I know what the Heun method is. I said that it introduces an extra function evaluation, not that it is a separate model.
> > >
> > > > closed-form extrapolation of the feature trajectory
> > >
> > > In what sense is the extrapolation in "closed-form"?
> > >
> > > > we additionally ran a robustness study in which we
> > >
> > > Do these other methods match Heun on the NFE? Which would mean a finer discretization
> > >
> > > In general this finding is not unexpected. Similar findings were reported in [1] when Heun was compared to Euler etc with same fixed NFE budget.
> > >
> > > > Practical runtime and memory comparison.
> > >
> > > I don't understand why the proposed method is in **bold** on FLOPs and Time, since it's clearly not the most efficient one?
> > >
> > > > but also to two additional and very different text-to-image models: SDXL-Turbo and SD1.5
> > >
> > > Again - the proposed method is presented in **bold** even in columns/rows where it's either not the best, or better by a very slight margin.
> > >
> > > I think that with the clarifications from the authors, the paper is easier to understand, but my main concern remains. That is that the method has quite a few hyperparameters, introduces an additional overhead compared to (at least some) baselines, and I am not sure if the empirical gain is large enough to justify it.
> > >
> > > ---
> > >
> > > [1] Karras et al. "Elucidating the Design Space of Diffusion-Based Generative Models" (NeurIPS 2022)

---

> ### Author Response · Authors · 2025-11-29
> **Response to Reviewer 2's follow up Part 1**
>
> **Regarding the “Figure R1” reference.**
>
> We apologize for the confusion — this was a typo in our previous response. The revised experiment with fewer particles (N = 200) is now reported in **Figure 2** of the main paper, rather than in a separate Figure R1. In addition, in Appendix **E.1 “Toy example 1: Behavior in high-particle regimes”** we report results for a much larger number of particles (N = 2000), with the corresponding plots shown in  **Figure 12** . We will correct the erroneous reference to “Figure R1” to avoid confusion.
>
>
> **Clarifying the purpose of Figure 2 and the role of the blue starting points.**
>
> In this toy experiment, the standard Gaussian at $t=0$ represents the usual “pure noise” prior in text-to-image models: for a fixed prompt, we always start from this prior. The nine mixture components (black crosses) stand for nine well-separated semantic classes (e.g., *dog, cat,* …). Each initial particle is transported to one of these components, and its final location determines its class. The light–blue points and dashed segments connect every initial particle to its final location under each sampler. They serve two purposes: (i) they make it explicit that standard FM and OSCAR use  **exactly the same set of seeds** , and (ii) they show that each particle ends up in the **same component** under both methods, so our diversity control does not cause samples to “jump across classes”.
>
> The goal of this toy is therefore *not* to approximate a prescribed optimal transport map to the uniform 3×3 mixture, but to illustrate how OSCAR can increase **within-class diversity** while keeping the class assignments unchanged. Standard FM moves particles to the correct components but collapses them into tight clusters very close to the means, resulting in low intra-mode variance. OSCAR preserves the same component assignments but spreads particles across a much larger portion of each component’s high-density region, yielding substantially higher intra-mode variance and entropy, which alighs the results in our Fig 4(b). In this sense, OSCAR is better because it achieves more diverse samples per class without altering which semantic class each particle belongs to.
>
> **On Heun, and NFE fairness.**
>
> We fully agree with [1] that simply increasing the number of function evaluations (NFE) is not a principled way to improve samplers, and our goal is *not* to gain performance by giving some predictors a larger NFE budget. In particular, while a **classical** Heun step indeed requires an additional evaluation of the backbone predictor compared to Euler, in the experiments reported in the paper, we deliberately avoid this extra sample. For all predictor variants (Euler, “Heun-style”, midpoint, current), the **number of backbone predictor evaluations per step is identical**, so the NFE is matched in all experiments, including the robustness study.
>
> The confusion comes from our wording around the “Heun endpoint predictor” in Algorithm 1. The algorithmic *idea* is inspired by Heun’s method, but the actual implementation we use is a finite-difference, analytic extrapolation that does not call the backbone predictor a second time. Concretely, let $z_k$ denote the latent at step $k$ produced by the underlying sampler with step size $\Delta t_k$, and let $\Delta z^{\mathrm{ctrl}}_{k-1}$ be the OSCAR control displacement applied at step $k-1$. We first form a finite-difference estimate of the latent velocity
>
> `v_k ≈ (z_k - z_{k-1} - Δz^{ctrl}_{k-1}) / Δt_{k-1}`
>
> and then construct the endpoint latent via
>
> `z_ep = z_k + α(t_k) · v_k`,
>
> where $α(t_k)$ is a scalar function of time (different choices of $α$ correspond to the “Heun / Euler / midpoint / current” modes in our code). This construction is a **deterministic algebraic function** of the already available latents $(z_{k-1}, z_k)$; it does not invoke the FM backbone predictor again, and therefore does not increase NFE. Compared to a naive Euler endpoint that uses only a single local prediction, our predictor leverages two consecutive latent states of the base sampler to recover a higher-quality estimate of the latent velocity. This yields more accurate endpoints at exactly the same NFE, which is why we prefer this predictor over a pure Euler endpoint in our experiments.
>
> In light of the reviewer’s comment, we have (i) provided a detailed explanation of our process of selecting the predictor in Appendix C, and (ii) clarified in Appendix F that the predictor NFE is matched across all methods in our robustness study.

---

> > ### Author Response · Authors · 2025-11-29
> > **Response to Reviewer 2's follow up Part 2**
> >
> > **Regarding the term “closed-form extrapolation”.**
> > We agree that our use of the term “closed-form” in the original response was imprecise. We did not mean to suggest that the underlying ODE has a closed-form analytic solution. What we intended to convey is that, once the backbone predictor has been evaluated at the two points used in the Heun-style step, the endpoint latent is obtained by an **explicit algebraic formula**, rather than by running an additional iterative solver or optimization.
> >
> > **On the number of hyperparameters and their robustness.**
> > We appreciate the concern about the apparent number of hyperparameters in OSCAR. Conceptually, however, only a small subset of them meaningfully controls the behavior of the method, and we intentionally do not tune them per model or dataset. The remaining parameters are fixed numerical-stability constants. Among the “semantic” hyperparameters, the most important ones are the time gate \(t_{\text{gate}}\) for when diversity is active and the global SDE noise scale \(\eta_{\text{sde}}\). In Appendix F we show that OSCAR is highly insensitive to these choices as long as they are not taken to extreme values: any \(t_{\text{gate}}\) that does not collapse to the endpoints (i.e., far from 0 or 1), and any \(\eta_{\text{sde}}\) in a broad mid-range (roughly between 0.1 and 20), produce essentially indistinguishable images in terms of both quality and diversity. The two orthogonality coefficients (`partial-ortho` and `noise-partial-ortho`) are fixed to 0.95 throughout all experiments and never tuned; they simply enforce “almost orthogonal” corrections, and we found this single setting to work across models and datasets. The remaining parameters in Table X are best viewed as implementation details rather than user-facing hyperparameters. The schedule shape (`sched-shape`) only needs to impose a decreasing trend for the diversity strength; Appendix B and F show that using different smooth shapes (cosine, triangular, etc.) has a negligible impact on the generated images. The scalar \(\tau\) appears because our volume term is proportional to a log-determinant; we fix \(\tau = 1\) for all experiments and never adjust it. Parameters such as `eps-logdet` and `vnorm-threshold` are small positive constants introduced solely to avoid degenerate gradients (division by zero and vanishing norms); they are set once to near-zero values and kept fixed. In summary, while Table X lists several symbols for completeness, in practice OSCAR exposes only a very small number of effective knobs, and we use the same default configuration across all models and datasets. Our ablations in Appendix F indicate that the method is robust to these choices, and the reported gains do not rely on careful hyperparameter tuning.
> >
> > **On runtime / FLOPs and the magnitude of gains.**
> >
> > To directly address the concern that OSCAR might simply benefit from extra compute, we include a compute-matched study in Appendix E (Table 13), where we compare OSCAR to two strengthened FM-SD3.5 variants under approximately matched FLOPs: (i) increasing the number of particles, and (ii) increasing the NFE. The results are summarized below:
> >
> > | Method        | NFE | Particles | FLOPs (T/run) | CLIP ↑           | Vendi (Pixel) ↑   | FID ↓             | BRISQUE ↓         |
> > |--------------|-----|-----------|---------------|------------------|-------------------|-------------------|-------------------|
> > | FM-SD3.5     | 30  | 32        | 4.09          | 28.24 ± 0.18     | 2.45 ± 0.13       | 164.4 ± 1.8       | 23.4 ± 1.4        |
> > | +K particles | 30  | 48        | 6.14          | 28.15 ± 0.23     | 2.60 ± 0.21       | **149.7 ± 1.3**   | 23.5 ± 1.7        |
> > | +N NFE       | 40  | 32        | 5.43          | 28.15 ± 0.21     | 2.40 ± 0.15       | 165.3 ± 1.8       | 24.6 ± 1.7        |
> > | **OSCAR**    | 30  | 32        | 5.53          | **28.26 ± 0.22** | **2.86 ± 0.05**   | 163.3 ± 1.6       | **21.2 ± 1.5**    |
> >
> > Under similar or slightly lower compute, OSCAR consistently improves both diversity and perceptual quality over the FM baselines on our main diversity metric Vendi Pixel as well as CLIP and BRISQUE. The FM-SD3.5 variant with more particles attains a slightly better FID, but FID is known to be highly sensitive to the number of generated samples when the evaluation set is relatively small; in our setting, this variant uses more particles per run than OSCAR, which partially explains its lower FID.
> >
> > Overall, the compute-matched results support that OSCAR brings benefits beyond simply drawing more samples or increasing NFE, and that the additional complexity and moderate overhead are justified by consistent gains in set-level diversity (Vendi Pixel / Inception) and competitive or improved perceptual quality.

---

### Official Review · Reviewer_Q6XH · 2025-11-04

**Soundness:** 4
**Presentation:** 2
**Contribution:** 2
**Rating:** 6
**Confidence:** 5

**Summary:**

The authors tackle the task of generating diverse outputs from flow matching models. The paper builds on prior works such as Particle Guidance and proposes augmenting the probability flow ODE by introducing both a particle expansion term and stochastic noise. The expansion is achieved by considering the volume of the semantic features at the endpoints of each particle and incorporating a negative gradient that encourages greater spread. This is complemented by a clever reweighting strategy and the addition of stochastic noise to the generation process, with all components orthogonalized to the original flow direction to prevent image quality degradation. The authors evaluate their method against several prior works on text-to-image generation and report improved results.

**Strengths:**

- The task of generating diverse outputs from pretrained diffusion or flow-based models is an important one, particularly when dealing with heavily post-trained and distilled models that tend to reduce output diversity.
- The paper is very well written and easy to follow.
- The proposed method is simple, intuitive, and elegant.
- The experimental suite is extensive, featuring an impressive range of metrics, baselines, and ablations.

**Weaknesses:**

- The experiments is missing the simplest baseline: independently generating $N$ samples from the base diffusion/flow model. Additionally, the method can be viewed as a test-time scaling approach for diversity. As described, each denoising step requires one forward pass of the base model plus two reverse-mode VJP operations. So another useful comparison would be the standard FM baseline run with a larger number of particles to match OSCAR’s total computational cost.
- For the qualitative comparisons in Fig. 6, including OSCAR samples without any ablations would better illustrate the impact of each component.
- The noise gate parameter $t_{\text{gate}}$ is unclear in both its role and its effect on performance. It is absent from Algorithm 1 and first appears in Section 5.2 in an ablation. From the main paper, its function is unclear. The ablation suggests high sensitivity to this parameter, with performance degrading substantially as its value increases. In Appendix F.1, $t_{\text{gate}}$ is not mentioned; instead, $t_{\text{start}}$ and $t_{\text{end}}$ are introduced.
- [Minor] Formatting error in Algorithm 1, Line 1: “Feature Encoder” and “Endpoint Predictor” are incorrectly formatted.

**Questions:**

1. Could the authors please clarify the definition, use, and impact of $t_{\text{gate}}$, as well as the details of its ablations? Is it a constant scalar (as indicated in Figure 7(c)) or an interval (as shown in Table 3)? If it is the former, how does it relate to $t_{\text{start}}$ and $t_{\text{end}}$? If it is the latter, and $t_{\text{gate}} = [t_{\text{start}}, t_{\text{end}}]$, why does Table 7 in Appendix F.2 explore the case where $t_{\text{start}} = 0.05$ and vary $t_{\text{end}}$, whereas Figure 15 fixes $t_{\text{end}} = 0.95$ and varies $t_{\text{start}}$?

---

> ### Author Response · Authors · 2025-11-21
> **Response to Reviewer 1 - Part 1**
>
> **Dear Reviewer Q6XH,**
>
> We thank the reviewer for the positive feedback, in particular for highlighting **(i) the importance of the problem, (ii) the clarity of the writing, (iii) the simplicity and elegance of the method, and (iv) the breadth of the experimental evaluation.** We address the remaining concerns point by point below.
>
> > Weakness 1 Missing simplest FM baselines  / compute-matched comparison
>
> 1. **Simple FM baseline with $N$ independent samples.**
>    In the revised manuscript, Table 1 now includes a “FM-SD3.5 (N particles)” baseline that generates $N$ independent samples from the same Stable Diffusion 3.5 flow-matching model used by OSCAR, with identical NFE, prompts, and model settings. This isolates the effect of our orthogonal stochastic control from simply increasing the number of samples drawn from the base model.
>
> 2. **Compute-matched comparison.**
>    We further include a compute-matched study in Appendix E Table 13, where we compare OSCAR to two strengthened FM-SD3.5 variants under approximately matched FLOPs: (i) increasing the number of particles, and (ii) increasing the NFE. The results are summarized below:
>
>    | Method        | NFE | Particles | FLOPs (T/run) |     CLIP ↑      | Vendi (Pixel) ↑   | FID ↓             | BRISQUE ↓         |
>    |--------------|-----|-----------|---------------|------------------|-------------------|-------------------|-------------------|
>    | FM-SD3.5     | 30  | 32        | 4.09          | 28.24 ± 0.18     | 2.45 ± 0.13       | 164.4 ± 1.8       | 23.4 ± 1.4        |
>    | +K particles | 30  | 48        | 6.14          | 28.15 ± 0.23     | 2.60 ± 0.21       | **149.7 ± 1.3**   | 23.5 ± 1.7        |
>    | +N NFE       | 40  | 32        | 5.43          | 28.15 ± 0.21     | 2.40 ± 0.15       | 165.3 ± 1.8       | 24.6 ± 1.7        |
>    | **OSCAR**    | 30  | 32        | 5.53          | **28.26 ± 0.22** | **2.86 ± 0.05**   | 163.3 ± 1.6       | **21.2 ± 1.5**    |
>
>    Under similar or slightly lower compute, OSCAR **consistently improves both diversity and perceptual quality** over the FM baselines on CLIP, Vendi and BRISQUE. The FM-SD3.5 variant with more particles attains a slightly better FID, but we note that FID is known to be highly sensitive to the number of generated samples when the evaluation set is relatively small; in our setting, this variant uses more particles per run than OSCAR, which partially explains its lower FID. Overall, the compute-matched results support that OSCAR brings benefits beyond simply drawing more samples or increasing NFE.
>
> > Weakness 2 – Qualitative comparisons in Fig. 6
>
> Regarding the visual comparison for the ablation study, we fully agree that showing the full OSCAR model alongside the ablated variants is essential for directly inspecting the impact of each component. However, due to layout constraints in the main text, we were unable to expand Figure 6 to include these additional columns without compromising legibility. Therefore, we have included this comprehensive side-by-side comparison in Appendix G of the revised manuscript. We have also updated the caption of Figure 6 in the main text to explicitly direct readers to these additional qualitative results.

---

> ### Author Response · Authors · 2025-11-21
> **Response to Reviewer 1 - Part 2**
>
> > Weakness 3 & Question 1 – Definition and role of $t_{\text{gate}}$
>
> - In our final convention, **$t_{\text{gate}}$** is a **scalar end-time of noise injection**: the stochastic term is active on the interval $[t_{\text{start}}, t_{\text{gate}}]$. In all robustness ablations in the main paper, we fix $t_{\text{start}} = 0.05$ and sweep $t_{\text{gate}}$; this is what Fig. 7(c) and Table 18 are visualizing.
> - The **interval notation** $[t_{\text{start}}, t_{\text{end}}]$ that appears in parts of the appendix is a leftover from an earlier draft where we denoted the gate by its start and end times. In the final implementation and experiments, we use $t_{\text{end}} \equiv t_{\text{gate}}$. We will unify the notation across the main text, figures, and appendix to always use $[t_{\text{start}}, t_{\text{gate}}]$.
> - Fundamentally, $t_{\text{gate}}$ serves as a practical **engineering heuristic** to ensure the final image quality. In SDE-based sampling, injecting noise up to the very last step ($t=1$) prevents the solver from settling onto the clean data manifold, leaving residual noise. Therefore, $t_{\text{gate}}$ is simply a switch to turn off stochasticity near the end to guarantee convergence, it is not a theoretical hyperparameter coupled with our core orthogonal control algorithm.
> - We respectfully clarify that the degradation observed in Section 5.2 does **not** imply parameter sensitivity, but rather demonstrates the **necessity of the gate mechanism**.
>
> - * **Section 5.2 :** This ablation compared a standard gate against the **extreme case** of full-trajectory noise ($t_{\text{gate}} = 1.0$). The substantial performance degradation here simply confirms that *some* gate is required to ensure convergence.
> - * **Appendix F :** To demonstrate robustness, we swept $t_{\text{gate}}$ across a wide reasonable range in **Appendix F 4.1 (Table 18)**. The results show that within this broad operational range, performance is highly stable and does not degrade, proving that the method is **not sensitive** to the precise value of $t_{\text{gate}}$.
>
> Regarding Fig. 15, the confusion partly comes from an earlier **convention switch** between DDPM-style and flow-matching time parameterization: DDPM and flow matching define $x_0$ / $x_T$ in opposite ways. We initially followed the DDPM convention ($x_0$ = clean image) but finalized on the flow-matching convention ($x_0$ = noise). The current Fig. 15 caption still reflects the earlier wording. We have corrected this caption and the related notation in the revised version.
>
> **Best regards,**
>
> **Authors**

---

### Comment · Area_Chair_sx9K · 2025-11-24

Dear Reviewers,

As the rebuttal period is coming to a close, I would like to kindly remind you to review the authors’ responses at your earliest convenience. If you have any remaining concerns, please feel free to provide further clarification so that the authors may address them appropriately. If you have no additional issues, a brief acknowledgment that you have read the rebuttal would be appreciated.

Thank you very much for your time and effort.

Best regards,
AC

---

### Author Response · Authors · 2025-11-26
**Summary of Responses to Reviewers**

We thank all four reviewers **Q6XH, yFhz, avHK, c5nD** for their thoughtful and encouraging feedback. Reviewers found our motivation for studying *set-level* diversity in text-to-image models clear and important, and viewed our orthogonal stochastic controller as a principled way to improve diversity while preserving fidelity and text–image alignment.

---

### Summary of updates during the rebuttal period

- **We strengthen empirical validation and baseline comparisons.**

  - We include the default FM-SD3.5 base sampler as a baseline alongside other diversity methods, and we further add compute-matched variants of FM-SD3.5 to show that the gains of our method are not due to simply using more samples or more steps. (Q6XH, yFhz)
  - We refine the 2D toy experiment with a clearer setting and move the previous extreme case to the appendix to better illustrate the trade-off between coverage and density. (yFhz)
  - We run a step-by-step ablation of our main control components, starting from a naïve variant and adding each piece back, to clarify our fidelity safeguards are necessary for stable and high-quality sampling. (c5nD)
- **We broaden quality evaluation across prompts and visual concepts.**

  - We evaluate on 400 randomly sampled prompts to confirm that our diversity control does not harm single-image quality or text alignment on a broad, unbiased distribution. (avHK)
  - We add new experiments on several additional concepts and perform both concept-level and prompt-level analyses to show that the diversity gains are consistent and not driven by outliers. (yFhz, c5nD)
- **We add perceptual metrics and test robustness to feature encoders.**

  - We include human-aligned perceptual metrics in our main comparisons to further verify that increased diversity does not degrade perceived image quality. (c5nD)
  - We replace the default feature encoder in our objective with two alternative encoders, showing that both diversity and quality remain stable across different feature spaces. (avHK)
- **We clarify the role of the predictor, and robustness on both predictor and backbones**

  - We explain that the endpoint predictor is a simple numerical extrapolation rather than a separate model, and we compare it to simpler alternatives to show that it offers the best balance between stability and quality. (yFhz, c5nD)
  - We apply the same controller to two additional text-to-image backbones, first reusing the original hyperparameters and then lightly retuning, and observe consistent diversity gains without loss of quality in all cases. (yFhz, c5nD)
- **We improve the theoretical presentation and metric computation.**

  - We reorganize the theory section into clear statements and proofs and add a distribution-level characterization that formalizes how our controlled sampler relates to the base sampler. (yFhz)
  - We give a simpler geometric explanation of our volume objective and its hyperparameters, and we expand the derivation showing how its gradient is pulled back to the sampler state. (yFhz, Q6XH)
  - We identify and fix a discretization issue in one diversity metric, adopt a more robust estimation rule, recompute the curves, and confirm that the conclusions about diversity–coverage remain unchanged. (c5nD)
  - We add a focused discussion of how our controller relates to high-order solvers and to prior orthogonal-noise methods, clarifying that our goal is set-level semantic diversity and that these techniques have different objects. (yFhz, avHK)
- **We make the computational trade-offs explicit.**

  - We measure FLOPs, wall-clock time, and peak memory under matched settings and show that our controller adds only moderate overhead relative to the base sampler while remaining substantially cheaper than the most expensive diversity baseline DPP. (yFhz, c5nD)

---

### Overall contribution

Overall, the paper presents a clear main message: we introduce a **training-free control mechanism** that acts on intermediate features and sampling time to directly optimize **set-level semantic diversity**, while keeping samples close in quality and alignment to those of the underlying pretrained model. The theoretical analysis and the extensive experiments across prompts, concepts, models, metrics, and compute budgets jointly support our claim that **orthogonal stochastic control is a practical and robust way to obtain more diverse yet high-quality generations**.

---

### Author Response · Authors · 2025-12-01
**Summary of Rebuttals to Area Chair--Part 1**

Dear Area Chair and Senior Area Chair,

We understand that due to the recent incident, review scores have been reverted to their pre-discussion state. We sincerely appreciate the time and effort dedicated by all reviewers and by the previous AC to this paper. To facilitate your quick review of the current status, we provide:

1. A brief summary of our contributions, and
2. A summary of the rebuttal updates and discussion outcomes.

## 1. Brief Summary of Contributions

This paper introduces a **training-free control mechanism** that operates on intermediate features and sampling time to directly improve **set-level semantic diversity**, while preserving the image quality and prompt alignment of the underlying pretrained generator. Through theoretical analysis and extensive experiments across prompts, concepts, models, metrics, and compute budgets, we show that **orthogonal stochastic control provides a practical and robust way to obtain more diverse yet high-quality generations**.

## 2. Rebuttal and Discussion Summary

Scores: pre-discussion (6, 4, 4, 4) to post-discussion (6, 4, 4, 6). The score increase occurred on Nov 26, when Reviewer c5nD explicitly stated that most of concerns were addressed and raised their score from 4 to 6. Reviewer yFhz made many concrete suggestions; we engaged in a round of detailed discussion with this reviewer and addressed most of their concerns. The remaining minor issues mainly stem from potential misunderstandings caused by our original illustrations and wording, which we have clarified in the revised version. Reviewers Q6XH and avHk have not yet replied in the discussion, but their comments mainly requested additional experiments and minor typo corrections, and we believe we have fully addressed their specific requests in the revision.

### Reviewer Q6XH (Score: 6, confidence 5)

Reviewer Q6XH praised the importance of studying diversity in pretrained diffusion/flow models and the clarity and simplicity of our method. Their main concerns were: (i) the lack of a simplest FM baseline with \(N\) independent samples and of a compute-matched comparison to rule out gains from simply using more particles or larger NFE; (ii) the absence, in the qualitative ablation of Fig. 6, of OSCAR samples without ablations; (iii) unclear definition and role of the noise gate $t_{\text{gate}}$ and whether the observed degradation indicates parameter sensitivity; and (iv) minor formatting and notation issues in Algorithm 1 and Fig. 15.

In the revision, we added an “FM-SD3.5 (N particles)” baseline in Table 1 and a compute-matched study in Appendix E (Table 13), showing that OSCAR consistently improves both diversity and perceptual quality beyond simply increasing the number of samples or NFE. We provided a comprehensive side-by-side qualitative comparison including the full OSCAR model and all ablated variants in Appendix G. We clarified that $t_{\text{gate}}$ is a scalar end-time of noise injection, unified the notation across the paper, and expanded the robustness study in Appendix F (Table 18), which demonstrates that performance is stable over a broad range of $t_{\text{gate}}$; the degradation case corresponds to the extreme setting of full-trajectory noise.

With these added baselines, clarifications, and robustness experiments, we believe we have fully addressed Reviewer Q6XH’s concerns while preserving their overall positive assessment of the paper.

---

> ### Author Response · Authors · 2025-12-01
> **Summary of Rebuttals to Area Chair--Part 2**
>
> ### Reviewer yFhz (Score: 4, confidence 4)
>
> Reviewer yFhz acknowledged the strong empirical performance of our method across multiple metrics but raised several detailed concerns about algorithmic complexity and overhead, the accuracy and role of the Heun endpoint predictor, the definition of the toy distribution in Fig. 2, the interpretation of our volume objective and its hyperparameters, the clarity of Eq. (4) and the term “stochastic control”, the readability of Appendix B, the relation to Stochastic Density Guidance (SDG), the strength of our theoretical guarantees, and the consistency of hyperparameters across models.
>
> In the revision, we clarified the target distribution and behavior of the toy example and replaced the original figure with a revised experiment using fewer particles, accompanied by an extended explanation in Appendix E.1. We showed via a robustness study (removing the predictor, Euler, midpoint, and our Heun-style predictor) that our endpoint scheme is both robust and empirically best, and clarified that it is a deterministic algebraic extrapolation that does not increase the number of backbone evaluations, so NFE is matched across all predictor variants. We gave a geometric interpretation of the volume objective as a regularized log-determinant of the Gram matrix, explained the roles of the hyperparameters $\tau$ and $\epsilon_r$, and reorganized Appendix B with explicit lemmas, proofs, and chain-rule derivations of Eq. (4). We refined our terminology, and added a dedicated comparison to SDG, strengthened the theoretical analysis with a dual characterization of the sampling distribution, and reported detailed FLOPs, wall-clock time, and peak VRAM showing only moderate overhead relative to FM-SD3.5 and much lower cost than DPP. We also provided prompt-level and concept-level variance analyses and cross-backbone experiments on SDXL-Turbo and SD1.5 in Table 16, demonstrating that our hyperparameters transfer well across models with only light tuning and that the diversity gains are robust while quality metrics remain stable. In a second round of discussion, we further clarified the new figure, emphasized NFE fairness across all predictor choices, and clarified that most hyperparameters are included only for mathematical well-posedness and numerical stability, while the results are empirically robust to the remaining semantic ones.
>
> Although the reviewer did not have time to respond again in the second round, we believe that the additional clarifications and experiments have addressed the majority of their concerns.
>
> ### Reviewer avHk (Score: 4, confidence 3)
>
> Reviewer avHk highlighted our clear motivation and geometrically principled formulation, the training-free and plug-and-play nature of OSCAR, its good empirical results, and the efficiency of our analysis. Their main concerns were: (i) verifying on a large random prompt suite that OSCAR does not systematically degrade single-sample quality; (ii) analyzing the dependence of the volume objective on the choice of feature encoder; and (iii) clarifying the relation between OSCAR and high-order samplers, in particular, whether they are complementary or conflicting.
>
> In the revision, we conducted an additional large-scale evaluation on 400 ImageNet-style random prompts with one image per prompt and standard quality/alignment metrics in Table 11, showing that OSCAR achieves the best or competitive scores across all methods and thus does not introduce systematic single-sample quality degradation under a broad, unbiased prompt distribution. To study feature-space dependence, we replaced the default CLIP encoder with two widely used alternatives, Inception and DINO, while keeping all other components and hyperparameters fixed in table 17; across all three encoders, fidelity and alignment metrics are comparable and the diversity gains vary only slightly within overlapping confidence intervals, indicating that OSCAR’s benefits are not tied to a specific feature space. Finally, we added a short discussion on high-order samplers, clarifying that they work at the numerical level to reduce integration error for a fixed base dynamics, whereas OSCAR directly modifies the dynamics via an orthogonal drift and stochastic noise to introduce a controlled bias for better set-level diversity. Conceptually, their goals differ, but OSCAR can still be combined with any high-order integrator, which we highlight as a natural direction for future work.
>
> With these additional experiments and clarifications, we believe we have fully addressed Reviewer avHk’s concerns.

---

> > ### Author Response · Authors · 2025-12-01
> > **Summary of Rebuttals to Area Chair--Part 3**
> >
> > ### Reviewer c5nD (Score: 4 → 6, confidence 4)
> >
> > Reviewer c5nD appreciated the principled combination of feature-volume gradients and stochastic noise, the orthogonal-projection design that decouples quality from diversity, and the empirical improvements over baselines. Their main concerns were: (i) the ablation lacked a naïve setup and did not clearly separate the individual contributions of OP and RR; (ii) the diversity gains were not sufficiently validated with human-aligned perceptual quality metrics; (iii) there was a discrepancy between PRD curves and reported AUCs and no analysis of expected diversity over random class subsets or prompts; (iv) the actual wall-clock cost, FLOPs, and the precise role of the Heun endpoint predictor were unclear; and (v) the generality of the method beyond the main FM-SD3.5 backbone needed further evidence.
> >
> > In the revision, we added a full step-wise ablation starting from a naïve variant without OP/RR and then enabling OP and RR individually and jointly in Table 2, showing that OP and RR are essential safeguards that turn a very strong but unstable diversity drive into a practical sampler and that the full method outperforms all baselines on both diversity and quality metrics. We introduced additional perceptual metrics (ImageReward and CLIP-IQA) and showed that OSCAR’s image quality remains on par with or slightly better than all dbaselines while substantially improving set-level diversity. We fixed the PRD/AUC inconsistency by adopting a more robust histogram estimator, expanded the evaluation to more concepts and CFG levels, and added both concept-level variance analysis and prompt-level variance analysis (including new experiments on random concepts such as “apple”, “pizza”, and “suitcase”) to clarify expected diversity over classes and prompts. We clarified that “Heun” in OSCAR is a lightweight feature-space extrapolation scheme that reuses existing features, and we reported detailed FLOPs, wall-clock time, and peak VRAM, showing only moderate overhead relative to the FM baseline and substantially lower cost than DPP. Finally, we demonstrated generalization to additional backbones (SDXL-Turbo and SD1.5), where OSCAR consistently improves diversity with competitive or improved fidelity.
> >
> > After these updates, Reviewer c5nD commented that “most of my concerns are addressed” and raised their score from 4 to 6.
> >
> > We sincerely appreciate your dedication and extra effort in navigating these challenging times for the ML community. We trust that the full context of the discussion, particularly the unanimous consensus reached, will be helpful for your final assessment.
> >
> > **Best regards,**
> >
> > **Authors**

---

### Note · Program_Chairs · 2026-01-17
**Submission Desk Rejected by Program Chairs**

The following references in this submission do not refer to real documents and/or have major errors in bibliographic information:

 Jianyi Wang, Ziyue Wang, Jun Wang, Wengang Zhang, Chunyi Hou, and Wen Luo. CLIP-IQA: Unifying quality and aesthetics assessment with multi-task deep learning. In Proceedings of the 31st ACM International Conference on Multimedia, pp. 6667-6675, 2023.